# Diurnal variations in oxygen and nitrogen isotopes of atmospheric nitrogen dioxide and nitrate: implications for tracing NO$_x$ oxidation pathways and emission sources

Sarah Albertin[1,2], Joël Savarino[2], Slimane Bekki[1], Albane Barbero[2], Roberto Grilli[2], Quentin Fournier[3], Irène Ventrillard[3], Nicolas Caillon[2], Kathy Law[1]

[1]LATMOS/IPSL, Sorbonne Université, UVSQ, CNRS, 75005 Paris, France.
[2]IGE, Univ. Grenoble Alpes, CNRS, IRD, Grenoble INP, 38000 Grenoble, France.
[3]LIPhy, Univ. Grenoble Alpes, CNRS, 38000 Grenoble, France.

*Correspondence to*: Sarah Albertin (sarah.albertin@univ-grenoble-alpes.fr)

**Abstract.** The oxygen ($\Delta^{17}O$) and nitrogen ($\delta^{15}N$) isotopic compositions of atmospheric nitrate (NO$_3^-$) are widely used as tracers of its formation pathways, precursor (nitrogen oxides (NO$_x$) $\equiv$ nitric oxide (NO) + nitrogen dioxide (NO$_2$)) emission sources, and physico-chemical processing. However, the lack of observations on the multi-isotopic composition of NO$_2$ maintains significant uncertainties regarding the quantitative links between the isotopic composition of NO$_x$ and NO$_3^-$, which ultimately may bias inferences on NO$_3^-$ formation processes and distribution of sources, particularly in winter urban atmospheres. We report here on the first simultaneous atmospheric observations of $\Delta^{17}O$ and $\delta^{15}N$ in NO$_2$ (n = 16) and NO$_3^-$ (n = 14). The measurements were carried out at sub-daily ($\sim$ 3 h) resolution over two non-consecutive days in an Alpine city in February 2021. A strong diurnal signal is observed in both NO$_2$ and NO$_3^-$ multi-isotopic composition. $\Delta^{17}O$ of NO$_2$ and NO$_3^-$ ranges from 19.6 ‰ to 40.8 ‰ and from 18.3 ‰ to 28.1 ‰, respectively. During the day and night, the variability of $\Delta^{17}O(NO_2)$ is mainly driven by the oxidation of NO by ozone, with a substantial contribution from peroxy radicals in the morning. NO$_3^-$ mass balance equations, constrained by observed $\Delta^{17}O(NO_2)$, suggest that during the first day of sampling, most of NO$_3^-$ was formed locally from the oxidation of NO$_2$ by hydroxyl radicals by day, and via heterogeneous hydrolysis of dinitrogen pentoxide at night. For the second day, calculated and observed $\Delta^{17}O(NO_3^-)$ do not match, particularly daytime values; the possible effects on $\Delta^{17}O(NO_3^-)$ of a Saharan dust event that occurred during this sampling period and of winter boundary layer dynamics are discussed. $\delta^{15}N$ of NO$_2$ and NO$_3^-$ ranges from $-10.0$ ‰ to 19.7 ‰ and from $-4.2$ ‰ to 14.9 ‰, respectively. Consistent with theoretical predictions of N isotope fractionation, the observed variability of $\delta^{15}N(NO_2)$ is explained by significant post-emission equilibrium N fractionation. After accounting for this effect, vehicle exhaust is found to be the primary source of NO$_x$ emissions at the sampling site. $\delta^{15}N(NO_3^-)$ is closely linked to $\delta^{15}N(NO_2)$ variability, bringing further support to relatively fast and local NO$_x$ processing. Uncertainties on current N fractionation factors during NO$_2$ to NO$_3^-$ conversion are underlined. Overall, this detailed investigation highlights the potential and necessity to use

simultaneously $\Delta^{17}O$ and $\delta^{15}N$ in $NO_2$ and $NO_3^-$ in order to better constrain quantitative inferences on the sources and
formation chemistry of $NO_3^-$ in urban environments in winter.

## 1    Introduction

Despite extensive efforts in emission controls in recent decades, global anthropogenic emissions of nitrogen oxides ($NO_x \equiv$
nitrogen monoxide (NO) + nitrogen dioxide ($NO_2$)) remain more than two orders of magnitude higher than before the
Industrial Revolution (Hoesly et al., 2018). Atmospheric nitrate ($NO_3^- \equiv$ nitric acid ($HNO_3$) + particulate nitrate ($p$-$NO_3^-$)), is
the main end-product of $NO_x$ oxidation and a key component of fine particulate matter (PM), which adversely affects human
health (WHO, 2021) and contributes to climate change (Masson-Delmotte et al., 2021). $NO_3^-$ can be transported far from
emission sources and can be removed from the atmosphere through dry and wet deposition within hours to days (Alexander
et al., 2020; Park et al., 2004). The additional input of this "reactive" nitrogen ($N_r$) into natural environments is known to
have detrimental consequences, particularly regarding biodiversity and water quality (Galloway et al., 2008; Vitousek et al.,
1997). It is therefore important to have a comprehensive understanding of $NO_x$ emission sources and oxidation processes, on
which effective air quality and climate change mitigation strategies rely (e.g., Bauer et al., 2007; Huang et al., 2014; Shah et
al., 2018; Tsimpidi et al., 2008; Wang et al., 2013, 2020).
Several studies noted that the response of $NO_3^-$ concentration in air to $NO_x$ emission reduction is contrasted,
particularly in winter (e.g., Shah et al., 2018; Tørseth et al., 2012; Wang et al., 2020; Zhou et al., 2019). This is because a
variety of factors controls the $NO_x$ conversion efficiency and the $NO_3^-$ content in PM, including precursor emission sources,
complex multiphase chemical reactions with other reactive species, and environmental conditions (e.g., temperature, relative
humidity, solar radiation) (Zhang et al., 2015). It remain difficult to assess the contribution of each parameter to the non-
linear $N_r$ chemistry, which is partly driven by close links between changes in aerosol acidity, gas-particle partitioning, and
atmospheric oxidation capacity (Shah et al., 2018; Fu et al., 2020; Shi et al., 2019; Li et al., 2021b). Ozone ($O_3$) and
hydroxyl radicals (OH) (Finlayson-Pitts and Pitts, 2000), are the major oxidants in the atmosphere whose chemical cycles
are largely controlled by solar radiation. As a result, there are significant diurnal and seasonal variations in $NO_x$ chemistry
(e.g., Prabhakar et al., 2017; Alexander et al., 2020). Notably, $NO_3^-$ formation is generally dominated by homogeneous OH
oxidation and heterogeneous $O_3$ chemistry during day/summer and night/winter, respectively (Alexander et al., 2020).
However, assessing the relative contributions of individual formation channels, together with their sensitivity to
environmental parameters, is not straightforward and requires extensive in situ observations combined with modelling tools
(e.g., (Alexander et al., 2020; Brown et al., 2006; Newsome and Evans, 2017; Xue, 2022; Prabhakar et al., 2017).

Upon release into the atmosphere, $NO_x$, mainly emitted as NO, undergoes oxidation to form $NO_2$. During the day, a
rapid photochemical equilibrium is established between NO and $NO_2$, known as the "photostationary state" (PSS; Leighton,
1961), via key interconversion reactions (Reactions R1−R3):

$$NO_2 + h\nu \xrightarrow{M} O(^3P) + NO \qquad\qquad R1$$

$$O(^3P) + O_2 \xrightarrow{M} O_3 \text{ with } M = N_2 \text{ or } O_2 \qquad\qquad R2$$

$$NO + O_3 \rightarrow NO_2 + O_2 \qquad\qquad R3$$

This cycle can be disturbed by peroxy radicals ($RO_2 \equiv$ hydroperoxyl radical ($HO_2$) + methyl peroxy radical ($CH_3O_2$)) via
typically Reaction (R4):

$$NO + RO_2 \rightarrow NO_2 + RO \qquad\qquad R4$$

Note that in polluted atmospheres where $NO_x$ mixing ratios often exceed ppb levels, Reaction (R4) followed by Reactions
(R1)−(R2) lead to the formation of $O_3$ (Crutzen, 1979). Although the role of $RO_2$ in $NO_x$ oxidation is crucial in $O_3$ formation
and $NO_x$ oxidation rate, measuring $RO_2$ mixing ratio remains challenging due to the need for state-of-the art instrumentation
coupled with photochemical models to establish chemical budgets (e.g., Ren et al., 2006; Tan et al., 2018). While NO is
usually oxidised relatively quickly into $NO_2$ in summer due to the high levels of $O_3$ and solar radiation, the shorter day
length and lower temperature in winter result in a contrasted $NO_x$ cycling. In particular, the formation of a temperature
inversion at the surface can trap pollutants emitted close to the surface in a shallow layer for hours to days (e.g., Largeron
and Staquet, 2016; Olofson et al., 2009). Under those conditions, it is not uncommon for $O_3$ levels to be very low due to
quasi-complete titration by NO, which can have further impacts on the atmospheric oxidation capacity. However,
pronounced $O_3$ pollution episodes may also arise in winter in highly polluted areas, such as in oil-producing regions due to
intense VOCs emissions (Edwards et al., 2014) or in China, where pollution control strategies mainly target $NO_x$ while
VOCs emissions remain more or less constant (Ren et al., 2022). In urban areas, $NO_2$ is generally mainly removed from the
atmosphere by reaction with OH during the day via Reaction (R5) (Dentener and Crutzen, 1993):

$$NO_2 + OH \xrightarrow{M} HNO_3(g) \qquad\qquad R5$$

$NO_2$ can also react with $O_3$ to form nitrate radicals ($NO_3$) via Reaction (R6):

$$NO_2 + O_3 \xrightarrow{M} NO_3 + O_2 \qquad\qquad R6$$

However, $NO_3$ is rapidly photolyzed during the day regenerating back $NO_2$ (Wayne et al., 1991). Another important $NO_3$
loss reaction is that with NO in polluted environments (Brown and Stutz, 2012). At night, without photolytic activity and
lower precursor emissions, the lifetime of $NO_3$ radicals substantially increases. $NO_3$ reacts with $NO_2$ to form dinitrogen
pentoxide ($N_2O_5$; Reaction R7), which then undergoes heterogeneous hydrolysis to form $HNO_3$ (Reaction R8):

$$NO_3 + NO_2 \overset{M}{\leftrightarrow} N_2O_5 \qquad\qquad\qquad\qquad\qquad R7$$

$$N_2O_5 + H_2O(aerosol) \rightarrow 2\,HNO_3(aq) \qquad\qquad\qquad\qquad\qquad R8$$

Reaction (R7) is temperature dependent, so $N_2O_5$ can eventually decompose to reform $NO_2$ and $NO_3$, with the $N_2O_5/NO_3$
ratio being negatively correlated with temperature. $N_2O_5$ is an important nocturnal sink for $NO_x$, notably in winter in urban
atmospheres due to high aerosol loads and low temperatures. However, the efficiency of Reaction (R8) is difficult to
determine because it strongly depends on parameters such as the aerosol surface density and its chemical composition
(Brown et al., 2006), which are not often well characterised. In addition, $NO_3$ can react with hydrocarbons to produce $HNO_3$,
which could significantly contribute to the formation of $NO_3^-$ in industrialised regions with high hydrocarbon emissions
(Brown et al., 2011). It is estimated that the Reactions (R1) to (R8) lead to the formation of 82 % of $NO_3^-$ near the surface on
a global scale (Alexander et al., 2020). In polluted environments, the respective contributions of Reaction (R5) (OH
pathway) and Reactions (R6)−(R8) ($N_2O_5$ pathway) are more contrasted and are still debated (e.g., Chan et al., 2021; Fu et
al., 2020). In addition, the reaction of $N_2O_5$ with chlorine on aerosols can contribute to $NO_3^-$ production in urban
atmospheres (Thornton et al., 2010), with further impacts on $O_3$ production in continental polluted atmosphere in winter
(Wang et al., 2019a). Other reactions, such as those involving halogen and organic intermediates, may become significant for
$NO_3^-$ production in specific regions, such as in polar, oceanic, and coastal areas (Alexander et al., 2020; Penkett et al., 2007;
Savarino et al., 2013; Simpson et al., 2015).

To help better constrain the atmospheric $N_r$ chemistry and budget, the last three decades have seen a growing interest in
stable oxygen (O) and nitrogen (N) isotopes, notably in $NO_3^-$ (Elliott et al., 2019; Savard et al., 2018). The isotopic
composition is reported as an isotopic enrichment ($\delta$) with respect to a reference material, defined as $\delta = (R_{sample}/$
$R_{reference} - 1)$, and expressed in per mill (‰). $R$ refers to the elemental abundance ratio of the heavy isotope to the light
isotope (e.g., $^{18}O/^{16}O$; $^{17}O/^{16}O$; $^{15}N/^{14}N$) in the sample, and in an international isotopic reference material (Vienna Standard
Mean Ocean Water for O; Li et al., 1988), and atmospheric $N_2$ for N; Mariotti, 1984). A powerful tool to help trace the
relative importance of different $NO_x$ to $NO_3^-$ oxidation pathways is through the use of the $^{17}O$-excess ($\Delta^{17}O = \delta^{17}O - 0.52 \times$
$\delta^{18}O$). $\Delta^{17}O$ is transferred to $NO_3^-$ by $O_3$ which possesses a very unique $\Delta^{17}O$ ((26.2 ± 1.3) ‰; Vicars and Savarino, 2014)
due to mass-independent fractionation during its formation process (Thiemens, 2006). In comparison, the $\Delta^{17}O$ of other
atmospheric oxidants such as OH is near zero due to isotopic exchange with atmospheric water vapor (Dubey et al., 1997).
Similarly, as the isotopic anomaly of atmospheric $O_2$ is very close to 0 ‰ (Barkan and Luz, 2003), and since $RO_2$ are mostly
produced by the reactions $R + O_2$ and $H + O_2$, $\Delta^{17}O$ of $RO_2$ can be considered negligible (Alexander et al., 2020). Therefore,
$\Delta^{17}O$ in $NO_3^-$ represents a unique tracer of the $O_3$ implication in its formation processes, that can provide valuable
constraints on the relative contributions of individual reactions (e.g., Morin et al. 2011, Alexander et al., 2009; Michalski et
al., 2003). By a simple mass balance calculation of O atoms in $NO_3^-$, the $^{17}O$-excess of $NO_3^-$ produced by an individual $NO_2$
to $NO_3^-$ conversion process $i$ (($\Delta^{17}O(NO_3^-)_i$) can be expressed as:

$$\Delta^{17}O(NO_3^-)_i = \frac{2}{3} \times \Delta^{17}O(NO_2) + \frac{1}{3} \times \Delta^{17}O(\text{add. O})_i \tag{1}$$

where $\Delta^{17}O(NO_2)$ is the $^{17}O$-excess of atmospheric $NO_2$ and $\Delta^{17}O(\text{add. O})_i$ is the transferrable $^{17}O$-excess of the oxidant
responsible for the conversion of $NO_2$ in $NO_3^-$ (Michalski et al., 2003). From Eq. (1), if $\Delta^{17}O(NO_2)$ is constrained, one can
derive individual $\Delta^{17}O$ transfer to $NO_3^-$ relative to a $i$ conversion process and compare this value with observed $\Delta^{17}O(NO_3^-)$.

Recent studies in urban areas have attempted to interpret the variability of $\Delta^{17}O(NO_3^-)$ in aerosols in order to quantify
the relative contribution of homogeneous and heterogeneous processes to $NO_3^-$ formation (e.g., Fan et al., 2023, 2022; He et
al., 2020, 2018; Lim et al., 2022; Wang et al., 2023, 2019b; Kim et al., 2023; Zhang et al., 2022b; Li et al., 2022b). However,
to that end, it is necessary to have a clear quantitative understanding of the transfers of $\Delta^{17}O$ in the $N_r$ cycle. To date, due to
very limited observational data, there is a lack of well-establish knowledge on the dynamics of $\Delta^{17}O$ in $NO_2$, the key
intermediate species in the formation of $NO_3^-$. Consequently, strong assumptions about $\Delta^{17}O(NO_2)$ have to be made when
interpreting $\Delta^{17}O(NO_3^-)$ measurements, which could potentially lead to biased conclusions. Notably, the most difficult
regions for the interpretation of $NO_3^-$ records are potentially polluted areas where the isotopic composition of $NO_2$ is
expected to be highly variable in space and time. Most studies typically estimate $\Delta^{17}O(NO_2)$ during the day by assuming that
an isotopic steady state (ISS) is reached between $NO_x$ and $O_3$ resulting in $\Delta^{17}O(NO_2)$ depending only on the relative
contributions of different oxidants to NO oxidation. A recent study reported the first in situ observations of $\Delta^{17}O(NO_2)$ in an
urban environment (Grenoble, France) in spring (Albertin et al., 2021). Time resolved $NO_2$ sampling (ca. 3 h) during 24 h
revealed a strong diurnal cycle in $\Delta^{17}O(NO_2)$, reaching ca. 40 ‰ during the day and decreasing down to ca. 20 ‰ at night.
The observed $\Delta^{17}O(NO_2)$ values and dial variability were consistent with its expected behaviour derived from $\Delta^{17}O$ mass
balance equations under the ISS assumption during the day. However, this first $\Delta^{17}O(NO_2)$ dataset is very short. More
measurements are needed to test in different environments and season the applicability of this new isotopic tool and to assess
whether the ISS is still valid. At night, the low $\Delta^{17}O(NO_2)$ measured by Albertin et al. (2021) is consistent with the oxidation
of freshly emitted NO by $O_3$. Nonetheless, since the timescale for the oxidation of $NO_2$ into $NO_3^-$ is thought to exceed the
duration of the night (Alexander et al., 2020), it is also common to assume that the isotopic composition of nocturnal $NO_2$
reflects more daytime formation and conditions of the previous days. While this assumption may hold true in remote areas
(Morin et al., 2011), significant uncertainties subsist in urban areas where the nighttime $NO_3^-$ chemistry may be more

efficient. In such circumstances, the production of $NO_3^-$ from $NO_2$ formed at night would lead to a lower than expected $\Delta^{17}O$ transfer to $NO_3^-$. For these reasons, the dual survey of the O isotopic composition of $NO_2$ and $NO_3^-$ would certainly help to accurately interpret $\Delta^{17}O(NO_3^-)$ observations in polluted atmospheres, particularly with sampling at sub-daily time scales which would allow to study the diurnal dynamics of $\Delta^{17}O(NO_2)$ and its links with $\Delta^{17}O(NO_3^-)$.

In addition to $\Delta^{17}O$, $\delta^{15}N$ in $NO_3^-$ ($\delta^{15}N(NO_3^-)$) can be used as a tracer of $NO_3^-$ sources and/or chemical processing. As different $NO_x$ emission sources have often distinct $\delta^{15}N$-fingerprints depending on the $NO_x$ production mechanism (Heaton, 1990; Felix et al., 2012; Fibiger and Hastings, 2016; Walters et al., 2015a, b; Yu and Elliott, 2017; Miller et al., 2018), $\delta^{15}N(NO_3^-)$ is a potentially valuable tool to trace the origins of its gaseous precursor. However, due to N fractionation effects associated with physico-chemical processing, $\delta^{15}N$ is altered during the conversion of $NO_x$ to $NO_3^-$ (Elliott et al., 2019). Therefore, the variability of $\delta^{15}N(NO_3^-)$ can be attributed to: (1) a change in $NO_x$ emission sources and (2) N isotopic fractionations between NO and $NO_2$, between $NO_2$ and $NO_3^-$, and during the transport of $NO_3^-$. These effects co-exist with relative contributions varying according to environmental conditions and the mix of $NO_x$ emissions. Numerous observations in diverse environments have emphasised the substantial influence of N fractionation effects in altering the original $^{15}N$ composition of emitted gaseous $NO_3^-$ precursors (e.g., (Bekker et al., 2023; Chang et al., 2018; Geng et al., 2014; Luo et al., 2023; Vicars et al., 2013; Li et al., 2021a). Although some N fractionation factors are available from calculations (Walters and Michalski, 2015) and laboratory experiments (Li et al., 2020; Walters et al., 2016), there is still a lack of observational constraints on the magnitude of the N isotopic partitioning between $NO_x$ and $NO_3^-$, which could lead to biased interpretations of $\delta^{15}N(NO_3^-)$ observations.

Following the preliminary work of Albertin et al. (2021), this study presents for the first time the simultaneous measurements of the atmospheric $NO_2$ and $NO_3^-$ multi-isotopic compositions. The sampling took place at high temporal resolution (~ 3 h) in late February 2021 in an urban Alpine city. $\Delta^{17}O/\delta^{15}N$ data of $NO_2$ and $NO_3^-$, meteorological parameters, and atmospheric observations (NO, $NO_2$, $O_3$, and PM) are collated in order to investigate diurnal $N_r$ chemistry, N fractionation effects and $NO_x$ emissions. Our winter measurements extend the atmospheric $NO_2$ multi-isotopic composition record which is only composed of spring measurements performed during a single day by Albertin et al. (2021). The general aim of this case study is to test $NO_2$-based isotopic approaches for tracing the origins and fate of $NO_x$, for instance in urban areas on sub-daily time scales. The added value of $\Delta^{17}O(NO_2)$ measurements in $N_r$ chemistry studies is more critically assessed here than in Albertin et al. (2021) through the use of accurate $NO_x$ measurements. Besides, using the isotopic theoretical framework developed previously, we explore the potential benefits of combining isotopic observations of $NO_2$ and $NO_3^-$ to gain a more detailed and quantitative on the links between atmospheric $N_r$ chemistry processes and variability in $NO_2$ and $NO_3^-$ isotopic composition. The framework used in inferring dominant $NO_x$ emission sources from $NO_2$ $\delta^{15}N$ measurements is also tested.

## 2 Material and methods

### 2.1 Study site and sample collection

The study was conducted in February 2021 in Chamonix-Mont-Blanc, France, (45°55′21″ N, 6°52′11″ E; altitude 1035 m above sea level). This narrow (~2 km wide on average in Chamonix) 23 km Alpine valley of about 12,000 inhabitants is surrounded by high-elevation mountains. The city can experience severe PM pollution events during the winter season, mainly due to wood-combustion for domestic heating and road traffic (Chazette et al., 2005; Quimbayo-Duarte et al., 2021; Weber et al., 2018; Aymoz et al., 2007). The study's sampling site was located at a CNRS (Centre National de la Recherche Scientifique) facility in a residential area, 1.2 km south of the Chamonix city centre, and 1.4 km north of the Mont-Blanc tunnel. Ambient air monitoring inlets and off-line gas and aerosol samplers were installed on the facility's terrace, 3 m above the ground level. Over the campaign, the surface was partly covered with snow.

Atmospheric particles (aerosols) were collected using a high-volume sampler (Digitel®, DH77, total suspended particle inlet, 1 m³ min⁻¹) and glass fiber (GF) filters (Whatman®, 150 mm-diameter). Evaluating the collection efficiency of total $NO_3^-$ has long been debated (e.g., Schaap et al., 2002; Appel et al., 1980) and, although not free from sampling artefacts (e.g., potential volatilisation of $HNO_3$ after exposure to ambient air), GF filters have been used on several times to study nitrate isotopes, mainly in coastal sites (e.g., Savarino et al., 2007; Michalski et al., 2003; Morin et al., 2009; Frey et al., 2009; Morin et al., 2007a; Patris et al., 2007; Vicars et al., 2013). Under these conditions, the aerosol alkalinity is supposed to allow the collection of $HNO_3$ (Prospero and Savoie, 1989). In our case, as the ambient air in Chamonix is expected to be free of sea salt, the GF filters were not washed before use in order to keep the initial NaCl coating inherited from the filter manufacturing process. Therefore, in addition to $p$-NO$_3^-$, we are confident that the high GF filter NaCl loading allowed the quantitative collection of $HNO_3$ at our site, as it has been shown previously in the literature (Appel et al., 1981; see also our reply and data to the comments of Reviewer #2 on this specific issue). Two field blanks were performed to evaluate the initial content of trace elements and possible contamination during handling. Atmospheric $NO_2$ was collected concurrently to filter samples using a pre-cleaned honeycomb denuder tube coated with a mixture of 2.5 M KOH (in methanol) and ultrapure guaiacol inserted into a ChemComb® 3500 speciation cartridge (Thermo Scientific®, USA). A second coated denuder was placed in series into the cartridge to check for $NO_2$ breakthrough. After sampling, denuders were rinsed with 10 mL of deionised water to solubilised trapped $NO_2$. Detailed information on the denuder sampling protocol is available in Albertin et al. (2021). Similar to blank filters, two blank denuders were performed. Blank filters and denuders were subjected to same handling, storage, and analytical treatment as field samples. Filters and denuder extractions were stored and transported frozen to IGE (Grenoble, France) for analysis.

Following the objective to investigate the diurnal isotopic composition of $NO_2$ and $NO_3^-$, denuder and filter samplings were conducted continuously for 24 hours with sampling time steps ranging from 1:30 to 7:30. During the day, denuder and

filter samplings were synchronised. At night, two filter samplings were performed while three sets of denuder tubes were collected. This sampling protocol was conducted during two non-consecutive days, from 19 February 2021 21:00 local time (LT) to 20 February 2021 21:00 LT (sampling period #1 = SP 1) and from 24 February 2021 7:30 LT to 25 February 2021 7:30 LT (sampling period #2 = SP 2).

## 2.2 Chemical and isotopic analysis

Concentrations of major ions from filter extractions in deionised water were determined by ion chromatography (Thermo Scientific™ Dionex™ Integrion™ HPIC). Reported to the total filter surface, the $NO_3^-$ contribution from blank filters represented on average $(8 \pm 9)$ % of sampled $NO_3^-$. Atmospheric mass concentrations (expressed in µg m$^{-3}$) were calculated as the ratio of the total ion filter loading (corrected for the blank contribution) to the total volume of air pumped through the filter at STP conditions. $NO_2^-$ concentration in denuder extractions were first estimated using the Griess-Saltzmann reaction and UV–Vis spectrometry at 544 nm. Even though the eluted matrix can interfere with colorimetric analyses, measured concentrations on first denuder tubes were relatively well correlated with ambient $NO_2$ measurements during atmospheric sampling and allowed to give indications on field blanks and on the volume needed to perform isotopic analysis.

Isotopic analyses were performed using an isotope ratio mass spectrometer (IRMS, Thermo Finnigan™ MAT 253) for analyses of $^{15}N/^{14}N$, $^{17}O/^{16}O$, and $^{18}O/^{16}O$ in $NO_3^-$ and $NO_2$ samples. Briefly, $NO_3^-$ from filter extractions were converted into gaseous $N_2O$ by the bacterial denitrifier method (Sigman et al., 2001; Casciotti et al., 2002; Kaiser et al., 2007) in which $\approx 100$ nmol of $NO_3^-$ ions were injected into a 2 mL of a bacteria medium (strain of *Pseudomonas aureofaciens*) under anaerobic conditions. $NO_2$ denuder extractions were treated separately with the azide method (McIlvin and Altabet, 2005; Albertin et al., 2021) in which 2 mL of a sodium azide 2M / acetic acid 100 % buffer were injected into $\approx 100$ nmol of $NO_2^-$ allowing quantitative conversion into $N_2O$. For both filter and denuder extractions, ions were converted into $N_2O$ which was further thermally decomposed into $O_2$ and $N_2$ in a gold tube heated at 850 °C. Then, $O_2$ and $N_2$ molecules were separated on a chromatography column and sent separately into the IRMS for the dual analysis of O and N isotopes (see Morin et al., 2009 for more details on the analytical line). The isotopic composition of $NO_3^-$ samples was analysed in triplicate (the mean value of replicate measurements and the associated repeatability are reported in Table S2 in the Supplement). The limited amount of $NO_2$ samples did not allow for replicate measurements. From UV-vis analysis, all $NO_2$ samples presented a negligible blank (< 4 %; mean of 1.7 nmol ml$^{-1}$) except for the sample collected between 13:30 and 16:30 LT during SP 2 which shown a blank around $(14.0 \pm 1.4)$ %. Therefore, the measured $\Delta^{17}O$ of this sample was corrected for blank effect assuming that the contaminated $NO_2^-$ possessed a $\Delta^{17}O = 0$ ‰. No correction from this blank effect was applied on the $\delta^{15}N$ measurements of $NO_2$ because the $\delta^{15}N$ fingerprint of the contamination could not be characterised. This uncertainty is propagated in the calculations of Section 3 and considered in the discussions. Possible isotopic changes resulting from the

conversion and analysis process of $NO_3^-$ and $NO_2$ samples were evaluated using international $NO_3^-$ and $NO_2^-$ isotopic reference materials, respectively (Table S3 in the Supplement). Accuracy of the analytical method was estimated as the standard deviation ($\sigma$) of the residuals between measurements of the reference materials and their expected values. In our study, average measurement uncertainties on $\delta^{15}N$, $\delta^{17}O$, $\delta^{18}O$, and $\Delta^{17}O$ were estimated to be $\pm 0.3$ ‰, $\pm 0.9$ ‰, $\pm 1.3$ ‰, and $\pm 0.4$ ‰, respectively, for $NO_3^-$ samples and $\pm 0.3$ ‰, $\pm 0.4$ ‰, $\pm 0.9$ ‰, and $\pm 0.3$ ‰, respectively, for $NO_2$ samples. Detailed information about the calibration procedure can be found in Morin et al. (2009) and in Albertin et al. (2021) for $NO_3^-$ and $NO_2$ samples, respectively.

## 2.3  Ancillary data

During atmospheric samplings, surface $NO_x$ mixing ratios were measured at the study site using an incoherent broadband cavity-enhanced absorption spectrometer for $NO_2$ (IBBCEAS; Barbero et al., 2020) and an optical-feedback cavity-enhanced absorption spectrometer for NO (OFCEAS; Richard et al., 2018). PM concentrations ($PM_{10}$ and $PM_{2.5}$) were monitored by an optical particle counter (GRIMM®, EDM 164). $O_3$ mixing ratio was monitored at the local air quality monitoring site located a kilometre north of the sampling site (Environnement SA®, O3 42M; https://www.atmo-auvergnerhonealpes.fr/, last access: 5 November 2021). Surface temperature ($T_{surface}$) and relative humidity (RH) were measured by a portable logger (Tinytag, TGP-4500, Gemini Data Loggers) located at the air quality monitoring site. Vertical temperatures were measured from 11 similar loggers fixed along the Plan-Praz cable car (45°55′39″ N, 6°51′55″ E) from 1098 to 2021 m above sea level (data obtained from personal communications with C. Coulaud, IGE). The $NO_2$ photolysis rate ($J_{NO_2}$) was calculated for the two sampling periods using a photochemical boxmodel (CiTTyCAT version 2.02; Galeazzo et al., 2018; Pugh et al., 2012) using the Fast-J photolysis scheme of Wild et al. (2000) and a surface albedo fixed to 0.65, a value representative of a snow-covered surface (average value between fresh and old snow; more details can be found in Text S1 in the Supplement).

## 2.4  Interpretation framework for isotopic signals

In this section, we briefly state the key concepts and equations necessary to interpret isotopic signals measured in $NO_2$ and $NO_3^-$. A more detailed description and complete equation derivations can be found in cited references.

### 2.4.1  $\Delta^{17}O$ mass balance equations

Because $NO_2$ and $NO_3^-$ loss processes do not fractionate in terms of the oxygen mass-independent anomaly and considering that each source reaction induces a transfer of $\Delta^{17}O$ to $NO_2$ and $NO_3^-$, one consider the mass conservation of $\Delta^{17}O$ during fractionation processes in the $N_r$ cycle. Hence, one can implement $\Delta^{17}O$ in the general mass balance equation of $NO_2$ and $NO_3^-$. An overall expression of the time derivative of $\Delta^{17}O$ in the species X ($\Delta^{17}O(X)$; with X = $NO_2$ or $NO_3^-$) is derived as a function of its deviation from $\Delta^{17}O$ transferred through each production channel $i$ ($P_i$) ($\Delta^{17}O_i(X)$), weighted according to the relative contributions of the production channels (Vicars et al., 2013):

$$\frac{d}{dt}\left(\varDelta^{17}O(X)\right) = \frac{1}{\tau(X)} \times \sum_i \frac{P_i}{\sum_i P_i} \times \left(\varDelta^{17}O_i(X) - \varDelta^{17}O(X)\right) \tag{2}$$

where $P_i$ expresses reaction rate constant times the atmospheric concentrations of reacting species, and $\tau$ is the atmospheric lifetime of the species X at steady state ($\tau = [X]/\sum_i P_i$ with [X] being the atmospheric mixing ratio of the species X).

During the day, the rapid photochemical cycling of $NO_x$ (Reactions R1–R4) leads to an isotopic equilibrium between NO and $NO_2$, i.e., $\varDelta^{17}O(NO) \approx \varDelta^{17}O(NO_2)$ (Michalski et al., 2014). Therefore, using the steady state approximation, and considering $NO + O_3$ (Reaction R3) and $NO + RO_2$ (Reaction R4) as the main sources of $NO_2$ at our site, the overall daytime $\varDelta^{17}O$ in $NO_2$ can be expressed by:

$$\varDelta^{17}O_{day}(NO_2) \approx T_{NO+O_3} \times \varDelta^{17}O_{NO+O_3}(NO_2) \tag{3}$$

where $\varDelta^{17}O_{NO+O_3}(NO_2)$ is the $O_3$ isotopic anomaly transferred to NO through Reaction (R3) (Savarino et al., 2008). $T_{NO+O_3}$, sometimes named $A$ in the literature, represents the proportion of O atoms originating from $O_3$ in $NO_2$, and hence the relative importance of Reaction (R3) in the conversion of NO into $NO_2$ (Michalski et al., 2003; Morin et al., 2007b; Albertin et al., 2021):

$$T_{NO+O_3} = \frac{k_{NO+O_3}[O_3]}{k_{NO+O_3}[O_3] + k_{NO+RO_2}[RO_2]} \tag{4}$$

where $k_{NO+O_3}$ and $k_{NO+RO_2}$ are the kinetic constants of Reactions (R3) and (R4), respectively. The kinetic constants used in this study are listed in Table A1 in the Appendix. At night, considering that (1) $\varDelta^{17}O(NO) \approx 0$ ‰ ($NO_x$ emission without $NO_2$ recycling), (2) no $\varDelta^{17}O$ equilibrium between NO and $NO_2$ (no photochemical cycling), and (3) $O_3$ is the main oxidant of NO (no nighttime production of $RO_2$), $\varDelta^{17}O(NO_2)$ is determined by the $\varDelta^{17}O$ transfer via Reaction (R3) and by the nighttime residuals of $NO_2$ formed during the previous daytime hours (Albertin et al., 2021) following:

$$\varDelta^{17}O_{night}(NO_2) \approx x \times \varDelta^{17}O_{day}(NO_2) + \frac{(1-x)}{2} \times \left(\varDelta^{17}O_{NO+O_3}(NO_2) + \varDelta^{17}O(NO)\right) \tag{5}$$

where $x$ is the fraction of $NO_2$ formed during the day to the total $NO_2$ measured at night.

At our sampling site, we hypothesise Reaction (R5) (OH pathway) and Reactions (R6)–(R8) ($N_2O_5$ pathway) as the main daytime and nighttime $NO_3^-$ production channels, respectively. At steady state, from Eq. (2), we derive general

expressions for $\Delta^{17}O$ in $NO_3^-$ during the day and night, associated with the OH and $N_2O_5$ pathways, respectively (Alexander et al., 2020):

$$\Delta^{17}O_{day}(NO_3^-) \approx \frac{2}{3} \times \Delta^{17}O_{day}(NO_2) \tag{6}$$

$$\Delta^{17}O_{night}(NO_3^-) \approx \frac{2}{3} \times \Delta^{17}O_{night}(NO_2) + \frac{1}{6} \times \Delta^{17}O_{NO_2+O_3}(NO_3) \tag{7}$$

where $\Delta^{17}O_{NO_2+O_3}(NO_3)$ is the $^{17}O$-excess transfer from $O_3$ to $NO_3$ during Reaction (R6) (Berhanu et al., 2012). Without wet scavenging, dry deposition is the main sink of $NO_3^-$ (Park et al., 2004). Assuming a mean $NO_3^-$ deposition velocity of 0.5 cm s$^{-1}$ (mean value of the dry deposition velocities of $HNO_3$ and $p$-$NO_3^-$; Zhang et al., 2009), and considering the maximum daytime and minimum nighttime boundary layer heights of 500 and 100 m above ground level, respectively (estimations based on measured vertical temperature profiles; Fig. S2 in the Supplement), the estimated residence time of $NO_3^-$ against dry deposition can reach up to 28 hours during the day, and 6 hours at night (Table B1). Therefore, on sub-daily time scales, the $^{17}O$-excess in $NO_3^-$ during the day is more likely to reflect a combination of daytime and nighttime production processes than during the night. Note that, our estimated residence times for $NO_3^-$ against dry deposition are upper limits as they represent the time required to reduce by a factor e the concentration of $NO_3^-$ present at the top of the boundary layer; $NO_3^-$ close to the surface would have a much shorter residence time.

### 2.4.2    Nitrogen isotopic fractionation effects

Each source of $NO_x$ generates a $\delta^{15}N$ fingerprint which depends on the type and conditions (temperature and pressure) of combustion, and on the type of fuel (e.g., coal, oil, gas) (Heaton, 1990; Felix et al., 2012; Fibiger and Hastings, 2016; Walters et al., 2015a, b; Yu and Elliott, 2017; Miller et al., 2018). The mean $\delta^{15}N$ of $NO_x$ ($\delta^{15}N(NO_x)$) emitted in the atmosphere results from the sum of each $NO_x$ emission $\delta^{15}N$ fingerprint weighted by their relative contribution to the total $NO_x$ emissions. Once in the atmosphere, $NO_x$ is subjected to oxidation processes and isotopic exchanges that alter the initial $\delta^{15}N(NO_x)$. As a result, $\delta^{15}N$ in $NO_2$ and in $NO_3^-$ is a complex function of both the $\delta^{15}N$ signature of $NO_x$ emissions and N isotopic effects. These latter can be categorised into three groups: (1) the equilibrium isotope effect (EIE), (2) the kinetic isotope effect (KIE), and (3) the photochemical isotope fractionation effect (PHIFE) (Miller and Yung, 2000; Young et al., 2002). The magnitude of these isotopic effects is quantified as $^{15}N$ enrichment factor ($\varepsilon$), which is defined as ($\alpha - 1$), where $\alpha$ represents the N isotopic fractionation factor.

A general expression for $\delta^{15}N(NO_2)$ can be derived as a function of a factor $F_N$ which represents the overall N isotopic fractionation effects between $NO_x$ emissions and $NO_2$ (expressed in ‰), the fraction of $NO_2$ with respect to $NO_x$ ($f_{NO_2} = [NO_2]/[NO_x]$), and of $\delta^{15}N(NO_x)$ (Albertin et al., 2021; Li et al., 2020):

$$\delta^{15}N(NO_2) = F_N \times (1 - f_{NO_2}) + \delta^{15}N(NO_x) \tag{8}$$

Therefore, the $^{15}N$ isotopic shift between $\delta^{15}N(NO_2)$ and $\delta^{15}N(NO_x)$ is given by:

$$\delta^{15}N(NO_2) - \delta^{15}N(NO_x) = \Delta^{15}(NO_2 - NO_x) = F_N \times (1 - f_{NO_2}) \tag{9}$$

Physico-chemical processes between NO and $NO_2$ can preferentially promote or deplete $^{15}N$ in $NO_2$ with respect to emissions of $NO_x$ (i.e., $\delta^{15}N(NO_2) \neq \delta^{15}N(NO_x)$). The importance of this fractionation shift is modulated by the factor $(1 - f_{NO_2})$. When NO is almost entirely converted into $NO_2$ ($f_{NO_2} \approx 1$), N fractionation effects can be neglected (i.e., $\delta^{15}N(NO_2) \approx \delta^{15}N(NO_x)$).

From samples collected at Jülich, Germany, Freyer et al. (1993), observed for the first time the linear relation described by Eq. (8), and set the theoretical framework to interpret $\delta^{15}N$ variabilities in atmospheric $NO_2$. They showed that the observed seasonal variation of $\delta^{15}N(NO_2)$ was driven by N fractionation effects (represented in the $F_N$ factor) caused by photochemistry and isotopic equilibrium. Based on this work, and that of Li et al. (2020), Albertin et al. (2021) derived an expression of $F_N$ during the day assuming that the $NO$-$NO_2$ system is in isotopic equilibrium (steady-state):

$$(F_N)_{day} \approx \frac{\alpha_{LCIE}{}^* A^*{}_{day} + (\alpha_{EIE(NO_2/NO)} - 1)}{A^*{}_{day} + 1} \tag{10}$$

with $\alpha_{LCIE}{}^* = \alpha_{KIE(NO+O_3)} - \alpha_{PHIFE}$

and $A^*{}_{day} = \frac{J_{NO_2}}{k_{NO+NO_2}[NO]}$

where $\alpha_{LCIE}{}^*$ is the fractionation factor of combined KIE and PHIFE (LCIE is for Leighton Cycle Isotope Effect), and $\alpha_{EIE(NO_2/NO)}$ is the EIE fractionation factor between NO and $NO_2$. $\alpha_{EIE(NO_2/NO)}$ and $\alpha_{KIE(NO+O_3)}$ are temperature dependent and can be calculated following the theoretical approach of Walters and Michalski (2015) (Table D1). From calculations based on the zero point energy of $^{15}NO_2$ and the absorption cross section of $^{14}NO_2$, $\alpha_{PHIFE}$ is estimated to vary between 1.0020 and 1.0042 for a range of solar zenith angles between 90 ° and 0 ° (Fang et al. 2021). In this study we use a mean value of $\alpha_{PHIFE}$ at 1.0031. $A^*{}_{day}$ is defined as the ratio of the $NO_2$ lifetime with respect to isotopic exchanges over the daytime $NO_2$ chemistry lifetime. $J_{NO_2}$ is the $NO_2$ photolysis rate, $k_{NO+O_3}$ is the rate constant of Reaction (R3), and $k_{NO+NO_2}$

is the rate constant of the isotopic exchange $^{15}NO_2 + {}^{14}NO \rightarrow {}^{14}NO_2 + {}^{15}NO$. During the day, $\Delta^{15}(NO_2 - NO_x)$ varies
according to the environmental conditions. In low-$NO_x$ conditions (e.g., remote and polar regions) $\Delta^{15}(NO_2 - NO_x)$ is
predicted to be controlled by LCIE factors ($A^*_{day} \gg 1$), whereas an EIE-dominated regime ($A^*_{day} \ll 1$) is expected in
polluted environments (high-$NO_x$ conditions). At night, $J_{NO_2}$ and $\alpha_{PHIFE}$ are null and $A^*_{night}$ is defined as the ratio of NO
lifetime with respect to isotopic exchange with $NO_2$ to NO chemical lifetime at night $\left( A^*_{night} = \frac{k_{NO+O_3}[O_3]}{k_{NO+NO_2}[NO_2]} \right)$. In this study,
we consider only one particular case with $A^*_{night} \ll 1$, which means that isotopic exchanges are much faster than NO
oxidation by $O_3$. In this scenario, KIE effects are negligible compared to EIE effects and $(F_N)_{night}$ can be expressed as:

$$(F_N)_{night} \approx \frac{(\alpha_{EIE(NO_2/NO)} - 1)}{\alpha_{EIE(NO_2/NO)}} \qquad (11)$$

The complete derivation of Eqs. (10) and (11) is given in Albertin et al. (2021).

KIE and EIE are also expected during the conversion of $NO_2$ to $NO_3^-$. The $^{15}N$ partitioning associated to isotopic
equilibrium between $N_2O_5$ and $NO_2$ (Reaction R7) can be theoretically computed as a function of temperature (Walters and
Michalski, 2015; Table D1). At 298 K, if N isotopic equilibrium is reached, $N_2O_5$ is predicted to have $\delta^{15}N$ values 27.6 ‰
higher than $NO_2$. Considering that the $NO_2/NO_3^-$ isotopic fractionation through the $N_2O_5$ pathway is solely controlled by
EIE, $NO_3^-$ is therefore expected to be enriched in $^{15}N$ relative to $NO_2$. However, to date, no experimental study has reported
on $^{15}N$ partitioning between atmospheric $NO_2$ and $NO_3^-$, and the fractionation factors are still being debated (Freyer 1991,
Fang et al. 2021). The dominant $NO_x$ to $NO_3^-$ conversion processes considered in this study, along with corresponding $\Delta^{17}O$
transfer factors and the known $^{15}N$ enrichment factors at 298 K (determined from both experimental and computational
studies) are illustrated in Figure C1.
**3    Results and Discussion**
**3.1    Temporal variations of general atmospheric observations**
Surface temperatures during SP 1 and SP 2 show similar values, with a marked diurnal cycle (from −2 °C to 16 °C; Figure
1). A positive temperature gradient with altitude is observed from late evening to morning. Surface temperature rises around
midday and reaches a maximum at around 15:00 LT, resulting in a negative temperature gradient with altitude. In deep
Alpine valleys, the diurnal variability of surface air temperature is strongly influenced by the temporal evolution of the
boundary layer structure, particularly in winter with the formation of a surface layer inversion (Whiteman, 1982). As
previously observed in Chamonix (Chazette et al., 2005), the nocturnal surface layer inversion regularly thickens during the
night of the sampling periods. After sunrise, air masses warm up until the nocturnal inversion layer breaks down in the late

morning. Observed RH behaviour relatively correlates with the increase in temperature during the day, showing a rapid decrease between mid-morning and early afternoon (from 96 % to 23 % and from 96 % to 30 % for SP 1 and SP 2, respectively; Figure 1).

During SP 1 and SP 2, the mixing ratios of NO, $NO_2$ and $O_3$ exhibit diurnal patterns (Figure 1) typical in Chamonix in late February (Figure S3 in the Supplement), and more generally in urban areas (Mayer, 1999). The highest NO level is observed in the morning, peaking around 10:30 LT (82 nmol $mol^{-1}$ and 152 nmol $mol^{-1}$ for SP 1 and SP 2, respectively). The concurrent increase in $NO_2$ and decrease in $O_3$ (down to 1 nmol $mol^{-1}$) can be attributed to Reaction (R3) (NO + $O_3$). $NO_x$ decreases in the late morning, likely due to a combination of lower emissions, $NO_2$ oxidation, and dilution effect. Meanwhile, $O_3$ gradually recovers to ca. 30 nmol $mol^{-1}$, a typical winter background air level in Europe (Gaudel et al., 2018). Due to local emissions, $NO_x$ increases again from 16:00 LT, resulting in $O_3$ titration, and subsequently to an increase in $NO_2$ (up to 40 nmol $mol^{-1}$). After 18:30 LT, NO remains low until the morning, and $NO_2$ decreases slowly until midnight, stalls around 10 nmol $mol^{-1}$, and then rises again at 5:30 LT. After the late afternoon titration, $O_3$ gently recovers and stay relatively low throughout the night likely due to a titration effect from nocturnal NO emissions which are confined in the surface layer inversion.

Both SP 1 and SP 2 show diurnal variations in PM mass concentrations (Figure 1), with morning and evening peaks related to local emissions from traffic and home heating (Aymoz et al., 2007). $PM_{10}$ concentrations display an additional increase at midday following the breakdown of the temperature inversion. At 12:30 LT, while $PM_{10}$ concentrations increase moderately during SP 1 to reached 38.6 $\mu$g $m^{-3}$, a sharp increase to 119 $\mu$g $m^{-3}$ is observed during SP 2. Then $PM_{10}$ concentrations decrease during the afternoon of SP 1, but remain high during the afternoon of SP 2 until the surface inversion layer forms. On average, $PM_{10}$ concentration is three times higher during SP 2 ((59.4 ± 37.6) $\mu$g $m^{-3}$) than during SP 1 ((20.6 ± 10.2) $\mu$g $m^{-3}$). The considerable increase in $PM_{10}$ concentrations between SP 1 and SP 2 is likely to be explained by a Saharan dust episode that started on February 23 (Fig. S3, S4, and S5 in the Supplement). Saharan dust deposition is a well-known phenomenon in the Alps, which is characterised by a sudden increase of coarse particles, mainly composed of alumino-silicates as well as calcium and potassium (Angelisi and Gaudichet, 1991; Delmas, 1994; Di Mauro et al., 2019; Goudie and Middleton, 2001; Greilinger et al., 2018; Schwikowski et al., 1995; Sodemann et al., 2006).

The $NO_3^-$ mass concentration varies from 0.3 $\mu$g $m^{-3}$ to 3.4 $\mu$g $m^{-3}$, with an average of (0.9 ± 0.6) $\mu$g $m^{-3}$ for SP 1 and of (1.2 ± 0.9) $\mu$g $m^{-3}$ for SP 2 (Figure 1). During both sampling periods, $NO_3^-$ concentration is within the range of previous observations made in Chamonix in winter (Allard, 2018). $NO_3^-$ shows a distinctive peak at 3.4 $\mu$g $m^{-3}$ during SP 2 between 10:30 and 13:30 LT, correlated with the $PM_{10}$ surge. During transport, dust can undergo heterogeneous uptake and conversion of gases on its surface, leading to the inclusion of secondary species such as $NO_3^-$, sulfate, and ammonium

(Usher et al., 2003). $NO_3^-$ on dust results mainly from $HNO_3$ uptake and heterogeneous reactions of $N_2O_5$ (see Usher et al.,
2003 for a review and references therein). Mineral dust is believed to significantly contribute to $NO_3^-$ formation and size
distribution, particularly in regions close to dust emission sources (Karydis et al., 2016). However, the origin of $NO_3^-$ during
SP 2 at our site remains unclear and could be attributed to the advection of both nitrated-dust particles formed through
heterogeneous processes during transport and anthropogenic fine particles (Aymoz et al. 2004).

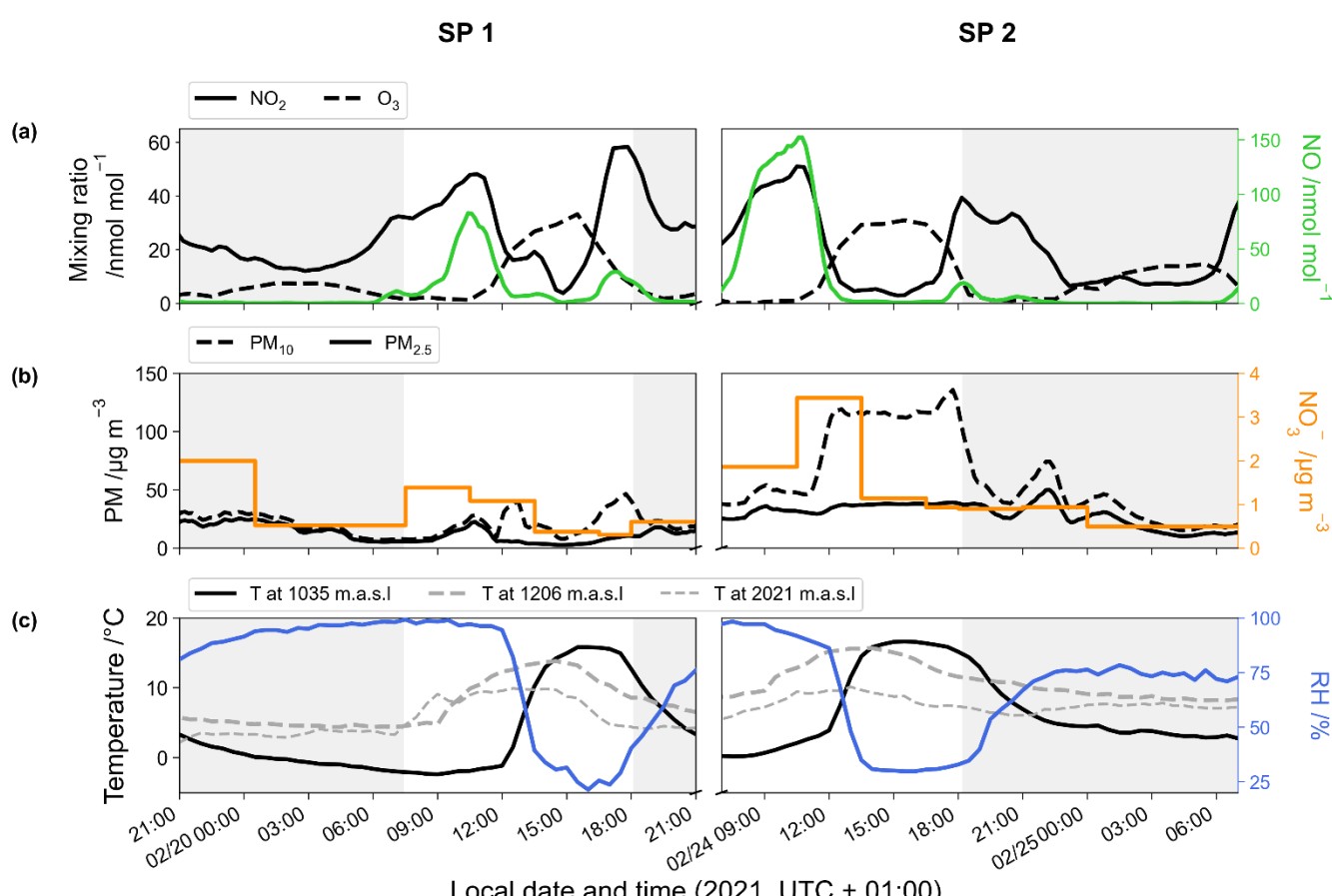

**Figure 1.** Temporal evolution of the 1-hour rolling mean of **(a)** $NO_2$ (black line), $O_3$ (dashed line), and NO (green line) mixing ratios, **(b)**
PM (dashed line for $PM_{10}$ and solid line for $PM_{2.5}$) and $NO_3^-$ (orange horizontal line) mass concentrations, and **(c)** temperature at the
surface (black line), at 1206 m (dashed grey line), and at 2021 m (light dashed grey line) and surface relative humidity (blue line). Data
were collected during the two sampling periods (SP 1 and SP 2) in Chamonix. Grey backdrop shaded areas represent the nighttime (sunset
to sunrise).

Figure 2 shows the temporal evolution of measured $\Delta^{17}O$ and $\delta^{15}N$ of $NO_2$ and $NO_3^-$ in Chamonix during the two
sampling periods (SP 1 and SP 2). All isotopic data used in this study are reported in Table S1 and Table S2 in the
Supplement. In the following analysis, first we describe $\Delta^{17}O(NO_2)$ measurements and gives interpretation in light of $NO_x$
chemistry cycling (Section 3.2). Then, an analysis of $\Delta^{17}O(NO_3^-)$ measurements is proposed aggregating daytime and
nighttime periods and comparing them with $\Delta^{17}O(NO_3^-)$ estimates derived from $\Delta^{17}O(NO_2)$ measurements and $\Delta^{17}O$ mass
balance for major chemical processes (Section 3.3). In light of these results and atmospheric conditions during SP 1 and SP
2, sub-daily $\Delta^{17}O(NO_3^-)$ dynamics are investigated. In Section 3.4, N fractionation effects in the $NO_x$ cycle are quantified,
and the dominant $NO_x$ emission source is identified. The dynamics of $\delta^{15}N(NO_3^-)$ is also described and its use to trace $NO_x$
emission sources and oxidation processes is discussed.

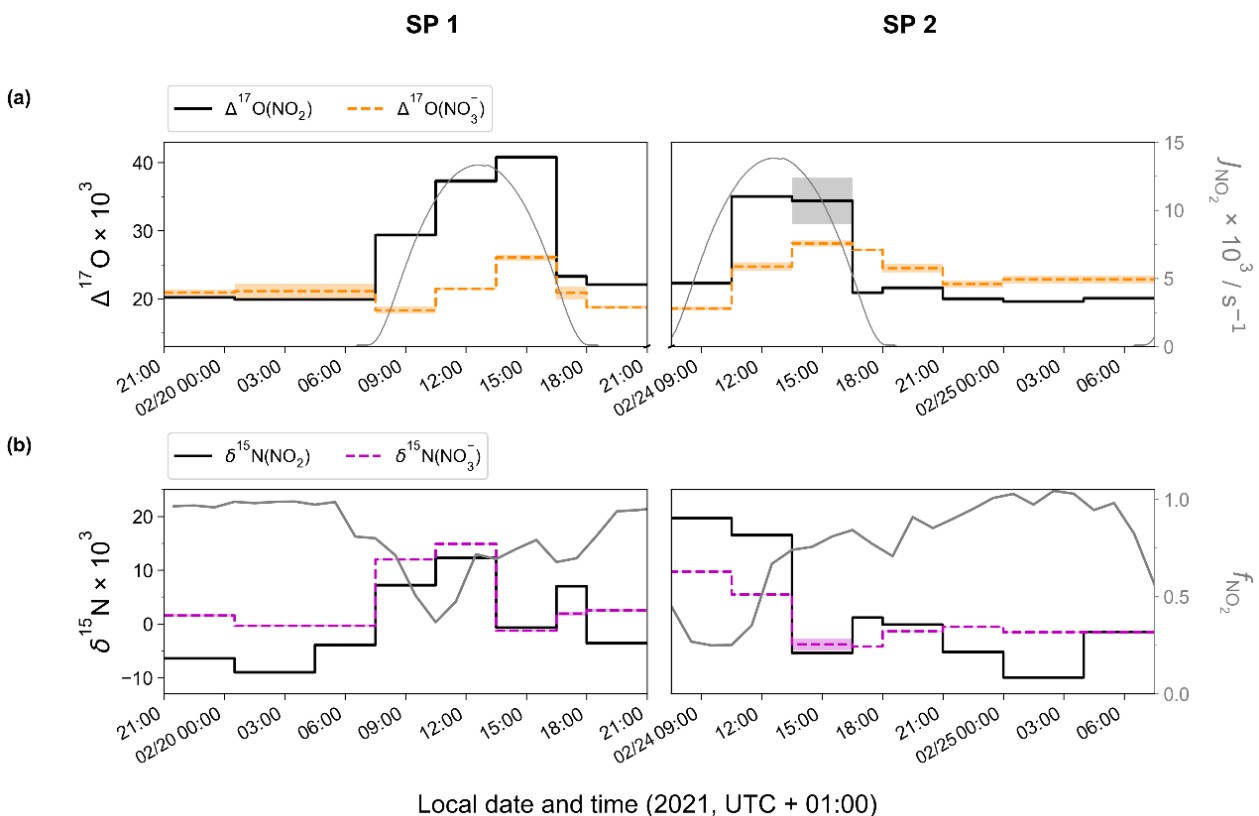


**Figure 2.** Temporal evolution of observed **(a)** $\Delta^{17}O$ and **(b)** $\delta^{15}N$ of atmospheric $NO_2$ (top and bottom solid black horizontal line) and
$NO_3^-$ (top orange and bottom magenta horizontal dashed line) in Chamonix (length of horizontal line = sampling period, shaded area =
overall analytical error). The $NO_2$ photolysis rate ($J_{NO_2}$, top grey line) is from CiTTyCAT boxmodel output. $f_{NO_2}$ ($f_{NO_2}$ = [$NO_2$] / ([$NO_2$]
+ [NO]); bottom grey line) is calculated from hourly mean mixing ratio of NO and $NO_2$.

### 3.2  $\Delta^{17}O$ of $NO_2$ and $NO_x$ diurnal cycling

Over the course of SP 1 and SP 2, $\Delta^{17}O(NO_2)$ shows a large diurnal variability (from 19.6 ‰ to 40.8 ‰) with a weighted
mean ± one standard deviation of (25.2 ± 7.1) ‰. $\Delta^{17}O(NO_2)$ values during the day (7:30−18:00 LT, (28.5 ± 7.3) ‰) are
significantly higher (*p*-value = 0.002, n = 16) than during the night (18:00−7:30 LT, (20.8 ± 1.0) ‰). By day, $\Delta^{17}O(NO_2)$
follows a similar increasing trend during SP 1 and SP 2, reaching a respective maximum of 40.8 ‰ between 13:30−16:30 LT
and 35.0 ‰ between 10h30−13h30 LT. For both sampling periods, after sunset, $\Delta^{17}O(NO_2)$ stabilises between 21:00 and
7:30 LT at ca. 20 ‰. Using the same sampling methodology in a mid-latitude urban area in spring, Albertin et al. (2021)
reported very similar $\Delta^{17}O(NO_2)$ values over the course of one day (20.5−39.2 ‰), following a comparable diurnal pattern.
As presented above (Section 2.4.1), according to the ISS (isotopic steady state) framework, the variability of $\Delta^{17}O(NO_2)$
reflects changes in the relative contributions of Reaction (R3) (NO + $O_3$) and Reaction (R4) (NO + $RO_2$) to the overall
production of $NO_2$. At our site, $\Delta^{17}O(NO_2)$ drops rapidly during the 16:30−18:00 LT interval to 23.3 ‰ and 20.9 ‰ during
SP 1 and SP 2, respectively. Since, the isotope recycling rate in the $NO_x$-$O_3$ system is driven at first order by $J_{NO_2}$ (Michalski
et al., 2014), due to low solar radiation between 16:30−18:00 LT at our site, such a rapid drop of $\Delta^{17}O(NO_2)$ suggests that
sampled $NO_2$ might not be at ISS anymore when the $NO_2$ photolysis is very slow, notably at the end of the day in winter.
Therefore, we only consider that ISS holds between 7:30 and 16:30 LT in order to avoid the questionable end-of-the day
measurements in our ISS-based analysis. At night (i.e., no ISS), the observed drop of $\Delta^{17}O(NO_2)$ in the early evening reflects
the rapid replacement of $NO_2$ formed during the day by $NO_2$ produced during the night via the conversion of freshly emitted
NO, in line with Eq. (5). Then, high $NO_2$ throughout the night, along with relatively low $O_3$, supports observations of low
$\Delta^{17}O(NO_2)$ at night (i.e., ≈20 ‰).

439        Using Eq. (3), we derive from $\Delta^{17}O(NO_2)$ observations the relative contribution of Reaction (R3) (NO + $O_3$) to

Reaction (R4) (NO + $RO_2$) in the formation of $NO_2$ ($T_{NO+O_3}$):

$$T_{NO+O_3} = \frac{\Delta^{17}O_{day}(NO_2)}{\Delta^{17}O_{NO+O_3}(NO_2)} \tag{12}$$

Between 7:30 and 16:30 LT, $T_{NO+O_3}$ varies from 0.55 to 1.00 (Table 1), with a mean of 0.88 and 0.75 for SP 1 and SP 2,
respectively. The NO + $O_3$ pathway is dominant between 13:30 and 16:30 LT, corresponding to the time when $O_3$ is highest
(Figure 1). In contrast, the maximum contribution for the NO + $RO_2$ pathway is observed between 7:30 and 10:30 LT, when
NO levels are high and rising continuously. Interestingly, previous studies reported a high sensitivity of $RO_2$ to changes in
$NO_x$, particularly at high $NO_x$ levels (Ren et al., 2006; Stone et al., 2012). Sources of $RO_2$ in wintertime are mainly driven by
the production of OH radicals from HONO photolysis, alkene ozonolysis, and formaldehyde photolysis (Tan et al., 2018).
During winter, HONO plays a crucial role in $NO_x$/$O_3$/$RO_2$ chemistry, particularly in the morning, as its photolysis can
potentially accelerate daytime oxidation processes, leading to increased $RO_2$ production (Alicke et al., 2003; Aumont et al.,
2003). Direct emissions from vehicle exhaust could be significant a source of VOCs and HONO at our site (Brulfert et al.,
2005; Gu et al., 2019; Kirchstetter et al., 1996; Kurtenbach et al., 2001; Liu et al., 2023). Heterogeneous processes on ground
surfaces and aerosols can also contribute to HONO formation (Aumont et al., 2003). In addition, snowpack releases may also

be a potential source of HONO (Grannas et al., 2007), as detected in Paris after a snow event, which could significantly impact the urban OH budget (Michoud et al. 2015).

Following the approach of Albertin et al. (2021), combining Eqs. (3) and (4) allows to derive $RO_2$ mixing ratio from observed $\Delta^{17}O(NO_2)$ and $O_3$ mixing ratio following:

$$[RO_2] = \frac{k_{NO+O_3}[O_3]}{k_{NO+RO_2}} \left( \frac{\Delta^{17}O_{NO+O_3}(NO_2)}{\Delta^{17}O_{day}(NO_2)} - 1 \right) \tag{13}$$

Between 7:30 and 16:30 LT, we estimate an average $RO_2$ mixing ratio at our site of $(0.88 \pm 0.88)$ pmol mol$^{-1}$ and $(4.92 \pm 5.16)$ pmol mol$^{-1}$ during SP 1 and SP 2, respectively (Table 1). Studies conducted in urban winter environments reported $RO_2$ measurements at a few pmol mol$^{-1}$ (Ren et al., 2006; Emmerson et al., 2005; Tan et al., 2018; Kanaya et al., 2007), in good agreement with our estimations. Similarly, $RO_2$ mixing ratios derived by Albertin et al. (2021) from observed $\Delta^{17}O(NO_2)$ in spring (mean of $(13.8 \pm 11.2)$ pmol mol$^{-1}$) were also found to be in line with studies conducted in the same season. The fact that our isotopic inference of $RO_2$ mixing ratios carried out in two different seasons (winter and spring) are both comparable to direct in situ $RO_2$ measurements confirm the sensitivity of our method in probing the $NO_x/O_3/RO_2$ chemical dynamics. We think that our method may be very valuable in deciphering oxidation processes of $N_r$ species, down to sub-daily temporal scales. Nonetheless, we recognise that without concurrent in situ measurements of $RO_2$ and $\Delta^{17}O(NO_2)$, it is not possible to validate unambiguously the quantitative estimation of $RO_2$ levels with our method.

| Sampling interval (start - end) | $T_{NO+O_3}$ | $RO_2$ /pmol mol$^{-1}$ |
|---|---|---|
| SP 1 | | |
| 20/02 07:30 - 20/02 10:30 | $0.72 \pm 0.01$ | $0.86 \pm 0.75$ |
| 20/02 10:30 - 20/02 13:30 | $0.91 \pm 0.01$ | $1.77 \pm 0.36$ |
| 20/02 13:30 - 20/02 16:30 | $1.00 \pm 0.01$ | $0.00 \pm 0.91$ |
| *Mean* | *0.88* | *0.88* |
| *Std dev.* | *0.14* | *0.88* |
| SP 2 | | |
| 24/02 07:30 - 24/02 10:30 | $0.55 \pm 0.01$ | $0.58 \pm 1.67$ |
| 24/02 10:30 - 24/02 13:30 | $0.86 \pm 0.01$ | $3.56 \pm 0.50$ |
| 24/02 13:30 - 24/02 16:30 | $0.84 \pm 0.08$ | $10.63 \pm 6.75$ |
| *Mean* | *0.75* | *4.92* |
| *Std dev.* | *0.18* | *5.16* |

**Table 1.** $T_{NO+O_3}$ and $RO_2$ mixing ratio (mean value ± overall uncertainty) derived from the isotopic measurements.

470

At this point, it is important to recall that the choice of the $\Delta^{17}O_{NO+O_3}(NO_2)$ in Eq. (12) is of a particular importance for quantifying $T_{NO+O_3}$ (as for $RO_2$). In the literature, $\Delta^{17}O_{NO+O_3}(NO_2)$ varies between 35 ‰ and 41 ‰ (Michalski et al., 2003; Savarino et al., 2016; Vicars et al., 2012; Zhang et al., 2022b; Li et al., 2022a). This relatively wide range of values is partly a result of some confusion in defining $\Delta^{17}O_{NO+O_3}(NO_2)$, the $\Delta^{17}O$ transfer from $O_3$ to NO. Indeed, the term $\Delta^{17}O(O_3^*)$ is sometimes erroneously used as the transfer function of $\Delta^{17}O$ from $O_3$ to $NO_2$ during Reaction (R3). $\Delta^{17}O(O_3^*)$ is actually defined as $\Delta^{17}O$ of $O_3$ terminal atoms and is also named $\Delta^{17}O(O_3)_{term}$ in the literature. As the $\Delta^{17}O$ in $O_3$ is borne by its terminal atoms, $\Delta^{17}O(O_3^*) = 1.5 \times \Delta^{17}O(O_3)_{bulk}$. However, $\Delta^{17}O_{NO+O_3}(NO_2)$ can be equal to $1.5 \times \Delta^{17}O(O_3)_{bulk}$ if only terminal atoms of $O_3$ reacts with NO. But laboratory experiments by Savarino et al. (2008) did show that $O_3$ does react with NO not solely with its terminal atoms but also, to a small but significant extent, with its central atom (probability of $(8 \pm 5)$ % for the abstraction of central atoms during the reaction NO + $O_3$). Consequently, $\Delta^{17}O_{NO+O_3}(NO_2)$ is slightly lower than $1.5 \times \Delta^{17}O(O_3)_{bulk}$ and the $\Delta^{17}O_{NO+O_3}(NO_2)$ expression determined by Savarino et al. (2008) should be used:

$\Delta^{17}O_{NO+O_3}(NO_2) = 1.18 \pm 0.07 \times \Delta^{17}O(O_3)_{bulk} + (6.6 \pm 1.5)$ ‰.

483

Assuming that their maximum measured daytime $\Delta^{17}O(NO_2)$ reflects the conversion of NO to $NO_2$ only through Reaction (R3) (i.e., $T_{NO+O_3} = 1$), Albertin et al. (2021) derived a $\Delta^{17}O_{NO+O_3}(NO_2)$ value of 39.2 ‰ from Eq. (3). Given the respective analytical uncertainties (around ±1 ‰), their value is in very good agreement with the maximum daytime value of 40.8 ‰ we observed in Chamonix. Similarly to Albertin et al. (2021), assuming that the highest daytime $\Delta^{17}O(NO_2)$ value at our site corresponds to $T_{NO+O_3} \approx 1$ leads to $\Delta^{17}O_{NO+O_3}(NO_2) = 40.8$ ‰. Using the experimental $\Delta^{17}O_{NO+O_3}(NO_2)$ transfer function determined by Savarino et al. (2008), we estimate a bulk $^{17}O$-excess of $O_3$ ($\Delta^{17}O(O_3)_{bulk}$) at $(29.0 \pm 2.2)$ ‰. This value is consistent with the range of direct $\Delta^{17}O(O_3)_{bulk}$ measurements at mid-latitudes (mean of $(26.2 \pm 1.3)$ ‰; Vicars and Savarino, 2014), although falling at the upper end of the range. Interestingly, Vicars and Savarino (2014) reported a significant peak in $\Delta^{17}O(O_3)_{bulk}$ during February-March in Grenoble, France (located 120 km southwest of Chamonix), based on year-round measurements with $\Delta^{17}O(O_3)_{bulk}$ values 2−3 ‰ higher than the annual mean of 26.2 ‰. Although the cause of increased values during this period is unknown, our derived $\Delta^{17}O(O_3)_{bulk}$ matches remarkably well the February-March measurements reported by Vicars and Savarino (2014).

496

It is worth pointing out that a more accurate calibration of $\Delta^{17}O(NO_2)$ measurements is desirable. There is currently no internationally accepted nitrite salt standard with a positive $\Delta^{17}O$. Nevertheless, we are rather confident in our present calibration methodology. Indeed, the $\Delta^{17}O(NO_2)$ values measured in Chamonix closely align with previous observations in Grenoble. In both studies, the maximum (daytime) and minimum (nighttime) $\Delta^{17}O(NO_2)$ measurements conform to the expected values derived from the $\Delta^{17}O$ theoretical framework (based on well-established $NO_x$ chemistry and $\Delta^{17}O(O_3)$

measurements) when $O_3$ overwhelmingly dominates the NO to $NO_2$ conversion. Therefore, at this stage, we do not consider that $\Delta^{17}O(NO_2)$ calibration is an issue for this study. However, as we cannot completely rule out a small bias in our calibration, we have manufactured enriched nitrite salts and are presently working on refined new nitrite salt standards. Note that the rapid exchange of nitrite O isotopes in aqueous solution (Casciotti et al., 2007) is a challenge for inter-laboratory data comparisons. If new measurements of nitrite $\Delta^{17}O$ references are found to differ from our current calibration, we plan to publish a correction to the paper with updated values and any potential implications.

### 3.3 Interpretation of $\Delta^{17}O$ in atmospheric nitrate

Over the two sampling periods, $\Delta^{17}O(NO_3^-)$ varies significantly (from 18.3 ‰ to 28.1 ‰), with a weighted mean of (22.5 ± 3.1) ‰. While $\Delta^{17}O(NO_2)$ values are relatively similar during the two sampling periods, $\Delta^{17}O(NO_3^-)$ values are systematically higher during SP 2 than during SP 1, except during the 7:30−10:00 LT interval. $\Delta^{17}O(NO_3^-)$ in Chamonix is in the same range of most previous observations in urban environments (9−44 ‰; e.g., Kim et al., 2023; Wang et al., 2023; Fan et al., 2023; Zhang et al., 2022b; Lim et al., 2022; Li et al., 2022b), but lower than most values measured during the cold season which are typically >25 ‰. Unlike $\Delta^{17}O(NO_2)$, daytime and nighttime $\Delta^{17}O(NO_3^-)$ values at our site are not significantly different ($p$-value > 0.05, n = 14). Similarly, from 12 h resolved sampling in winter Beijing, He et al. (2018) found no significant difference between daytime and nocturnal $\Delta^{17}O(NO_3^-)$ and suggest that each sample reflects $NO_3^-$ produced during both the day and night. From high-time-resolved (3 h) aerosol sampling in winter Beijing, Zhang et al. (2022b) reported $\Delta^{17}O(NO_3^-)$ values between 23.4 ‰ to 39.3 ‰, with higher values observed at night ((31.0 ± 2.6) ‰) than during the day ((29.3 ± 3.0) ‰). This diurnal behaviour of $\Delta^{17}O(NO_3^-)$ was attributed to the changes in the branching ratio of nocturnal and photochemical reactions on $NO_3^-$ formation. In Chamonix, the range of $\Delta^{17}O(NO_3^-)$ values are very different from Zhang et al., (2022b) observations, with consistently lower values and a distinct diurnal tendency. However, in the cases of $\Delta^{17}O(NO_3^-)$ measurements at sub-daily temporal scale, the atmospheric lifetime of $NO_x$ and $NO_3^-$ is critical for comparing $\Delta^{17}O(NO_3^-)$ records from one site to another. Pollutant levels and atmospheric conditions between Chamonix and Beijing are very different, notably in winter when Asian urban areas can experience severe haze pollution episodes with $NO_3^-$ mass concentration exceeding 70 μg m$^{-3}$, which is over 10 times higher than in Chamonix (Lim et al., 2022; He et al., 2018; Zhang et al., 2022b). In such conditions, PM can reach several hundreds of μg m$^{-3}$ for several days, which can significantly impact atmospheric processes involved in the formation of secondary species. Aside from the intrusion of Saharan dust during SP 2, the pollutant level in Chamonix is indicative of a moderately polluted region, with significant diurnal variations.

### 3.3.1 Steady state evaluation of $\Delta^{17}(NO_3^-)$

To investigate the factors influencing the variability of $\Delta^{17}O(NO_3^-)$ at our site, one compare observed $\Delta^{17}O(NO_3^-)$ with estimated values of $\Delta^{17}O(NO_3^-)$ derived from $\Delta^{17}O$ mass balance and observed $\Delta^{17}O(NO_2)$, assuming the OH and $N_2O_5$ pathways dominate the formation of $NO_3^-$ at our site. Therefore, calculated $\Delta^{17}O(NO_3^-)$ reflect the theoretical $\Delta^{17}O$ transfer during the oxidation of $NO_2$ to $NO_3^-$ at our site through the dominant chemical process during the day (i.e. OH pathway) and at night (i.e. $N_2O_5$ pathway).

As presented, during the day, we consider that the conversion of $NO_2$ into $NO_3^-$ is predominantly influenced by Reaction (R5) (OH pathway). Hence, the theoretical corresponding $^{17}O$-excess transfer to $NO_3^-$ is estimated using Eq. (6) and observed $\Delta^{17}O(NO_2)$ between 7:30 and 18:00 LT (n = 3 per sampling period). Then, in order to estimate a daytime average value of $\Delta^{17}O(NO_3^-)$ which is representative of the potential for the formation of $NO_3^-$ from surface $NO_2$ by the OH pathway, each calculated $\Delta^{17}O(NO_3^-)$ is weighted by the product $[NO_2] \times J_{NO_2}$ (the diurnal variability of the OH mixing ratio is assumed to follow the diurnal $J_{NO_2}$ variation; Liu et al., 2021). Finally, an overall mean daytime $\Delta^{17}O(NO_3^-)$ for SP 1 and SP 2 is estimated by taking the sum of the weighted calculated values ($=\Delta^{17}O_{calc}(NO_3^-)$). The same approach is used during the night, assuming that the conversion of $NO_2$ into $NO_3^-$ is dominated by Reactions (R6)−(R8) ($N_2O_5$ pathway). Eq. (7) and observed $\Delta^{17}O(NO_2)$ between 18:00 and 7:30 LT (n = 3 per sampling day) are used to estimate $\Delta^{17}O(NO_3^-)$. Each calculated $\Delta^{17}O(NO_3^-)$ is weighted by the product $[NO_2] \times [O_3]$ (i.e., $NO_3$ production rate) and summed to estimate a mean nighttime $\Delta^{17}O(NO_3^-)$ for SP 1 and SP 2. The $^{17}O$-excess transferred from $O_3$ to $NO_2$ during Reaction (R6) ($\Delta^{17}O_{NO_2+O_3}(NO_3)$) is fixed at 44.7 ‰. This value is set accordingly to the transfer function reported by Berhanu et al. (2012) whereby $\Delta^{17}O_{NO_2+O_3}(NO_3) = (1.23 \pm 0.19) \times \Delta^{17}O(O_3)_{bulk} + (9.02 \pm 0.99)$ and $\Delta^{17}O(O_3)_{bulk} = 29.0$ ‰ (see Section 2.4.1). We compare hereafter $\Delta^{17}O_{calc}(NO_3^-)$ with the weighted day and night averages of observed $\Delta^{17}O(NO_3^-)$ at our site. During the day, $\Delta^{17}O_{calc}(NO_3^-)$ is compared with $\Delta^{17}O(NO_3^-)$ observations averaged between 7:30 and 18:30 LT (n = 3). At night, $\Delta^{17}O_{calc}(NO_3^-)$ is compared with $\Delta^{17}O(NO_3^-)$ observations averaged between 18:30 to 7:30 LT.

At night during SP 1, observed $\Delta^{17}O(NO_3^-)$ and $\Delta^{17}O_{calc}(NO_3^-)$ are in good agreement ($\Delta^{17}O_{calc}(NO_3^-) - \Delta^{17}O(NO_3^-) = \Delta^{17}(NO_3^-{}_{calc} - NO_3^-{}_{obs}) = 0.9$ ‰), suggesting a local and rapid (< 12 h) conversion of $NO_2$ into $NO_3^-$ via the $N_2O_5$ pathway. During the day, observed $\Delta^{17}O(NO_3^-)$ is 0.5 ‰ higher than $\Delta^{17}O_{calc}(NO_3^-)$, also suggesting that $NO_3^-$ is formed locally during the day for oxidation of surface $NO_2$ through the OH pathway. Small differences between observed and calculated $\Delta^{17}O$ of $NO_3^-$ during the day/night could be explained by the presence of $NO_3^-$ residues formed during the previous night/day, which are not considered in the calculations since they do not account for $NO_3^-$ lifetime. In contrast to SP 1, $\Delta^{17}O_{calc}(NO_3^-)$ during SP 2 is significantly lower than the mean observed $\Delta^{17}O(NO_3^-)$, particularly during the day with a

$\Delta^{17}(NO_3^-{}_{calc} - NO_3^-{}_{obs})$ of −6.4 ‰. The significant gap between observed and calculated $\Delta^{17}O(NO_3^-)$ suggests a different
origin and/or formation process of $NO_3^-$ during SP 2 compared to SP 1. Although less important than during the day,
$\Delta^{17}O_{calc}(NO_3^-)$ values for SP 2 at night is lower by 2.2 ‰ to the observed value. This small shift can be explained by
residuals of enriched daytime $NO_3^-$. It is important to point out that, although the $NO_2$ sample collected on Feb 24 between
13:30 and 16:30 LT presents an important blank (ca. 14 %), ambient $NO_2$ is low during the sampling period (mean of (4.5 ±
1.8) nmol mol$^{-1}$). Therefore, as each $\Delta^{17}O$ value used to estimate $\Delta^{17}O_{calc}(NO_3^-)$ is weighted by the mean ambient $NO_2$
mixing ratio over the sampling period, the incertitude related to this blank has little influence on the daily average of
$\Delta^{17}O_{calc}(NO_3^-)$. Given the low $\Delta^{17}(NO_3^-{}_{calc} - NO_3^-{}_{obs})$ during SP 1, observed $\Delta^{17}O(NO_3^-)$ can be explained by the local and
rapid (< 12 h) oxidation of $NO_2$, dominated by the OH and $N_2O_5$ pathway during the day and night, respectively. However,
in contrast to SP 1, the $^{17}O$-excess measured in $NO_3^-$ during the day of SP 2 cannot be fully constrained by the oxidation of
surface $NO_2$ through the OH pathway, suggesting that the formation mechanisms of $NO_3^-$ are different between SP 1 and SP
2 and/or the presence of $NO_3^-$ not formed locally during SP 2. Below we examine the changes in the sub-daily dynamics of
$\Delta^{17}O(NO_3^-)$ between SP 1 and SP 2 in light of atmospheric observations.

|  |  | $\Delta^{17}O(NO_2)$ /‰ | $\Delta^{17}O(NO_3^-)$ /‰ | $\Delta^{17}O_{calc}(NO_3^-)$ /‰ | $\Delta^{17}(NO_3^-{}_{calc} - NO_3^-{}_{obs})$ |
|---|---|---|---|---|---|
| Daytime | SP 1 | 30.0 ± 7.3 | 23.0 ± 3.1 | 22.5 ± 4.6 | −0.5 |
| (7:30-18:00) | SP 2 | 26.1 ± 6.9 | 23.9 ± 3.8 | 17.5 ± 4.6 | −6.4 |
| Nighttime | SP 1 | 21.2 ± 1.1 | 20.5 ± 1.1 | 21.4 ± 0.7 | 0.9 |
| (18:00-7:30) | SP 2 | 20.8 ± 1.0 | 23.2 ± 1.0 | 21.0 ± 0.6 | −2.2 |

**Table 2.** Mean observed $\Delta^{17}O$ data of $NO_2$ ($\Delta^{17}O(NO_2)$) and $NO_3^-$ ($\Delta^{17}O(NO_3^-)$) in Chamonix, and mean calculated $\Delta^{17}O$ of $NO_3^-$
($\Delta^{17}O_{calc}(NO_3^-)$) using Eqs. (6) and (7) at day and night, respectively, constrained with observed $\Delta^{17}O(NO_2)$. Day and night calculated
values were weighted by $[NO_2] \times J_{NO_2}$ and $[NO_2] \times [O_3]$, respectively

### 578   3.3.2    $\Delta^{17}(NO_3^-)$ sub-daily dynamics

Between 7:30−10:30 LT, $\Delta^{17}O(NO_3^-)$ is very similar during SP 1 (18.3 ‰) and SP 2 (18.6 ‰). Nonetheless, on the
following sampling time step (i.e., between 10:30−13:30 LT), $\Delta^{17}O(NO_3^-)$ is significantly different between SP 1 (21.5 ‰)
and SP 2 (24.7 ‰). Intriguingly, between 7:30−13:30 LT, $\Delta^{17}O(NO_2)$ during SP 2 (26.9 ‰) is lower compared to SP 1 (32.8
‰). As a consequence, if one consider that $NO_3^-$ is formed from the oxidation of local $NO_2$ through identical pathways
during SP 1 and SP 2, observed $\Delta^{17}O(NO_3^-)$ should be lower during SP 2 than during SP 1. Therefore, the more pronounced
increase of observed $\Delta^{17}O(NO_3^-)$ during the 10:30−12:30 LT interval of SP 2 suggests a different origin and/or formation
channel of $NO_3^-$, as mentioned previously. One more piece of evidence is that, during this period of time, $PM_{10}$ and $NO_3^-$

levels increase significantly during SP 2, alongside the disruption of the inversion layer (depicted in Figure 1). It can be inferred that this rise in $PM_{10}$ is mostly due to the presence of Saharan dust. The simultaneous increase of $NO_3^-$ and of $\Delta^{17}O(NO_3^-)$ corroborates the hypothesis that this $NO_3^-$ was not formed from the oxidation of ambient $NO_2$. Furthermore, such an increase in $\Delta^{17}O(NO_3^-)$ can only be supported by the oxidation of $NO_2$ through the $N_2O_5$ pathway, which is not expected to be important during the day due to the rapid photolysis of $NO_3$ and its titration by NO (Brown and Stutz, 2012).

Interestingly, aerosol samplings conducted at various heights (8 m, 120 m, and 260 m above ground level) in Beijing, China, revealed a positive vertical gradient of $\Delta^{17}O(NO_3^-)$ in winter, from on average 29 ‰ to 33 ‰ (Fan et al., 2022). In summer, the $\Delta^{17}O(NO_3^-)$ values at the three altitudes were very similar. This increase of $\Delta^{17}O(NO_3^-)$ with altitude in winter was believed to result from a stratification of $NO_2$ to $NO_3^-$ oxidation processes due to low vertical mixing and elevated surface $NO_x$ emissions. However, the authors did not consider the potential variability of $\Delta^{17}O(NO_2)$ with altitude, which can be substantial in urban areas at night as low $\Delta^{17}O(NO_2)$ results from surface NO oxidation. For our study, we propose an alternative interpretation of the vertical variability of $\Delta^{17}O(NO_3^-)$, where $\Delta^{17}O(NO_2)$ is considered as the main driver. During the formation of the nocturnal boundary layer, $NO_2$ formed during the day can be trapped above the surface layer in the nocturnal residual layer (NRL). This $NO_2$ has a high $\Delta^{17}O$ because it was formed during the previous daytime hours under the ISS framework (Eq. (3)). Throughout the night, this highly enriched $NO_2$ (ca. 37 ‰ which is the average of the maximum $\Delta^{17}O(NO_2)$ during SP 1 and SP 2) can be converted to $NO_3^-$ via the $N_2O_5$ pathway, hence leading to a substantial $\Delta^{17}O$ transfer to $NO_3^-$ at around 32 ‰, which is in the range of $\Delta^{17}O(NO_3^-)$ observed by Fan et al. (2022) in winter. In the meantime, NO emitted at the surface during the night can be converted to $NO_2$ by $O_3$, with a $\Delta^{17}O$ transfer of ca. 20 ‰ (Eq. (5) with $x = 0$). This low enriched $NO_2$ can be further oxidised to $NO_3^-$ by the $N_2O_5$ pathway which results in a $\Delta^{17}O$ transfer at around 21 ‰. This $NO_2$ with a low $\Delta^{17}O$ is very likely to be formed only at the surface during the night in areas experiencing important $NO_x$ emissions (Michalski et al., 2014). Furthermore, surface $NO_2$ with low $\Delta^{17}O$ is not expected to be transported aloft as it is formed in the surface inversion layer during the night. Therefore, $NO_3^-$ formed in the NRL during winter nights may be more enriched than the $NO_3^-$ formed concurrently at the surface, regardless of the $NO_2$ oxidation process involved. When the inversion layer breaks during the following day, the $NO_3^-$ that was formed in the NRL during the night is mixed with the $NO_3^-$ formed at the surface, resulting in an increase in the overall surface $\Delta^{17}O$. In this scenario, the presence of the Saharan dust during SP 2 may have increased the $NO_3^-$ loading aloft by promoting heterogeneous processes on aerosol surfaces in the vicinity of Chamonix. Hence, $NO_2$ stratification at night could explain the observed increase in $\Delta^{17}O$ of $NO_3^-$ at the surface following the collapse of the nocturnal inversion layer. However, we cannot determine whether the enriched $NO_3^-$ were formed in the vicinity of Chamonix and/or transported to our site by Saharan dust.

Although the exact nature of the high $^{17}$O-excess measured in $NO_3^-$ during SP 2 remains unclear, boundary layer
dynamics is thought to play a significant role in the variability of $\Delta^{17}O(NO_3^-)$ at the surface due to the stratification of $NO_2$.
Therefore, a wider consideration of such factors should be explored to avoid possible over-interpretation of $\Delta^{17}O(NO_3^-)$
variabilities at the surface, especially in urban areas experiencing significant boundary layer dynamics in winter and high
surface emissions of $NO_x$ at night. Measuring $\Delta^{17}O(NO_2)$ at various altitudes could provide better insights on the vertical
dynamics of $\Delta^{17}O(NO_3^-)$, and subsequently quantitative information on $NO_3^-$ production processes.

### 625    3.4    Nitrogen isotopic compositions

### 626    3.4.1    N fractionation effects in the $NO_x$ cycle

Over the two sampling periods, $\delta^{15}N(NO_2)$ shows substantial diurnal variability (from −10.0 ‰ to 19.7 ‰, n = 16) with a
weighted mean of (4.0 ± 9.1) ‰. In contrast, Albertin et al. (2021) reported a weak diurnal fluctuation of $\delta^{15}N(NO_2)$ in
spring in Grenoble, in a narrow range from about −12 ‰ to −10 ‰. In summer in an urban/suburban location, Walters et al.
(2018) also observed a wide range of $\delta^{15}N(NO_2)$ values, however, unlike our study, these are almost consistently negative
(from −31.4 ‰ to 0.4 ‰) with an overall mean at (−11.4 ± 6.9) ‰. As shown in Eq. (8), fluctuations in $\delta^{15}N(NO_2)$ reflect
changes in $NO_x$ emission sources and/or N fractionation effects, these latter being weighted by $1 - f_{NO_2}$ i.e., the more $NO_x$ is
under the form of NO, the greater the N fractionation effects (see Section 2.4). Hence, in the previous works of Albertin et
al. (2021) and Walters et al. (2018), due to high $f_{NO_2}$ (> 0.7), isotope effects were small (<2.7 ‰) and $\delta^{15}N(NO_2)$ was mostly
driven by changing contribution of $NO_x$ emission sources. At our site, $f_{NO_2}$ shows a wider range, from 0.3 to 1.0, suggesting
significant N isotopic fractionation effects, with minimum and maximum contributions corresponding to the highest and
lowest observed value of $\delta^{15}N(NO_2)$, respectively. This pronounced seasonal behaviour of N isotope fractionation effects
within the $NO_x$ cycle has previously been outlined in the seminal study of Freyer et al. (1993). Overall, compared with
summer, lower $f_{NO_2}$ during winter months due to lower $O_3$ concentrations and higher $NO_x$ emissions favour EIE between NO
and $NO_2$, which also has a higher fractionation factor due to the lower temperatures (see Appendix D: Equilibrium N
fractionation factors). Besides, this seasonal fluctuation of $f_{NO_2}$ can be expected to be observed on smaller time scales,
typically on the diurnal scale in urban areas where NO is generally fully oxidised into $NO_2$ at night due to a lower $NO_x$
emission rate resulting in higher $f_{NO_2}$ at night than during the day, as observed at our sampling (Figure 2).

Figure 3 shows the linear dependence of $\delta^{15}N(NO_2)$ on $(1 - f_{NO_2})$ over the two sampling periods, indicating the
significant influence of atmospheric processes that alter the N isotopic distribution during the conversion of $NO_x$ into $NO_2$.
The linear regression gives a slope and an intercept of about (43.6 ± 3.3) ‰ and (−8.8 ± 1.0) ‰, respectively. According to
Eqs. (10) and (11), the linearity between daytime (07:30−18:00 LT) and nighttime (18:00−07:30 LT) values suggests that
EIE dominates the N fractionation processes between $NO_x$ and $NO_2$. The influence of LCIE during the day could explain the
greater variability around the linear fit in the daytime observations.

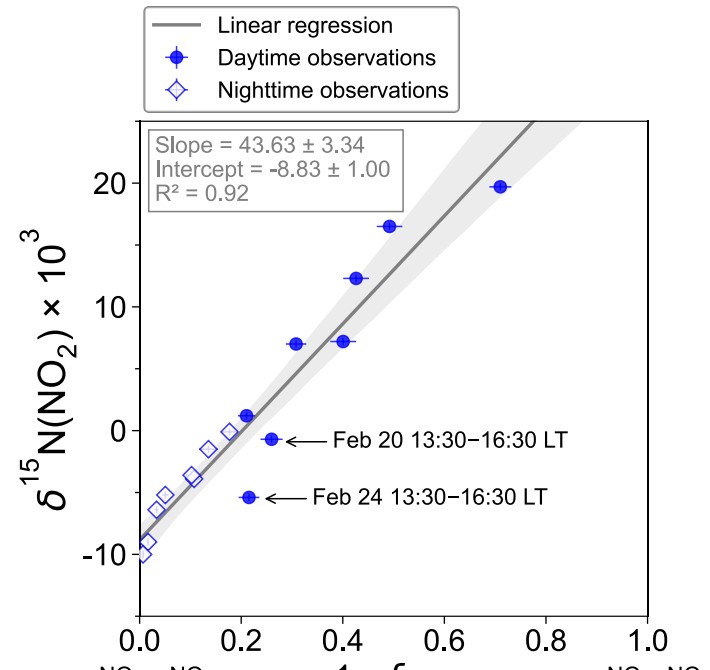

**Figure 3.** Correlation plot of $\delta^{15}N$ of atmospheric $NO_2$ vs. $(1 - f_{NO_2})$ from observations in Chamonix in February 2021. $f_{NO_2}$ are averaged
over the collection period of each $NO_2$ sample. The grey shade is the 95 % confidence interval. White diamonds and blue dots represent
the nighttime (18:00−07:30 LT) and daytime (07:30−18:00 LT) observations, respectively. The linear regression is plotted over the
nighttime and daytime observations.
The relative importance of EIE and LCIE in the N fractionation between emitted $NO_x$ and $NO_2$ is assessed by
calculating the $A^*$ factor during the day (7:30−18:00 LT; $A^*_{day}$) and night (18:00−7:30 LT; $A^*_{night}$) (Table 3). $A^*_{day}$ and $A^*_{night}$
expressions are given in Section 2.4.2. Overall, during $NO_2$ sampling intervals, the $A^*$ values are small (mean ± one standard
deviation: $0.21 \pm 0.51$) and reflect an EIE-dominated regime with high $NO_x$ (Li et al., 2020). It is interesting to note that the
highest $A^*$ values are observed between 13:30 and 16:30 LT, and correspond to the two data points in Figure 3 that lie
outside the 95 % confidence interval of the regression line. These results suggest that EIE is the dominant N fractionation
processes between $NO_x$ and $NO_2$ during both day and night ($A^* < 0.46$), with the exception of mid-afternoon when LCIE
competes with EIE ($A^* > 0.46$).

667        To quantify the overall N fractionation effect ($F_N$) between $NO_x$ and $NO_2$, we dissociate the two samples collected

between 13:30 and 16:30 LT into a different group (Group #1 = GP 1) from the other samples (Group #2 = GP 2). $F_N$ of GP
1 and GP 2 is calculated using Eq. (10) (which combines LCIE and EIE regimes) and Eq. (11) (which considers only the EIE
regime), respectively. Calculated $F_N$ are reported in Table 3 and data used for calculations can be found in Section 2.4.2 and
in the Supplement. Calculated $F_N$ is significantly different between GP 1 and GP 2, with a mean of 16.4 ‰ and 42.3 ‰,
respectively. The close match between the calculated average $F_N$ of GP 2 and the observed $F_N$ ((43.6 ± 3.3) ‰; slope of the
regression line in Figure 3) provides strong evidence for the reliability of Eq. (11), as well as the expression of $\alpha_{EIE(NO_2/NO)}$
used therein, to accurately describe the N fractionation between $NO_x$ emissions and $NO_2$ at our site, and hence, to describe
most of the variability of $\delta^{15}N(NO_2)$ measurements. This result holds significant importance in confirming the theoretical N
isotopic fractionation framework used in prior research studies. It is also important to stress the influence of LCIE effects for
GP 1, highlighting the high dependency of $\delta^{15}N(NO_2)$ to local environmental conditions. According to the $A^*$ factor, a
greater influence of LCIE in mid-afternoon could have contributed to the outlying of the two samples collected between
13:30 and 16:30 LT (GP 1). However, as mentioned above, the sample collected on Feb 24 between 13:30 and 16:30 LT has
a significant blank. Therefore, it cannot be confirmed with certainty that the reason this sample falls outside the 95 %
confidence interval of the regression line is solely due to LCIE. Nevertheless, the overall conclusion that EIE dominates the
variability of $\delta^{15}N(NO_2)$ at our site is not affected by this uncertainty.

684        The $\delta^{15}N$ shift in $NO_2$ relative to emitted $NO_x$ ($\Delta^{15}(NO_2 - NO_x)$) is calculated for individual $NO_2$ sample using the mean

ambient temperature during each sampling period. The mean atmospheric $\delta^{15}N$ of $NO_x$ ($\delta^{15}N(NO_x)$) is then estimated by
subtracting the $\Delta^{15}(NO_2 - NO_x)$ value from the observed $\delta^{15}N(NO_2)$ value. $\Delta^{15}(NO_2 - NO_x)$ and $\delta^{15}N(NO_x)$ estimates are
reported in Table 3. $\Delta^{15}(NO_2 - NO_x)$ varies greatly over the two sampling periods (from 0.7 ‰ to 30.7 ‰) with a mean value
of ca. 9 ‰ (mean of GP #1 and GP #2). $\delta^{15}N(NO_x)$ show much less variability with an overall mean at (−7.8 ± 1.9) ‰ (mean
of GP #1 and GP #2), in very good agreement with the value derived from the regression relationship (−8.8 ‰; intercept of
the regression line in Figure 3). Therefore, there appears that there is little variation in $NO_x$ emission sources at our site, and
the wide variability in $\delta^{15}N(NO_2)$ is mainly driven by important equilibrium post-emission isotopic effects.

| Sampling interval (start – end) | $A^{*(1)}$ | $F_N^{(2)}$ | $f_{NO_2}$ | $\Delta^{15}(NO_2 - NO_x)^{(3)}$ /‰ | $\delta^{15}N(NO_x)$ /‰ |
|---|---|---|---|---|---|
| **GP #1** | | | | | |
| 20/02 13:30 – 20/02 16:30 | 0.46 ± 0.08 | 25.00 ± 2.27 | 0.74 ± 0.02 | 6.5 ± 0.9 | −7.2 ± 0.9 |
| 24/02 13:30 – 24/02 16:30 | 2.09 ± 0.39 | 6.90 ± 1.97 | 0.78 ± 0.02 | 1.5 ± 0.4 | −6.9 ± 0.6 |
| *Mean* | *1.27* | *16.39* | *0.76* | *4.1* | *−7.1* |
| *Std dev* | *1.14* | *12.58* | *0.03* | *3.5* | *0.2* |
| **GP #2** | | | | | |
| 19/02 21:00 – 20/02 00:30 | 0.03 ± 0.01 | 43.06 ± 0.17 | 0.97 ± 0.01 | 1.4 ± 0.1 | −7.8 ± 0.3 |
| 20/02 00:30 – 20/02 04:30 | 0.07 ± 0.01 | 43.51 ± 0.18 | 0.98 ± 0.01 | 0.7 ± 0.1 | −9.7 ± 0.3 |
| 20/02 04:30 – 20/02 07:30 | 0.02 ± 0.01 | 43.78 ± 0.22 | 0.89 ± 0.01 | 4.7 ± 0.4 | −8.6 ± 0.5 |
| 20/02 07:30 – 20/02 10:30 | 0.05 ± 0.01 | 43.92 ± 0.18 | 0.60 ± 0.02 | 17.6 ± 1.0 | −10.4 ± 1.1 |
| 20/02 10:30 – 20/02 13:30 | 0.10 ± 0.02 | 43.06 ± 0.18 | 0.57 ± 0.02 | 18.4 ± 1.0 | −6.1 ± 1.0 |
| 20/02 16:30 – 20/02 18:00 | 0.03 ± 0.01 | 39.97 ± 0.18 | 0.69 ± 0.02 | 12.3 ± 0.8 | −5.3 ± 0.8 |
| 20/02 18:00 – 20/02 21:00 | 0.01 ± 0.01 | 41.75 ± 0.19 | 0.90 ± 0.01 | 4.3 ± 0.5 | −7.9 ± 0.6 |
| 24/02 07:30 – 24/02 10:30 | 0.01 ± 0.01 | 43.21 ± 0.18 | 0.29 ± 0.02 | 30.7 ± 0.9 | −11.9 ± 0.9 |
| 24/02 10:30 – 24/02 13:30 | 0.07 ± 0.01 | 41.95 ± 0.18 | 0.51 ± 0.02 | 20.6 ± 1.0 | −4.1 ± 1.1 |
| 24/02 16:30 – 24/02 18:00 | 0.16 ± 0.03 | 39.80 ± 0.16 | 0.79 ± 0.02 | 8.4 ± 0.6 | −7.2 ± 0.7 |
| 24/02 18:00 – 24/02 21:00 | 0.01 ± 0.01 | 40.88 ± 0.18 | 0.82 ± 0.02 | 7.2 ± 0.6 | −7.3 ± 0.6 |
| 24/02 21:00 – 25/02 00:00 | 0.03 ± 0.02 | 42.20 ± 0.19 | 0.95 ± 0.01 | 2.1 ± 0.3 | −7.3 ± 0.4 |
| 25/02 00:00 – 25/02 04:00 | 0.19 ± 0.03 | 42.48 ± 0.18 | 0.99 ± 0.01 | 0.3 ± 0.1 | −10.3 ± 0.3 |
| 25/02 04:00 – 25/02 07:30 | 0.09 ± 0.01 | 42.69 ± 0.17 | 0.86 ± 0.02 | 5.8 ± 1.0 | −7.3 ± 1.0 |
| *Mean* | *0.06* | *42.31* | *0.77* | *9.6* | *−7.9* |
| *Std dev* | *0.06* | *1.32* | *0.21* | *9.1* | *2.0* |

[1] Calculated from $A^*_{day}$ between 7:30–18:00 LT and from $A^*_{night}$ between 18:00–07:30 LT ($A^*_{day}$ and $A^*_{night}$ expressions are given in Section 2.4.2).
[2] Calculated from Eq. (10) for GP 1 and from Eq. (11) for GP 2
[3] Calculated from Eq. (9)

**Table 3.** Summary table of data used to estimate the N isotopic fractionation between $NO_x$ emissions and $NO_2$ at our site ($\Delta^{15}(NO_2 - NO_x)$) and derive $NO_x$ emissions $\delta^{15}N$-fingerprint $\delta^{15}N(NO_x)$. The data reported are the mean values for each $NO_2$ sampling period (mean value ± absolute uncertainty).

### 3.4.2 $NO_x$ emission sources derived from $\delta^{15}N(NO_2)$

To identify the main source of $NO_x$ that contributes to the calculated $\delta^{15}N(NO_x)$ values at our site, Figure 4 displays the temporal variation of $\delta^{15}N(NO_x)$ obtained from individual $NO_2$ samples (dashed horizontal line) and the $\delta^{15}N$ range for different $NO_x$ emission sources (coloured bands) such as for coal combustion ((19.5 ± 2.3) ‰ for power plant with selective catalytic reduction technology; Felix et al., 2012; Elliott et al., 2019), fossil gas combustion ((−16.5 ± 1.7) ‰; Walters et al., 2015), and fertilised soils ((−33.8 ± 12.2) ‰; Miller et al., 2018). $\delta^{15}N$ of $NO_x$ released during biomass combustion is primarily driven by the $\delta^{15}N$ of the biomass burnt (Fibiger and Hastings, 2016). We estimate an average $\delta^{15}N$ of biomass combustion $NO_x$ at (−0.1 ± 1.3) ‰, using the empirical relationship of Chai et al. (2019) (which was derived from

combustions of several North American wood species) and an average $\delta^{15}N$ of biomass at $(-2.8 \pm 2.0)$ ‰ representative of
temperate forests (Martinelli et al., 1999). Regarding road traffic emissions, we have to stress that $\delta^{15}N$ values reported in the
literature are rather variable mainly because N fractionations during the process of $NO_x$ production can vary depending on
the type of fuel used, the type of vehicle, the presence of an emission control system, and the time of commuting (Ammann
et al., 1999; Felix and Elliott, 2014; Heaton, 1990; Miller et al., 2017; Walters et al., 2015b; Zong et al., 2020, 2017). We use
here the mean vehicle-emitted $\delta^{15}N(NO_x)$ value given by Song et al. (2022) at $(-7.1 \pm 4.1)$ ‰, calculated from 181
measurements reported in the literature.

712   As previously noted, the values of estimated $\delta^{15}N(NO_x)$ show much less variability than $\delta^{15}N(NO_2)$, with no significant

differences observed between daytime and nighttime values. The values of $\delta^{15}N(NO_x)$ range from $-11.0$ ‰ to $-4.1$ ‰, and
despite the associated uncertainty, they are consistent with the $\delta^{15}N$ range of $NO_x$ emissions from vehicle exhaust. The two
sampling periods show similar $\delta^{15}N(NO_x)$ values with a slight diel variability. The estimated small variation in $\delta^{15}N(NO_x)$
throughout the day can be attributed to the temporal changes in the $\delta^{15}N$ signature of mobile $NO_x$ sources. It has been shown
that $NO_x$ emitted by cold engines has a lower $\delta^{15}N$ signature compared to $NO_x$ emitted from warm engines (Walters et al.,
2015b). Hence, the early morning drop in $\delta^{15}N(NO_x)$ could be attributed to the influence of $NO_x$ emitted from cold engines.
As the day progresses, the time of commuting increases and therefore $\delta^{15}N(NO_x)$ tends to be less negative. Conversely,
during the night, the slow $\delta^{15}N(NO_x)$ decline could be due to the replacement of $NO_x$ from vehicle exhaust by $NO_x$ emitted
by fossil gas combustion, which is commonly used in Chamonix for home heating. Although biomass burning used for home
heating would also tend to increase $\delta^{15}N(NO_x)$ during the day, it is unlikely to contribute more during the day than at night.

724   According to local $NO_x$ emission inventories (Atmo-Auvergne-Rhône-Alpes, 2018; ORCAE, 2022), road transport is

responsible of 64 % of $NO_x$ emissions, ahead of heating oil and fossil gas combustion. Despite the consistency between our
results and existing inventories, the significant variability in the $\delta^{15}N$ signature of $NO_x$ emissions from vehicle exhaust
precludes a reliable quantitative source apportionment of $NO_x$ emissions from our estimated $\delta^{15}N(NO_x)$. Furthermore, the
lack of information on the exact $\delta^{15}N$ signature of $NO_x$ emitted from heating-oil combustion could also contribute to the
potential bias of the emission source apportionment.

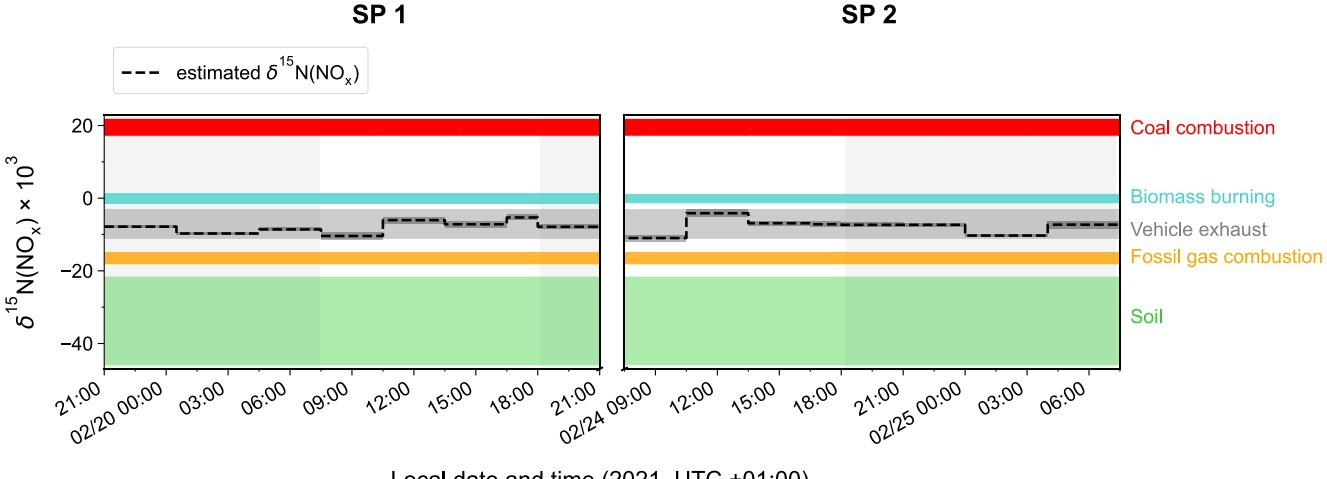

**Figure 4.** Time evolution of $\delta^{15}N(NO_x)$ (black dashed line) estimated from $\delta^{15}N(NO_2)$ observations in Chamonix after correction of N fractionation effects (length of horizontal line = sampling period, black shaded area = overall calculation error bar). Coloured shaded areas represent the standard deviation of the mean $\delta^{15}N$ value of individual $NO_x$ emission source (coal combustion in red, biomass burning in blue, vehicle exhaust in grey, fossil gas in orange, and soil emissions in green). Grey backdrop shaded areas represent the nighttime (sunset to sunrise).

### 3.4.3 Interpretation of $\delta^{15}N(NO_3^-)$ observations

$\delta^{15}N(NO_3^-)$ also exhibits substantial variability during the day, ranging from −1.3 ‰ to 14.9 ‰ and from −4.2 ‰ to 9.7 ‰ during SP 1 and SP 2, respectively. At night, $\delta^{15}N(NO_3^-)$ is less variable, with an overall mean of (1.4 ± 1.2) ‰ and (−1.1 ± 0.4) ‰ during SP 1 and SP 2, respectively. $\delta^{15}N(NO_3^-)$ is within the range of observations reported in urban areas (He et al., 2020; Zhang et al., 2022a). A similar diurnal pattern was observed in samples collected during a cruise along the Californian coast in spring 2010 (Vicars et al., 2013), and isotopic exchanges between NO and $NO_2$ during the day were found to be the primary driver of the diel variability. In the previous section, we demonstrated that there is a significant $^{15}N$ partitioning between $NO_x$ emissions and $NO_2$, the latter being enriched in $^{15}N$ compared to $NO_x$ emissions. Interestingly, important enrichments in $^{15}N$ are also observed in $NO_3^-$.

As described above, at night during SP 1 and SP 2, $\delta^{15}N(NO_2)$ is close to $\delta^{15}N(NO_x)$ due to small N fractionation effects. However, between 18:00−7:30 LT, $NO_3^-$ is enriched in $^{15}N$ relative to $NO_2$ by +6.3 ‰ and +1.4 ‰ in average during SP 1 and SP 2, respectively (Table 4). If we assume that, at night, $NO_3^-$ is formed mainly by the conversion of surface $NO_2$ via the $N_2O_5$ pathway, then the difference between $\delta^{15}N(NO_3^-)$ and $\delta^{15}N(NO_2)$ ($\Delta^{15}(NO_3^- - NO_2)$) should reflect the N enrichment factor associated to this oxidation process. It is likely that an isotopic equilibrium is established between $NO_2$, $NO_3$, and $N_2O_5$, hence affecting the partitioning of $^{15}N$ between $NO_2$ and $NO_3^-$ produced at night (Walters and Michalski, 2016). Neglecting KIE associated with the $N_2O_5$ pathway and using the expression of the EIE fractionation factor between $N_2O_5$ and $NO_2$ given by Walters and Michalski (2015) (Appendix D) constrained with the mean nighttime temperature at

our site, the isotopic composition of $NO_3^-$ is expected be enriched in $^{15}N$ by about 29 ‰ compared to $NO_2$. This estimated $^{15}N$ enrichment is about three times higher than the observed $\Delta^{15}(NO_3^- - NO_2)$ at our site. As daytime $NO_3^-$ exhibits higher $\delta^{15}N$ values than during the night, it is not possible for daytime residuals at night to account for the lower than predicted fractionation effect between $NO_2$ and $NO_3^-$. These results highlight the importance of improving our understanding of the $^{15}N$ fractionation between $NO_2$ and $NO_3^-$ associated with the $N_2O_5$ pathway. This could be achieved in an atmospheric simulation chamber that allows to reproduce individual processes in controlled conditions. The $^{15}N$ isotopic enrichment of $NO_2$ and $NO_3^-$ collected from 7:30 to 18:00 LT shows a very contrasted distribution between SP 1 and SP 2, with a respective average $\Delta^{15}(NO_3^- - NO_2)$ of –0.4 ‰ and –10.0 ‰ (Table 4). Although subjected to significant uncertainties (Fan et al., 2019), the OH pathway is often associated to a KIE effect of −3 ‰ (Freyer, 1991), which is at odds with our observations. Similarly to the $N_2O_5$ pathway, there is an important need to better estimate the fractionation factor associated with the OH pathway.

There are significant differences in $\Delta^{15}(NO_3^- - NO_2)$ between SP 1 and SP 2, providing further evidence that $NO_3^-$ collected during these two periods has undergone different formation processes and/or originate from different sources of $NO_2$. In addition, possible fractionation associated with phase change between $HNO_3$ and $p$-$NO_3$ during transport of Saharan dust could influence the $\delta^{15}N$ of collected $NO_3^-$ during SP 2. However, given the lack of knowledge about N fractionation factors between $NO_2$ and $NO_3^-$ and our limited dataset, we cannot conclude whether the changes in the distribution of $NO_3^-$ isotopes during SP 2 result from changes in the phase distribution of $NO_3^-$ or in $NO_2$ oxidation processes.

|  |  | $\delta^{15}N(NO_2)$ /‰ | $\delta^{15}N(NO_3^-)$ /‰ | $\delta^{15}N(NO_x)$ /‰ | $\delta^{15}N(NO_3^- - NO_2)$ /‰ |
|---|---|---|---|---|---|
| Daytime | SP 1 | $7.4 \pm 4.7$ | $7.0 \pm 6.7$ | $-9.9 \pm 2.9$ | $-0.4$ |
| (7:30-18:00) | SP 2 | $14.0 \pm 13.9$ | $4.0 \pm 6.4$ | $-10.8 \pm 2.1$ | $-10.0$ |
| Nighttime | SP 1 | $-5.1 \pm 2.3$ | $1.4 \pm 1.2$ | $-9.0 \pm 0.8$ | $6.3$ |
| (18:00-7:30) | SP 2 | $-2.5 \pm 4.2$ | $-1.1 \pm 0.4$ | $-9.9 \pm 1.9$ | $1.4$ |

**Table 4.** Mean observed $\delta^{15}N$ data of $NO_2$ ($\delta^{15}N(NO_2)$) and $NO_3^-$ ($\delta^{15}N(NO_3^-)$), calculated atmospheric $\delta^{15}N$ of $NO_x$ ($\delta^{15}N(NO_x)$), and $\delta^{15}N$ shift between $\delta^{15}N(NO_3^-)$ and $\delta^{15}N(NO_2)$ ($\Delta^{15}(NO_3^- - NO_2)$).

## 4  Summary and implications

This study reports the first simultaneous measurements and analysis of $\Delta^{17}O$ and $\delta^{15}N$ in $NO_2$ and $NO_3^-$. The samplings were conducted at high temporal resolution (~ 3 h) in Chamonix, French Alps, over two distinct days in late February 2021. The isotopic signals of both $NO_2$ and $NO_3^-$ show substantial diurnal variabilities which are investigated in the light of local meteorological parameters and atmospheric observations (NO, $NO_2$, $O_3$, and PM).

781

782 The observed variability of $\Delta^{17}O(NO_2)$ can be well explained using $\Delta^{17}O$ mass balance equations and corroborates the

783 analysis of previous observations carried out in Grenoble, French Alps, over a single day in spring (Albertin et al., 2021). On

784 average, the high levels of $NO_2$ at our site are primarily driven by oxidation of local NO emissions by $O_3$. The observed

785 diurnal variability in $\Delta^{17}O(NO_2)$ appears to be consistent with the diurnal variability expected in the $NO_x/O_3/RO_2$ chemistry

786 with $RO_2$ levels of the order of pmol mol$^{-1}$ which is in agreement with the range of direct winter $RO_2$ measurements reported

787 in the literature. $RO_2$ is thought to contribute significantly to the formation of $NO_2$ in the early morning under high-$NO_x$

788 conditions, which is in line with effective morning production of radical species reported in urban areas in winter. At night,

789 $\Delta^{17}O(NO_2)$ reflects the nocturnal oxidation of surface NO emissions by $O_3$. These results provide additional evidence that

790 $\Delta^{17}O(NO_2)$ measurements represent valuable constraints in the study of the reactive $NO_x$ chemistry, down to the sub-daily

791 temporal scales.

792

793 A clear linear relationship is found between $\delta^{15}N(NO_2)$ and the $NO_2/NO_x$ ratio, indicating significant post-emission N

794 fractionation effects. Theoretical N isotopic fractionation factors between NO and $NO_2$ at equilibrium and fractionation

795 factors derived from the isotopic observations are found to be in good agreement, providing further support for the N

796 isotopic fractionation theoretical framework commonly applied to the Leighton cycle. Observed $\delta^{15}N(NO_2)$ corrected for N

797 fractionation effects allow to estimate the overall $\delta^{15}N$ signature of ambient $NO_x$ at our site. Based on the existing $\delta^{15}N$-

798 fingerprints of different $NO_x$ emission sources, the main contribution at our site is very likely to be vehicle exhaust, which is

799 confirmed by local emission inventories.

800

801 We use $\Delta^{17}O$ mass balance equations of $NO_3^-$ constrained by observed $\Delta^{17}O(NO_2)$ to assess whether $NO_3^-$ could

802 originate locally from the oxidation of $NO_2$ at our site. During the first day of sampling, $\Delta^{17}O$ records of $NO_2$ and $NO_3^-$

803 support the local oxidation of $NO_2$ to $NO_3^-$ by OH radicals during the day, and via the heterogeneous hydrolysis of $N_2O_5$

804 during the night. The second day of sampling was affected by a Saharan dust event, accompanied by notable changes in the

805 isotopic composition of $NO_3^-$. We propose that the formation of a surface inversion layer at night could have influenced the

806 vertical distribution of $\Delta^{17}O(NO_2)$ and resulted in a positive gradient of $\Delta^{17}O(NO_3^-)$ with altitude, independently of the local

807 $NO_2$ to $NO_3^-$ conversion processes near the surface. In such scenario, the presence of Saharan dust could have promoted

808 heterogeneous $NO_2$ oxidation leading to higher $\Delta^{17}O$ in $NO_3^-$ formed aloft. The latter would have then mixed with the $NO_3^-$

809 formed near the surface when the inversion breaks up during the day. Although still uncertain, the influence of the boundary

810 layer dynamics on the distribution of $\Delta^{17}O$ in $NO_3^-$ should be investigated in the future, notably for urban areas in winter.


812 The combined analysis of the first concurrent observations of $\delta^{15}N$ in $NO_2$ and $NO_3^-$ highlights persistent uncertainties

813 in the current estimates of the N fractionation factors associated with $NO_2$ and $NO_3^-$ conversion processes. However,

$\delta^{15}N(NO_3^-)$ records need to be corrected for N fractionation effects if they are to be used to trace back the $\delta^{15}N$ fingerprint of
the primary $NO_x$ emission sources. Detailed simulation chamber experiments could provide more kinetic data on the various
N fractionation processes in order to better exploit $\delta^{15}N(NO_3^-)$ records to identify and quantify of the sources of reactive
nitrogen.

The present thorough investigation of the $\Delta^{17}O$ and $\delta^{15}N$ in $NO_2$ and $NO_3^-$ highlights (1) the potential to use sub-daily
$\Delta^{17}O$ and $\delta^{15}N$ records to trace the sources and formation chemistry of $NO_3^-$, (2) the importance of measuring the $NO_2$
isotopic composition to avoid misinterpretation of $NO_3^-$ isotopic records, and (3) the persistent knowledge gaps that prevent
a complete assessment of the factors driving the variability in $NO_3^-$ isotopic records. In most studies, the $NO_3^-$ isotopic
composition is interpreted on the basis of estimates of the isotopic composition of its precursor gases, assuming that both the
chemistry of $NO_2$ (including its conversion to $NO_3^-$) and N isotopic fractionation effects are known. However, these
assumptions are subject to very significant uncertainties, mainly in urban atmospheres. Hence, given the recent development
of a method for measuring the multi-isotopic composition of $NO_2$, the accuracy and validity of the current interpretation
framework of $NO_3^-$ isotopic records should be tested in various environments. Such investigation can be performed by
collecting simultaneously $NO_2$ and $NO_3^-$, as done here. We recommend to use this combined isotopic in order to avoid
biased interpretations of $NO_3^-$ isotopic records, particularly in urban areas during winter, and preferably at high temporal
resolution (<24 h). In addition, the vertical distribution of $NO_2$ and $NO_3^-$ isotopic composition should be documented in
order to explore the possible role of the boundary layer dynamics in the variability of $NO_2$ and $NO_3^-$ isotopic composition
observed at the surface.

 **5     Appendix A: Reaction chemical rate**

| Reactions | Rate constants /$cm^3$ $mol^{-1}$ $s^{-1}$ | References |
|---|---|---|
| $NO + O_3 \rightarrow NO_2 + O_2$ | $k_{NO+O_3} = 1.4 \times 10^{-12} \exp(-1310(K)/T)$ | Atkinson et al. (2004) |
| $NO + RO_2 \rightarrow NO_2 + RO$ | $k_{NO+RO_2} = 2.3 \times 10^{-12} \exp(360(K)//T)$ | Atkinson et al. (2006) |
| $NO_2 + O_3 \overset{M}{\rightarrow} NO_3 + O_2$ | $k_{NO_2+O_3} = 1.4 \times 10^{-13} \exp(-2470(K)//T)$ | Atkinson et al. (2004) |
| $^{15}NO_2 + {}^{14}NO \rightarrow {}^{14}NO_2 + {}^{15}NO$ | $k_{NO+NO_2} = 8.14 \times 10^{-14}$ | Sharma et al. (1970) |

**Table A1.** Kinetic constants used in this study.
**6     Appendix B: Atmospheric lifetime of $NO_2$ and $NO_3^-$**

| | $\tau_{NO_2}$ [1] | $\tau_{NO_3^-}$ [2] | $k_{d(NO_2)}$ ($s^{-1}$) | $k_{d(NO_3^-)}$ ($s^{-1}$) |
|---|---|---|---|---|
| Daytime (7:30–18:00) | 5.1 min | 27.8 h | $0.5 \times 10^{-5}$ | $1.0 \times 10^{-5}$ |
| Nighttime (18:00–07:30) | 10.0 h | 5.6 h | $2.5 \times 10^{-5}$ | $5.0 \times 10^{-5}$ |

[1] Atmospheric lifetime relative to photolysis during the day (dry deposition and reaction $NO_2 + OH$ are negligible) and to dry deposition and oxidation via $O_3$ during the night.
[2] Atmospheric lifetime relative to dry deposition
The boundary layer is fixed at 500 m during the day and at 100 m during the night. Dry deposition velocity ($V_d$) is fixed at 0.25 cm $s^{-1}$ and 0.50 cm $s^{-1}$ for $NO_2$ and $NO_3^-$, respectively (Holland et al., 1999; Zhang et al., 2009).

**Table B1.** Mean daytime (07:30−18:00 LT) and nighttime (18:00−07:30 LT) atmospheric lifetime of $NO_2$ ($\tau_{NO_2}$) and $NO_3^-$ ($\tau_{NO_3^-}$) and dry
deposition constant ($k_d = V_d \times$ BLH where $V_d$ is the dry deposition velocity and BLH is the boundary layer height).

## 7 Appendix C: Schematic of the N reactive cycle and associated $\Delta^{17}O$ transfers and N enrichment factors

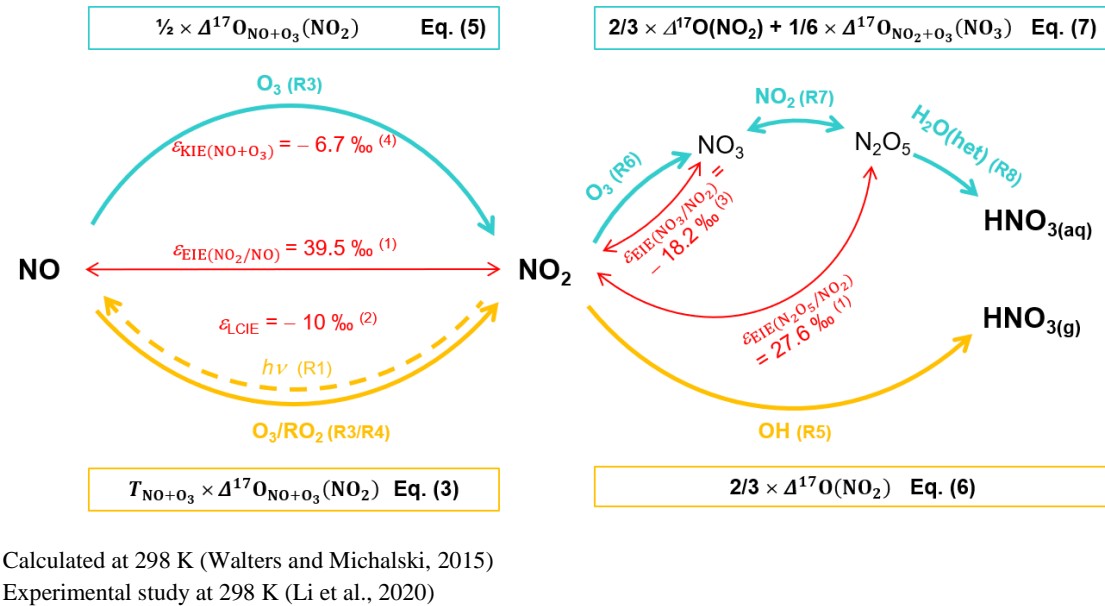

[1] Calculated at 298 K (Walters and Michalski, 2015)
[2] Experimental study at 298 K (Li et al., 2020)
[3] Calculated at 298 K (Walters et al., 2016)
[4] Calculated at 298 K (Fang et al., 2021)

**Figure C1.** Adapted from Elliott et al., (2019). Sketch of dominant daytime (thick yellow arrows) and nighttime (thick cyan arrows) $NO_x$ to $NO_3^-$ conversion processes and associated quantified N fractionation effects at 298 K (thin red arrows and text) and $\Delta^{17}O$ transfers (yellow and cyan boxes)

## 8 Appendix D: Equilibrium N fractionation factors

$$(\alpha_{EIE(X/Y)} - 1) \times 1000 = \frac{A}{T^4} \times 10^{10} + \frac{B}{T^3} \times 10^8 + \frac{C}{T^2} \times 10^6 + \frac{D}{T} \times 10^4$$

| X/Y | A | B | C | D |
|---|---|---|---|---|
| $NO_2/NO$ | 3.847 | −7.680 | 6.003 | −0.118 |
| $N_2O_5/NO_2$ | 1.004 | −2.525 | 2.718 | 0.135 |

$$(\alpha_{KIE(X+Y)} - 1) \times 1000 = A \times \exp(B/T)$$

| X + Y | A | B |
|---|---|---|
| $NO + O_3$ | 0.982 | 3.352 |

**Table D1.** Calculated regression coefficients for the N isotope exchange between $NO_2/NO$ and $N_2O_5/NO_2$ over the temperature range of 150 to 450 K (Walters and Michalski, 2015) and for the N kinetic fractionation for the reaction $NO + O_3$ over the temperature range of 220 to 320 K (Fang et al., 2021).

## 9    Data availability

Data presented in this article are included in the Supplement.

## 10    Author contributions.

Grants obtained by KL and JS funded the project. AB, RB, QF, and IV performed the calibration of IBBCEAS and OFCEAS instruments, data acquisition and post-processing. SA conducted the sampling and laboratory analysis. NC provided technical support for isotopic mass spectrometry analysis. The study was designed as part of SA's PhD thesis supervised by SB and JS. The paper was written by SA under the supervision of JS and SB and contributions from all co-authors.

## 11    Competing interests.

The authors declare that they have no conflict of interest.

## 12    Acknowledgements.

This work benefited from CNRS and IGE infrastructures and laboratory platforms. The authors acknowledge the support of the CASPA program (Climate-relevant Aerosol Sources and Processes in the Arctic). The authors gratefully acknowledge Catherine Coulaud for providing temperature data in Chamonix, Jean-Luc Jaffrezo for providing data of previous filter measurements in Chamonix, and Anthony Lemoine, Matthieu Lafaysse and Louis Le Toumelin for providing S2M reanalysis and meteorological data. The authors particularly thanks Patrick Ginot, Anthony Vella, Armelle Crouzet, and Bruno Jourdain for instrumental and laboratory technical support. Finally, the authors thank Alexis Lamothe for assistance during the sampling campaign and Pete Akers for his thorough proofreading of the manuscript and his constructive comments.

## 13    Financial support.

This research has been supported by the Agence Nationale de la Recherche (ANR) via contract ANR-21-CE01-0017 CASPA and INSU-CNRS (National Institute of Sciences of the Universe) via its national LEFE program (Les Enveloppes Fluides et l'Environnement) and by a grant from Labex OSUG@2020 (Investissements d'avenir – ANR10 LABX56) and IDEX-UGA ANR project ANR-15-IDEX-02.

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
