# Peer review of "Diurnal variations in oxygen and nitrogen isotopes of atmospheric"

_EGUsphere, 2023_

## Referee Comment (RC1)

The manuscript by Sarah Albertin reported the measurements of diurnal variations in oxygen and nitrogen isotopes of atmospheric NO2 and NO3 collected at a mountain site, and from which they discussed the conversion mechanisms of NOx and nitrate, as well as the implications for NOx source emissions. Overall the manuscript represents a step forward in understanding the sources and transformations of NOx using stable isotopes, though I have some comments which should be addressed before the manuscript can be published.

Major comments:
1) the way to derive RO2 using measured D17O(NO2): I don't think it make any sense to use the other method (i.e., Case B) to estimate RO2 and compare the results with Case A.

First of all, as shown in Table 1, in both SP1 and SP2, neither the derived RO2 concentration at each sampling period nor the averages can be considered as "consistent"; Don't get the points how can these values can be called "closeness" .

Secondly, to derived RO2 from  D17O(NO2) (i.e., Case A), one simply assumes that it is only RO2 completes with O3 to oxidize NO as indicated by Equation 13. This ignores the contributions of HO2. While in Case B, a RO2 /HO2 ratio of 0.859 was applied. So does this mean that in Case A the same ratio of RO2/HO2 can also be applied and then compared with Case B? In fact, it is highly doubtable that the same ratio in one study can be applied to another, given variations in concentrations of CO, CH4 and NMVOCs, don't even to mention the uncertainties associated with the empirical formula of HO2 calculation using O3.

So just saying from D17O(NO2) to estimate RO2 and note it is actually representing the sum of RO2 and HO2 is enough here. This provides a new methods to estimate RO2 and HO2 radicals, and can be verified with actual measurements in the future. The current way of Case B is just too uncertain and the results are not comparable.

2) the discussion on the D17O(NO3-) values of SP2: there are several issues, first, some values in Table 2 are wrongly calculated (this is an indicator that the manuscript should be better proofread):

The night average values of D17O(NO2) in SP2 night, is this really 29.8 permil?  In Figure 2 it never exceeds ~25 permil at nights. This leads to doubt on the calculated D17O(NO3-) values at SP 2 night as you need D17O(NO2) to calculate D17O(NO3).

In this same table, the difference between the calculated and observed D17O(NO3-) values at nights are simply wrong, it is -0.4 and -2.1 for SP 1 and SP 2.

Now back to the discussions on the dynamics of D17O(NO3-), I agree that the high  D17O(NO3-) values in SP 2 (both compared to SP1 and calculated values from D17O(NO2)) indicate no local surface NO2 oxidation, but it may also indicate no local nitrate production. Regional transport of nitrate could explain the disconnections between the observed D17(NO2) and D17(NO3-). It is problematic to attribute the anomalous D17O(NO3-) in SP2 (Only) to dynamics of boundary layer structure, as which may explain the high D17O(NO3-) at the daytime in SP 2, but how about at the night? Ass shown in Figure 2, in SP2, not only day time D17O(NO3-) are different,

but also the night values (apparently higher than D17(NO2) in SP2 night but not in SP 1 night). The authors paid too much attentions to the observations of Beijing and tried to make analogies between the two in order to explain their observations. However, the authors overlooked the differences in concentrations of PM2.5, PM10 and nitrate between the two days, and all of these concentrations in SP 2 are higher in SP2 than SP1, would this suggest a transport events and non-local sources of nitrate? Note the d15N(NO3-) values and its relationship with d15N(NO2) are also different in SP2 compared to SP1, this may indicate the same thing: a regional transport event occurred and which brought non-local nitrate. The d15N and D17O data should be combined to discuss at this point.

The last, references or other evidence should be provided when attributed the high load of PM10 in SP2 to Saharan dust, e.g., back trajectory analysis or something similar.

Technical comments:

1) Blank correction of isotope measurement of NO2: I wondered where the blank comes from, if it is coming from the chemicals used to trap NO2, there is no problem to correct as what was done here; however, if the blank NO2 was coming from absorption of atmospheric NO2 during the preparation and the installation of the coated denuder tubes, it won't makes sense to do the correction by assuming blank D17O(NO2) = 0 permil, which would over correct.

2) Equation (4) and relatives in the text: don't get the point why defines a new term $T_{NO+O3}$ instead of using what has been long used in the literature, i.e., the A value to represent the fraction of O3 oxidation of NO and this "A" is almost reserved in the study of D17O of nitrate. I suggest to keep consistent with the literature and don not define new terms unless necessary so that peers can easily follow.  I understand here the authors used "A*" later to define the lifetime differences, but it can be simply replaced with any other symbols.

3) Line 246: These latter, not "later"

4) line 409, I would not call 29.0 +/- 2.2 is consistent with the values of 26 +/- 1 in the literature. In fact, why not just use the relationship of D17O(O3*) = 1.5 D17O(O3)bulk? The latter would derive a lower D17O(O3)bulk which is more consistent with Vicars and Savarino 2014;

5) line 458: it should be $D17O_{NO2+O3}$(NO3) ? Again, if considering terminal O transfer when reacting with O3, why not simply using the relationship of 1.5? The transfer function D17O from O3 to NO and NO2 are different?

---

## Author Comment (AC1)

**Response to the referee #1 (egusphere-2023-744)**

Dear Lei Geng,

We thank you for your valuable review of this work and relevant comments that greatly improved the manuscript. Your comments/suggestions are given below in bold, followed by our answers. Changes/new elements added to the manuscript are presented in blue.

**Major comments:**

1) **The way to derive $RO_2$ using measured $\Delta^{17}O(NO_2)$: I don't think it make any sense to use the other method (i.e., Case B) to estimate $RO_2$ and compare the results with Case A.**

**First of all, as shown in Table 1, in both SP1 and SP2, neither the derived $RO_2$ concentra4onat each sampling period nor the averages can be considered as "consistent"; Don't get the points how can these values can be called "closeness".**

**Secondly, to derived $RO_2$ from $\Delta^{17}O(NO_2)$ (i.e., Case A), one simply assumes that it is only $RO_2$ completes with $O_3$ to oxidize NO as indicated by Equation 13. This ignores the contributions of $HO_2$. While in Case B, a $RO_2 /HO_2$ ratio of 0.859 was applied. So does this mean that in Case A the same ratio of $RO_2/HO_2$ can also be applied and then compared with Case B? In fact, it is highly doubtable that the same ratio in one study can be applied to another, given variations in concentrations of CO, $CH_4$ and NMVOCs, don't even to mention the uncertainties associated with the empirical formula of $HO_2$ calculation using $O_3$.**

**So just saying from $\Delta^{17}O(NO_2)$ to estimate $RO_2$ and note it is actually representing the sum of $RO_2$ and $HO_2$ is enough here. This provides a new method to estimate $RO_2$ and $HO_2$ radicals, and can be verified with actual measurements in the future. The current way of Case B is just too uncertain and the results are not comparable.**

Thank you for pointing this out, which was also a concern for Reviewer #2. We agree with your comments and have removed the comparison with Case B in the revised version of the manuscript which now only discusses $RO_2$ derived from isotopic measurements. We invite you to read our answers to Reviewer #2 pages 26-27 and pages 30-32, where you will find the revised manuscript structure and the reworded/modified section 3.2 on isotopically derived $RO_2$.

**The authors paid too much attentions to the observations of Beijing and tried to make analogies between the two in order to explain their observations. However, the authors overlooked the differences in concentrations of $PM_{2.5}$, $PM_{10}$ and nitrate between the two days, and all of these concentrations in SP 2 are higher in SP 2 than SP 1, would this suggest a transport events and non-local sources of nitrate? Note the $\delta^{15}N(NO_3^-)$ values and its relationship with $\delta^{15}N(NO_2)$ are also different in SP 2 compared to SP 1, this may indicate the same thing: a regional transport event occurred and which brought non-local nitrate. The $\delta^{15}N$ and $\Delta^{17}O$ data should be combined to discuss at this point.**

We were not sought to draw analogies but rather to show how an interpretation of $\Delta^{17}O(NO_3^-)$ variabilities without taking into account of the stratification of $\Delta^{17}O(NO_2)$ at night in polluted environments could lead to biased conclusions. We believe it is important to alert the community on such over-interpretations, which could arise from poor considerations of $\Delta^{17}O(NO_2)$ dynamics.

**The last, references or other evidence should be provided when attributed the high load of $PM_{10}$ in SP 2 to Saharan dust, e.g., back trajectory analysis or something similar.**

The Saharan dust episode which began around 23 February was indeed more diffuse that that of 7 February. NASA AQUA MODIS satellite images show the formation of a dusty air mass over the Saharan region on February 20 and the displacement of this air mass over the following days towards the south of France (**Erreur ! Source du renvoi introuvable.**). In addition, backtracking trajectory (HYSPLIT) shows the air mass from the south entering the Chamonix valley on 24 February (**Figure 1**). Figure 1 and 2 have been added to the revised Supplement.

To note, we had visual evidence of the presence of Saharan dust in the Chamonix atmosphere during SP 2 and the extractions from the GF filters collected during SP 2 showed a yellow coloration, very distinct from the filter extractions from SP 1.

[Figure]

**Figure 1.** Images from the AQUA satellite (MODIS; © NASA) from 19 February 2021 to 25 February 2021. One can see a plume of Saharan dust forming over North Africa on 20 February (localised by the red circle) and moving above the south of France and the Alps until 24 February. The yellow pin points to Chamonix, France.

[Figure]

**Figure 1.** HYSPLIT 72 hours backward trajectory on 24 February 2021 ending at Chamonix, France, at 12:00 UTC (13:00 local time). The model was run every 6 hours. The starting height (in meter above ground level) is half of the boundary layer height estimated by the model from meteorological data set.

**Technical comments:**

- **Blank correction of isotope measurement of $NO_2$: I wondered where the blank comes from, if it is coming from the chemicals used to trap $NO_2$, there is no problem to correct as what was done here; however, if the blank $NO_2$ was coming from absorption of atmospheric $NO_2$ during the preparation and the installation of the coated denuder tubes, it won't make sense to do the correction by assuming blank $\Delta^{17}O(NO_2) = 0$ permil, which would over correct.**

Although difficult to assess, it is unlikely that this blank came from atmospheric $NO_2$ as the denuders of the batches used for SP 1 and SP 2 were handled using the same procedure. It is more likely that this contamination came from the batch of MQ water used to extract the SP 2 denuders or from a contaminated pipette. Since the sample collected between 13:30 and 16:30 LT during SP 2 captured less atmospheric $NO_2$ than other samples (due to lower ambient $NO_2$ during this period), the blank has more impact.

- **Equation (4) and relatives in the text: don't get the point why defines a new term $T_{NO+O_3}$ instead of using what has been long used in the literature, i.e., the A value to represent the fraction of $O_3$ oxidation of NO and this "A" is almost reserved in the study of $\Delta^{17}O$ of nitrate. I suggest to keep consistent with the literature and do not define new terms unless necessary so that peers can easily follow. I understand here the authors used "A*" later to define the lifetime differences, but it can be simply replaced with any other symbols.**

We understand your concern. However, the "$A*$" notation was initially used in Albertin et al. (2021) to be consistent with the literature (see Li et al., 2020), so it follows that we could not use this letter for the term relating to $T_{NO+O_3}$. Thus, for the sake of consistency with Albertin et al. (2021), we would like to keep the term "$T_{NO+O_3}$". In addition, "$T$" symbolises the word "transfer", a more explicit letter than $A$. Nonetheless, to avoid confusing, we have added in the revised manuscript that this term can also be called $A$ in part of the literature.

- **Line 246: These latter, not "later"**

Thank you for pointing this mistake out, which has been corrected.

- **Line 409, I would not call 29.0 +/- 2.2 is consistent with the values of 26 +/- 1 in the literature. In fact, why not just use the relationship of $\Delta^{17}O(O_3^*) = 1.5 \times \Delta^{17}O(O_3)_{bulk}$? The latter would derive a lower $\Delta^{17}O(O_3)_{bulk}$ which is more consistent with Vicars and Savarino 2014.**

Regarding the point that we state $(29.0 \pm 2.2)$ ‰ to be consistent with the values of $(26.2 \pm 1.3)$ ‰, in view of the uncertainty range, this is acceptable in terms of overlapping. However, although our mean value remains consistent with previous studies, we recognise that it is at the lower end of the SD of 29.0 ‰ and at the upper range of the SD of 26 ‰. To support the consistency of our derived $\Delta^{17}O(O_3)_{bulk}$ at $(29.0 \pm 2.2)$ ‰ with the study of Vicars and Savarino (2014), we propose to add to the revised manuscript a comparison of our derived $\Delta^{17}O(O_3)_{bulk}$ with the measurements of Vicars and Savarino (2014) performed in February-March. Indeed, it is interesting to note that, over year-round measurements of $\Delta^{17}O(O_3)_{bulk}$ in Grenoble, France (120 km south-west of Chamonix), Vicars and Savarino (2014) reports a large peak of $\Delta^{17}O(O_3)_{bulk}$ values in February-March, 2-3 ‰ greater than the annual mean of $(26.2 \pm 1.3)$ ‰. Although the reasons for the higher values at that period remain unknown, our derived $\Delta^{17}O(O_3)_{bulk}$ is surprisingly in very good agreement with the values reported for this period by Vicars and Savarino (2014). We have added this point to the revised manuscript (lines 484-495) as:

"Assuming that their maximum measured daytime $\Delta^{17}O(NO_2)$ reflects the conversion of NO to $NO_2$ only through Reaction (R3) (i.e., $T_{NO+O_3} = 1$), Albertin et al. (2021) derived a $\Delta^{17}O_{NO+O_3}(NO_2)$ value of 39.2 ‰ from Eq. (3). Given the respective analytical uncertainties (around ±1 ‰), their value is in very good agreement with the maximum daytime value of 40.8 ‰ we observed in Chamonix. Similarly to Albertin et al. (2021), assuming that the highest daytime $\Delta^{17}O(NO_2)$ value at our site corresponds to $T_{NO+O_3} \approx 1$ leads to $\Delta^{17}O_{NO+O_3}(NO_2) = 40.8$ ‰. Using the experimental $\Delta^{17}O_{NO+O_3}(NO_2)$ transfer function determined by Savarino et al. (2008), we estimate a bulk $^{17}O$-excess of $O_3$ ($\Delta^{17}O(O_3)_{bulk}$) at $(29.0 \pm 2.2)$ ‰. This value is consistent with the range of direct $\Delta^{17}O(O_3)_{bulk}$ measurements at mid-latitudes (mean of $(26.2 \pm 1.3)$ ‰; Vicars and Savarino, 2014), although falling at the upper end of the range. Interestingly, Vicars and Savarino (2014) reported a significant peak in $\Delta^{17}O(O_3)_{bulk}$ during February-March in Grenoble, France (located 120 km southwest of Chamonix), based on year-round measurements with $\Delta^{17}O(O_3)_{bulk}$ values 2–3 ‰ higher than the annual mean of 26.2 ‰. Although the cause of increased values during this period is unknown, our derived $\Delta^{17}O(O_3)_{bulk}$ matches remarkably well the February-March measurements reported by Vicars and Savarino (2014)."

Now, regarding the relationship of $\Delta^{17}O(O_3^*) = 1.5 \times \Delta^{17}O(O_3)_{bulk}$, a clarification is needed because there is too much confusion on this, even in published literature. $\Delta^{17}O(O_3^*)$ is defined as the $\Delta^{17}O$ bears by the terminal atoms of $O_3$ (we can also find in the literature $\Delta^{17}O(O_3)_{term}$). It does not represent in any way the actual $\Delta^{17}O$ transfer of $O_3$ to the species with which it reacts, in our case NO. The relation $\Delta^{17}O(O_3^*) = 1.5 \times \Delta^{17}O(O_3)_{bulk}$ is theoretically justified as well as observed in laboratory and follows

that the $\Delta^{17}O$ is only borne by the terminal atoms of $O_3$. The $\Delta^{17}O$ transfer from $O_3$ to NO can only be equal to $1.5 \times \Delta^{17}O(O_3)_{bulk}$ if for example NO reacts only with the terminal atoms of $O_3$. Actually, the work of Savarino et al. (2008) did show that NO does not react only with the $O_3$ terminal atoms, but also, albeit to a lower extent, with its central atom (probability of $(8 \pm 5)$ % for the abstraction of central atoms during the reaction NO + $O_3$). As a result, the $\Delta^{17}O$ transfer of $O_3$ to NO is slightly lower than $1.5 \times \Delta^{17}O(O_3)_{bulk}$. Therefore, one need to define the term $\Delta^{17}O_{NO+O_3}(NO_2)$ which represents the true $\Delta^{17}O$ transfer during the reactions NO + $O_3$ and this has been quantified by Savarino et al. (2008) with:

$$\Delta^{17}O_{NO+O_3}(NO_2) = (1.18 \pm 0.07) \times \Delta^{17}O(O_3)_{bulk} + (6.6 \pm 1.5) \text{ ‰}$$

Unfortunately, the literature is very heterogeneous in the choice of the $\Delta^{17}O$ transfer functions, mainly due to confusion in the definition of $\Delta^{17}O(O_3^*)$, which is wrongly associated with the transfer of $\Delta^{17}O$ from $O_3$ to NO. This is why, in the interests of clarification for the community and in order to minimise the uncertainties that can arise from a poor choice of $\Delta^{17}O$ transfer function, we have added few lines on this to our revised manuscript (lines 471-482) as:

"At this point, it is important to recall that the choice of the $\Delta^{17}O_{NO+O_3}(NO_2)$ in Eq. (12) is of a particular importance for quantifying $T_{NO+O_3}$ (as for $RO_2$). In the literature, $\Delta^{17}O_{NO+O_3}(NO_2)$ varies between 35 ‰ and 41 ‰ (Michalski et al., 2003; Savarino et al., 2016; Vicars et al., 2012; Li et al., 2022; Zhang et al., 2022). This relatively wide range of values is partly a result of some confusion in defining $\Delta^{17}O_{NO+O_3}(NO_2)$, the $\Delta^{17}O$ transfer from $O_3$ to NO. Indeed, the term $\Delta^{17}O(O_3^*)$ is sometimes erroneously used as the transfer function of $\Delta^{17}O$ from $O_3$ to $NO_2$ during Reaction (R3). $\Delta^{17}O(O_3^*)$ is actually defined as $\Delta^{17}O$ of $O_3$ terminal atoms and is also named $\Delta^{17}O(O_3)_{term}$ in the literature. As the $\Delta^{17}O$ in $O_3$ is borne by its terminal atoms, $\Delta^{17}O(O_3^*) = 1.5 \times \Delta^{17}O(O_3)_{bulk}$. However, $\Delta^{17}O_{NO+O_3}(NO_2)$ can be equal to $1.5 \times \Delta^{17}O(O_3)_{bulk}$ if only terminal atoms of $O_3$ reacts with NO. But laboratory experiments by Savarino et al. (2008) did show that $O_3$ does react with NO not solely with its terminal atoms but also, to a small but significant extent, with its central atom (probability of $(8 \pm 5)$ % for the abstraction of central atoms during the reaction NO + $O_3$). Consequently, $\Delta^{17}O_{NO+O_3}(NO_2)$ is slightly lower than $1.5 \times \Delta^{17}O(O_3)_{bulk}$ and the $\Delta^{17}O_{NO+O_3}(NO_2)$ expression determined by Savarino et al. (2008) should be used: $\Delta^{17}O_{NO+O_3}(NO_2) = 1.18 \pm 0.07 \times \Delta^{17}O(O_3)_{bulk} + (6.6 \pm 1.5) \text{ ‰.}$"

- **Line 458: it should be $\Delta^{17}O_{NO_2+O_3}(NO_3)$? Again, if considering terminal O transfer when reacting with $O_3$, why not simply using the relationship of 1.5? The transfer function $\Delta^{17}O$ from $O_3$ to NO and $NO_2$ are different?**

Similarly to our comment above, $\Delta^{17}O_{NO_2+O_3}(NO_3)$ can be associated to $1.5 \times \Delta^{17}O(O_3)_{bulk}$ only if $O_3$ reacts solely with $NO_2$ with its terminal atoms. Berhanu et al. (2012) quantified this transfer in the gas phase and found :

$$\Delta^{17}O_{NO_2+O_3}(NO_3)= (1.23 \pm 0.19) \times \Delta^{17}O(O_3)_{bulk} + (9.02 \pm 0.99) \text{ ‰}$$

However, it should be noted that, unlike reaction $NO + O_3$, $O_3$ do indeed reacts with $NO_2$ almost entirely with its terminal atoms (see Berhanu et al., 2012 for more details).

**References**

Albertin, S., Savarino, J., Bekki, S., Barbero, A., and Caillon, N.: Measurement report: Nitrogen isotopes ($\delta^{15}$N) and first quantification of oxygen isotope anomalies ($\Delta^{17}$O, $\delta^{18}$O) in atmospheric nitrogen dioxide, Atmospheric Chemistry and Physics, 21, 10477–10497, https://doi.org/10.5194/acp-21-10477-2021, 2021.

Berhanu, T. A., Savarino, J., Bhattacharya, S. K., and Vicars, W. C.: $^{17}$O excess transfer during the $NO_2$ + $O_3$ → $NO_3$ + $O_2$ reaction, The Journal of Chemical Physics, 136, 044311, https://doi.org/10.1063/1.3666852, 2012.

Li, Y., Shi, G., Chen, Z., Lan, M., Ding, M., Li, Z., and Hastings, M. G.: Significant Latitudinal Gradient of Nitrate Production in the Marine Atmospheric Boundary Layer of the Northern Hemisphere, Geophysical Research Letters, 49, e2022GL100503, https://doi.org/10.1029/2022GL100503, 2022.

Michalski, G., Scott, Z., Kabiling, M., and Thiemens, M. H.: First measurements and modeling of $\Delta^{17}$O in atmospheric nitrate., Geophysical Research Letters, 30, 1870, https://doi.org/10.1029/2003GL017015, 2003.

Savarino, J., Bhattacharya, S. K., Morin, S., Baroni, M., and Doussin, J.-F.: The $NO+O_3$ reaction: A triple oxygen isotope perspective on the reaction dynamics and atmospheric implications for the transfer of the ozone isotope anomaly, J. Chem. Phys., 128, 194303, https://doi.org/10.1063/1.2917581, 2008.

Savarino, J., Vicars, W. C., Legrand, M., Preunkert, S., Jourdain, B., Frey, M. M., Kukui, A., Caillon, N., and Roca, J. G.: Oxygen isotope mass balance of atmospheric nitrate at Dome C, East Antarctica, during the OPALE campaign, Atmos. Chem. Phys., 16, 2016.

Vicars, W. C. and Savarino, J.: Quantitative constraints on the $^{17}$O-excess ($\Delta^{17}$O) signature of surface ozone: Ambient measurements from 50°N to 50°S using the nitrite-coated filter technique, Geochimica et Cosmochimica Acta, 135, 270–287, https://doi.org/10.1016/j.gca.2014.03.023, 2014.

Vicars, W. C., Bhattacharya, S. K., Erbland, J., and Savarino, J.: Measurement of the 17O-excess (Δ17O) of tropospheric ozone using a nitrite-coated filter, Rapid Communications in Mass Spectrometry, 26, 1219–1231, https://doi.org/10.1002/rcm.6218, 2012.

Zhang, Y.-L., Zhang, W., Fan, M.-Y., Li, J., Fang, H., Cao, F., Lin, Y.-C., Wilkins, B. P., Liu, X., Bao, M., Hong, Y., and Michalski, G.: A diurnal story of $\Delta^{17}$O($NO_3^-$) in urban Nanjing and its implication for nitrate aerosol formation, npj Clim Atmos Sci, 5, 1–10, https://doi.org/10.1038/s41612-022-00273-3, 2022.

---

## Author Comment (AC2)

**Response to the referee #2 (egusphere-2023-744)**

Dear anonymous referee,

We thank you for your thorough review of this work and thoughtful comments that greatly improved the readability of the manuscript. Your comments/suggestions are given below in bold, followed by our answers. Changes/new elements added to the manuscript are presented in blue.

**General comments:**

1) **Two new works that should be digested/interrogated by the authors and incorporated into the work here are listed below. These should provide fodder for discussion of seasonal difference as well as additional datasets that may be more useful to compare with than is currently in the manuscript since this is also a mid-latitude site with significant diurnal variability.**

   **Bekker, C., Walters, W.W., Murray, L.T., Hastings, M.G. (2023), Nitrate chemistry in the northeast US part I: nitrogen isotope seasonality tracks nitrate formation chemistry, Atmospheric Chemistry and Physics, 23(7), 4185-420, https://doi.org/10.5194/acp-23-4185-2023.**

   **Kim, H., Walters, W.W., Bekker, C., Murray, L.T., Hastings, M.G. (2023), Nitrate Chemistry in the Northeast US Part II: Oxygen Isotopes Reveal Differences in Particulate and Gas Phase Formation, Atmospheric Chemistry and Physics, 23(7), 4203-4219, https://doi.org/10.5194/acp-23-4203-2023.**

These two publications are now referenced in the manuscript as additional data for comparisons. However, please note that, although these measurements are generally in agreement with our results, these two studies collected nitrate on a weekly basis with gas and particulate phase segregation for nitrate whereas our measurements were performed over 2 days on sub-daily sampling with collection of total nitrate. As a result, the comparison is to some extent limited.

2) **A few aspects related to methodology – evidence needs to be provided that under the conditions in this study that both $HNO_3$ and particle phase nitrate are quantitively collected. The references cited do not actually prove this, but rather suggest that conditions are alkaline enough that this "should" be true. This can have an important impact as there is large fractionations between the gas and particle phase nitrate. This is ignored in the current study, presumably because of the assumption that total nitrate is collected. But if the particles are decoupled from the gas upon collection, or upon transport, then this could indeed be making a difference in the dataset here. This topic should be better addressed in the manuscript (see general comments below tied to line/section numbers). See for instance Geng et al. (2017) treatment of this (this work is already cited in manuscript) and Li et al. (2020) (citation below).**

Regardless of the type of filter substrate, determining the exact nature of the nitrate collected on filter samples has always been a matter of debate, and no medium can completely prevent sampling artefacts

(e.g., Schaap et al., 2002; Appel et al., 1980). For instance, nitric acid ($HNO_3$) can potentially be absorbed on aerosols that have already deposited on the filter material. In addition, $HNO_3$ can evaporate from the filter after exposure. We recognise that the references cited do not prove that glass fibre filters effectively collect both $HNO_3$ and particulate nitrate ($p$-$NO_3^-$), but only assume that the ambient conditions (i.e., marine boundary layer and therefore the presence of sea salts) allow effective collection of $HNO_3$, as has already been discussed in the literature (e.g., Prospero and Savoie, 1989). In our case, and we agree that this should be mentioned more explicitly, since the ambient air in Chamonix is likely free of sea salts, we did not pre-wash the glass fibre filters before using them. In fact, they are already highly loaded with NaCl due to their manufacturing process. Therefore, we assume that $HNO_3$ is collected quantitatively on the glass fibre filters due to their NaCl coating, which has been shown in the literature to collect $HNO_3$ (Appel et al., 1981). If you with, we can share the IC analyses of virgin filters, which prove the high NaCl loading.

However, although we cannot cite any published study that proves the efficiency of $HNO_3$ collection using GF filters, we would like to share here our own investigation into this topic which provides some support. Figure 1 below shows a comparison of nitrate concentrations obtained from collections on GF filters using two high-volume samplers installed in parallel at the Concordia station on the Antarctic plateau. One of the samplers was equipped with GF filters without any pre-treatment, while the second used NaCl-coated GF filters. It can be seen that there is almost no difference between the two type of filters. Furthermore, at Concordia, nitrates are mostly present under $HNO_3$ in summer, reaching around 100 ng m$^{-3}$, which is 10 times higher than the $p$-$NO_3^-$ concentration (Legrand et al., 2017). As our nitrate concentrations also reach around 100 ng m$^{-3}$ in summer, this means that we are necessarily effectively collecting $HNO_3$. In addition, Figure 2 shows a comparison of nitrate collection at Concordia with two different filter types, GF and cotton (Whatman® 41). Clearly, it appears again that the GF collected more than 100 ng m$^{-3}$ of nitrate in summer, in agreement with Legrand et al. (2017), whereas cotton filters collected much less nitrate, probably corresponding to the particulate phase only. Finally, Figure 3 shows a comparison of nitrate collections at Concordia with coated denuder tubes (collecting only the gas phase), Teflon, and GF filters. Please note that the sampling steps are not the same between the different sampling media. However, it is clearly visible that the $HNO_3$ concentrations collected by the denuders are in good agreement with the GF collections, while the Teflon filters show much lower nitrate concentrations, suggesting that only the particulate phase was collected.

[Figure]

**Figure 1.** Comparison of filter collection efficiency for atmospheric nitrate sampling at Concordia, Antarctica. The black triangles represent concentration of nitrate collected with glass fiber filters without any pre-treatments and the red squares represent concentration of nitrate collected with glass fibre filter coated with NaCl. Data courtesy : Joël Savarino, unpublished data.

[Figure]

**Figure 2.** Comparison of filter collection efficiency for atmospheric nitrate sampling at Concordia, Antarctica. The blue triangles represent concentration of nitrate in ng m$^{-3}$ collected with glass fiber filters without any pre-treatments and the orange squares represent concentration of nitrate collected with cotton filter. Data courtesy : Joël Savarino, unpublished data.

[Figure]

**Figure 3.** Comparison of filter collection efficiency for atmospheric nitrate sampling at Concordia, Antarctica. The blue triangles represent concentration of nitrate collected with glass fiber filters without any pre-treatments, the grey triangles represent concentration of nitrate collected with coated denuders, and the yellow crosses represent concentration of nitrate collected with Teflon filter. Data courtesy : Joël Savarino and Michel Legrand, unpublished data.

We have added a few lines in the revised manuscript to better explain our approach (lines 184-193):

"Evaluating the collection efficiency of total $NO_3^-$ has long been debated (e.g., Schaap et al., 2002; Appel et al., 1980) and, although not free from sampling artefacts (e.g., potential volatilisation of $HNO_3$ after exposure to ambient air), GF filters have been used on several times to study nitrate isotopes, mainly in coastal sites (e.g., Savarino et al., 2007; Michalski et al., 2003; Morin et al., 2009; Frey et al., 2009; Morin et al., 2007a; Patris et al., 2007; Vicars et al., 2013). Under these conditions, the aerosol alkalinity is supposed to allow the collection of $HNO_3$ (Prospero and Savoie, 1989). In our case, as the ambient air in Chamonix is expected to be free of sea salt, the GF filters were not washed before use in order to keep the initial NaCl coating inherited from the filter manufacturing process. Therefore, in addition to $p$-$NO_3^-$, we are confident that the high GF filter NaCl loading allowed the quantitative collection of $HNO_3$ at our site, as it has been shown previously in the literature (Appel et al., 1981; see also our reply and data to the comments of Reviewer #2 on this specific issue)."

With regard to your concern on the distribution of $NO_3^-$ between the gaseous phase ($HNO_3$) and/or aerosols phase ($p$-$NO_3^-$) during transport, the studies you cite were carried out in remote regions (Greenland and the Tibetan Plateau) and are based on $NO_3^-$ analysed in ice cores. The problematic is therefore different, in particular because our sampling site is much closer to the emission sources than these two sites. Numerous studies have shown the dominant influence of local pollution in our sampling area (e.g., Chazette et al., 2005; Quimbayo-Duarte et al., 2021; Allard, 2018; Chevrier, 2016). In consequence, most of the $NO_3^-$ at our site is most likely formed locally, i.e., in the vicinity of Chamonix and/or in a surrounding area of a few tens of kilometres. The residence time of $NO_3^-$ in the atmosphere before being sampled at our site is much shorter than that analysed in Greenland and Tibetan Plateau ice cores, and we can therefore expect post-$NO_3^-$ portioning to be negligible, all the more so if we

consider our sampling steps at high temporal resolution (typically 3 h). However, although probably minor in winter due to high local $NO_x$ emissions and poor dispersion due to the low height of the surface boundary layer and surrounding high-altitude mountains, we acknowledge the possibility of an influence of $NO_3^-$ input through long range transport, as it could be the case during SP 2 due to Saharan dust entering the Chamonix valley. In such circumstances, we agree that the distribution between the gas and the particulate phase may be affected due to the promotion of $HNO_3$ dry scavenging by dust and/or heterogeneous processes at the aerosol surface, both of which may have an impact on the isotopic distribution of $NO_3^-$ collected at the surface. This influence of a change in $NO_3^-$ origins on its O and N isotopic composition during SP 2 is discussed in the manuscript.

3) **Also, how is it ensured that there is no exchange of water with the analyte nitrite (ie $NO_2$)? And are the oxygen isotopes scaled to the nitrate reference materials or only the nitrite reference materials? Exchange is relatively fast for nitrite in solution (even in frozen solutions) and has been the subject of corrections and difficulties in several studies (see for instance Casciotti and McIlvin, 2007; Wankel et al. 2009).**

**Casciotti, K. and M. McIlvin (2007), Isotopic analyses of nitrate and nitrite from reference mixtures and application to Eastern Tropical North Pacific waters, Marine Chemistry, 107(2), 184-201, https://doi.org/10.1016/j.marchem.2007.06.021.**

**Li, Z., M.G. Hastings, W.W. Walters, L. Tian, S.C. Clemens, L. Song, L. Shao, Y. Fang (2020), Isotopic evidence that recent agriculture overprints climate variability in nitrogen deposition to the Tibetan Plateau, Environment International, 138, https://doi.org/10.1016/j.envint.2020.105614.**

**Wankel, S.D., et al., (2009) Sources of aerosol nitrate to the Gulf of Aqaba: Evidence from d$^{15}$N and d$^{18}$O of nitrate and trace metal chemistry, Marine Chemistry, doi:10.1016/j.marchem.2009.01.013.**

The integrity of the isotopic composition of $NO_2^-$ has already been extensively tested in the cited reference (Albertin et al., 2021). Furthermore, prior to Albertin et al. (2021), Walters et al. (2018) also validated this analytical procedure for O isotopic analysis of $NO_2^-$. In short, matrix alkalinity limits O isotopic exchanges between $NO_2^-$ and water to some extent. For more details on our study of the stability of the $NO_2^-$ isotopic composition, we invite you to read section 2.2 of Albertin et al. (2021) and its Appendix B which demonstrates the stability of the O isotopic composition of $NO_2^-$ isotopic reference materials over several weeks. Please note that we were also able to control O isotopic exchange by preparing several nitrite standards enriched in $^{17}O$ (unpublished data currently being analysed for intercomparison purposes).

The N and O isotopes of $NO_2^-$ analytes are only scaled relative to the $NO_2^-$ reference materials (see the detailed calibration procedure in Albertin et al., 2021). The isotopic composition of $NO_3^-$ cannot be measured with the azide method because it does not convert $NO_3^-$ to $N_2O$, but only to $NO_2^-$. We chose to carry out a calibration based on $NO_2^-$ standards, assuming conservation of $\Delta^{17}O$. If $NO_3^-$ were used, this would require a reduction step to $NO_2^-$ on activated cadmium, a process that is difficult to reproduce under equivalent conditions and during which we cannot certify the conservation of $\Delta^{17}O$.

**4) The crux of the interpretation and conclusion rely on $NO_2$ having a short lifetime such that production of nitrate from $NO_2$ is controlled by local chemistry. The lifetime of $NO_2$ photolysis is calculated, but the lifetime of $NO_2$ loss (as nitrate) is not. The calculation of $\Delta^{17}O$-nitrate relies on $\Delta^{17}O(NO_2)$. At each time step (3 h interval) the nitrate is calculated from the $NO_2$. A weighted average of the $\Delta^{17}O$-$NO_3^-$ values is then taken and compared to the daytime average after the first 3 h interval. This 3 h offset is suggested to take care of the time to convert $NO_2$ to nitrate. The weighting of each 3 h interval for $NO_3^-$ is based on $[NO_2]*J_{NO_2}$ as a proxy for OH production. These values also represent a time when $NO_2$ is being photolyzed, not converted by OH to nitrate. When the nitrate $\Delta^{17}O$ calculated values match with the observations it is taken as proof that the short lifetime is robust. When they do not agree, there has to be another (non-local) source of nitrate. But the authors started with the assumption that at each time interval the $\Delta^{17}O$-$NO_2$ is relevant to the $\Delta^{17}O$-$NO_3^-$ collected. This all ends up a bit messy (albeit complex). It seems that it might make more sense to calculate an average (weighted) daytime $\Delta^{17}O$-$NO_2$ and then calculate an average $\Delta^{17}O$-$NO_3^-$ from this weighted $NO_2$. There is not a direct link between the simultaneously collected $NO_2$ and $NO_3^-$ - in other words, the $NO_3^-$ collected in that time interval cannot be from the $NO_2$ that was also collected in that time interval. So really what can be tested is whether there is diurnal variability in the signals and then for the comparison purposes use the average $\Delta^{17}O$-$NO_2$ to predict average $NO_3^-$ and see how that compares with the observations. If the authors disagree, then the approach they are currently taking needs to be better justified.**

The time lag was intended to take account of the time required to convert $NO_2$ in to $HNO_3$. We agree that this 3 hr shift seems somewhat arbitrary. Following your comment, we have revised our methodology. In the revised version of the manuscript, we average the daytime observations of $\Delta^{17}O(NO_2)$ between 7:30 and 18:00 LT, which are now compared with observed $\Delta^{17}O(NO_3^-)$ averaged over the same period. At night, we average observed $\Delta^{17}O(NO_2)$ and $\Delta^{17}O(NO_3^-)$ between 18:00 and 7:30 LT. Although the mean values are different compared with the initial version of the manuscript, the general conclusions do not change because the difference between calculated and observed $\Delta^{17}O(NO_3^-)$ is still greater during SP 2 than during SP 1. We have also modified the average period for $\delta^{15}(NO_2)$ and $\delta^{15}(NO_3^-)$ and in a same way, despite the changes in the average values, the general conclusions are similar to the initial version, showing a distinctive behaviour of the N isotopes of $NO_3^-$

during SP 2. As a result, Table 2 and 4 have been updated in the revised manuscript with the new values as:

| | | $\Delta^{17}O(NO_2)$ /‰ | $\Delta^{17}O(NO_3^-)$ /‰ | $\Delta^{17}O_{calc}(NO_3^-)$ /‰ | $\Delta^{17}(NO_3^-{}_{calc} - NO_3^-{}_{obs})$ |
|---|---|---|---|---|---|
| Daytime | SP 1 | $30.0 \pm 7.3$ | $23.0 \pm 3.1$ | $22.5 \pm 4.6$ | $-0.5$ |
| (7:30-18:00) | SP 2 | $26.1 \pm 6.9$ | $23.9 \pm 3.8$ | $17.5 \pm 4.6$ | $-6.4$ |
| Nighttime | SP 1 | $21.2 \pm 1.1$ | $20.5 \pm 1.1$ | $21.4 \pm 0.7$ | $0.9$ |
| (18:00-7:30) | SP 2 | $20.8 \pm 1.0$ | $23.2 \pm 1.0$ | $21.0 \pm 0.6$ | $-2.2$ |

**Table 2.** Mean observed $\Delta^{17}O$ data of NO$_2$ ($\Delta^{17}O(NO_2)$) and NO$_3^-$ ($\Delta^{17}O(NO_3^-)$) in Chamonix, and mean calculated $\Delta^{17}O$ of NO$_3^-$ ($\Delta^{17}O_{calc}(NO_3^-)$) using Eqs. (6) and (7) at day and night, respectively, constrained with observed $\Delta^{17}O(NO_2)$. Day and night calculated values were weighted by $[NO_2] \times J_{NO_2}$ and $[NO_2] \times [O_3]$, respectively

| | | $\delta^{15}N(NO_2)$ /‰ | $\delta^{15}N(NO_3^-)$ /‰ | $\delta^{15}N(NO_x)$ /‰ | $\delta^{15}N(NO_3^- - NO_2)$ /‰ |
|---|---|---|---|---|---|
| Daytime | SP 1 | $7.4 \pm 4.7$ | $7.0 \pm 6.7$ | $-9.9 \pm 2.9$ | $-0.4$ |
| (7:30-18:00) | SP 2 | $14.0 \pm 13.9$ | $4.0 \pm 6.4$ | $-10.8 \pm 2.1$ | $-10.0$ |
| Nighttime | SP 1 | $-5.1 \pm 2.3$ | $1.4 \pm 1.2$ | $-9.0 \pm 0.8$ | $6.3$ |
| (18:00-7:30) | SP 2 | $-2.5 \pm 4.2$ | $-1.1 \pm 0.4$ | $-9.9 \pm 1.9$ | $1.4$ |

**Table 4.** Mean observed $\delta^{15}N$ data of NO$_2$ ($\delta^{15}N(NO_2)$) and NO$_3^-$ ($\delta^{15}N(NO_3^-)$), calculated atmospheric $\delta^{15}N$ of NO$_x$ ($\delta^{15}N(NO_x)$, and $\delta^{15}N$ shift between $\delta^{15}N(NO_3^-)$ and $\delta^{15}N(NO_2)$ ($\Delta^{15}(NO_3^- - NO_2)$).

These new values has been inserted in the revised manuscript in sections describing these data as (lines 518-560):

[revised manuscript text omitted]

In hindsight, a key aspect to consider when comparing calculated and observed $\Delta^{17}O(NO_3^-)$ is not actually the timescale for conversion of $NO_2$ into $NO_3^-$, but rather the residence time of $NO_3^-$, which is determined by deposition. Indeed, if $NO_3^-$ residence time was significantly shorter than the averaging period of $\Delta^{17}O(NO_2)$ and $\Delta^{17}O(NO_3^-)$, most of the $NO_3^-$ sampled during a specific averaging period would originate from $NO_2$ oxidised during the same period. Reversely, if $NO_3^-$ residence time was much longer than the averaging period, most of $NO_3^-$ sampled during a specific averaging period would come from $NO_2$ from previous averaging periods. Here, $NO_3^-$ residence time against dry deposition is estimated to reach a maximum of 28 h when the boundary layer (BL) is at 500 m, and 6 h when the BL is at 100 m. These two altitudes are considered respectively as the maximum daytime and minimum nighttime altitudes of the BL at our sampling site. Therefore, most of nocturnal $NO_3^-$ is expected to reflect nocturnal oxidation of $NO_2$, especially towards the end of the night. During the day, $NO_3^-$ is more likely to reflect both daily oxidation of $NO_2$ and nocturnal $NO_3^-$ because residues of nocturnal $NO_3^-$ present after sunrise should persist longer during the day than during the night.

5) **The presence and photolysis of HONO is used in the discussion of the oxygen isotopes to infer a major source of $RO_2$ radicals in wintertime. HONO, however, is not brought at all into the interpretation of the $\delta^{15}N$. Both the $\delta^{15}N$ and $\delta^{18}O$ are impacted by the oxidation processes because of the fractionations detailed by the authors. So an explanation for one isotope necessarily impacts the interpretation of the other when that species can also be source of N.**

Although the photolysis of HONO can indeed be considered as a source of OH radicals at our site at this time of the year, given the high flux of direct $NO_x$ emissions from anthropogenic sources, it is unlikely that the production of NO by HONO photolysis would have a significant impact on the $\delta^{15}N$ of $NO_2$ and $NO_3^-$. First, HONO is generally below or in the order of ppb (e.g., Michoud et al., 2014), which is important for OH production in winter but is negligible compared to $NO_x$ levels of a few tens of ppb observed at our site. Secondly, HONO production has been estimated during field campaigns at different sites, for instance 3 ppb h$^{-1}$ in Tai'an (Xue et al., 2022), 2.6 ppb h$^{-1}$ in Beijing (Spataro et al., 2013), 0.7 ppb h$^{-1}$ in Paris (Michoud et al., 2014), and 1.77 ppb h$^{-1}$ in Santiago (Elshorbany et al., 2009). All these references are cited in Ye et al. (2023). Based on these estimates, the HONO production rate cannot compete with the $NO$-$NO_2$ conversion/recycling rates in our case. Thirdly, it is clear from data described in Section 3.4.1 that N isotopic equilibrium effects between NO and $NO_2$ drive the main variability of $\delta^{15}N$ at our site.

**Major comments:**

*Abstract*

- **Lines 17, 23-24: add n= # of samples when reporting on the deltas measured**

Thank you for pointing this out, this information has been added to the revised manuscript. Besides, the abstract has been slightly reworded to ease the reading as:

"The oxygen ($\Delta^{17}O$) and nitrogen ($\delta^{15}N$) isotopic compositions of atmospheric nitrate ($NO_3^-$) are widely used as tracers of its formation pathways, precursor (nitrogen oxides ($NO_x$) $\equiv$ nitric oxide (NO) + nitrogen dioxide ($NO_2$)) emission sources, and physico-chemical processing. However, the lack of observations on the multi-isotopic composition of $NO_2$ maintains significant uncertainties regarding the quantitative links between the isotopic composition of $NO_x$ and $NO_3^-$, which ultimately may bias inferences on $NO_3^-$ formation processes and distribution of sources, particularly in winter urban atmospheres. We report here on the first simultaneous atmospheric observations of $\Delta^{17}O$ and $\delta^{15}N$ in $NO_2$ (n = 16) and $NO_3^-$ (n = 14). The measurements were carried out at sub-daily ($\sim$ 3 h) resolution over two non-consecutive days in an Alpine city in February 2021. A strong diurnal signal is observed in both $NO_2$ and $NO_3^-$ multi-isotopic composition. $\Delta^{17}O$ of $NO_2$ and $NO_3^-$ ranges from 19.6 ‰ to 40.8 ‰ and from 18.3 ‰ to 28.1 ‰, respectively. During the day and night, the variability of $\Delta^{17}O(NO_2)$ is mainly driven by the oxidation of NO by ozone, with a substantial contribution from peroxy radicals in the morning. $NO_3^-$ mass balance equations, constrained by observed $\Delta^{17}O(NO_2)$, suggest that during the first day of sampling, most of $NO_3^-$ was formed locally from the oxidation of $NO_2$ by hydroxyl radicals by day, and via heterogeneous hydrolysis of dinitrogen pentoxide at night. For the second day, calculated and observed $\Delta^{17}O(NO_3^-)$ do not match, particularly daytime values; the possible effects on $\Delta^{17}O(NO_3^-)$ of a Saharan dust event that occurred during this sampling period and of winter boundary layer dynamics are discussed. $\delta^{15}N$ of $NO_2$ and $NO_3^-$ ranges from −10.0 ‰ to 19.7 ‰ and from −4.2 ‰ to 14.9 ‰, respectively. Consistent with theoretical predictions of N isotope fractionation, the observed variability of $\delta^{15}N(NO_2)$ is explained by significant post-emission equilibrium N fractionation. After accounting for this effect, vehicle exhaust is found to be the primary source of $NO_x$ emissions at the sampling site. $\delta^{15}N(NO_3^-)$ is closely linked to $\delta^{15}N(NO_2)$ variability, bringing further support to relatively fast and local $NO_x$ processing. Uncertainties on current N fractionation factors during $NO_2$ to $NO_3^-$ conversion are underlined. Overall, this detailed investigation highlights the potential and necessity to use simultaneously $\Delta^{17}O$ and $\delta^{15}N$ in $NO_2$ and $NO_3^-$ in order to better constrain quantitative inferences on the sources and formation chemistry of $NO_3^-$ in urban environments in winter."

- **Lines 16 and 24-25: the meaning of "important diurnal variabilities is unclear, please rephrase to be quantitative or use a better description than just "important"**

We agree that the terms were not appropriate. In the revised manuscript, line 16 has been rephrased with "A strong diurnal signal is observed …" and line 24-35 with " … the observed variability of …" (see our response above for the full abstract).

- **Line 19: rephrase "NO$_3^-$ local mass balance equations"**

The term "local" has been removed in the revised manuscript.

- **Line 30: "particularly in urban environments in winter" – measurements represent winter only**

The term "particularly" has been removed in the revised manuscript.

*Introduction*:

**The introduction well covers most of the important literature on the topic, but is really too expansive for the purposes of this case study. The introduction should focus on atmospheric chemistry in urban regions under the conditions in this study. It would also be useful to refer directly to the reactions as they are being described to keep readers following the text and the series of reactions. All of the chemistry presented as "the important reactions" are based on global studies and may not be representative of chemistry in urban areas. It should also be noted that only one of the reactions includes a phase (e.g., HNO$_3$(g)) and this should be made more consistent throughout the reactions. Furthermore, the titration of ozone should be explicitly discussed in the introduction as this is a major aspect of the nighttime chemistry in regions where there are continuous, fresh NO emissions at night (i.e., urban areas).**

The introduction has been modified/reworded in the revised manuscript accordingly as (line 33-171):

[revised manuscript text omitted]

- **Line 60: be more specific quantitatively about what "the lowest temperatures" means**

The sentence has been rephrased in the revised manuscript as (lines 83-84):

"Reaction (R7) is temperature dependent, so $N_2O_5$ can eventually decompose to reform $NO_2$ and $NO_3$, with the $N_2O_5/NO_3$ ratio being negatively correlated with temperature."

- **Line 75: "…$NO_3^-$ is usually investigated in light of its $^{17}O$-excess" – there are many studies that utilize $\delta^{18}O$ AND $\delta^{17}O$ because there are uniquely enriched in ozone, in addition to a focus on $\Delta^{17}O$. It is important to also incorporate the two studies mentioned above, one of which includes $\Delta^{17}O$ and $\delta^{18}O$ observations in both urban and suburban areas of the US. Please rephrase this line.**

We agree that the use of $\delta^{18}O$ in complement to $\delta^{17}O$ can be very valuable. However, as we do not use $\delta^{18}O$ in the manuscript, we prefer not to mention $\delta^{18}O$ for clarity and to avoid confusing the reader. However, we have added the measurement data of $\delta^{18}O$ in the revised Supplement as they are used to calculate $\Delta^{17}O$.

We have added the study by Kim et al. (2023) in the revised manuscript.

- **Line 87-88: The mention of ice cores and the lack of definition of AOC here is strange. This is not connected at all to the current study or the conclusions that can be drawn from the 2 days of measurements in France. Suggest removing this line and any discussion of ice core related data/conclusions.**

We agree and the line has been removed in the revised manuscript.

- **Line 95-104: This section should include an overview of the work of Albertin et al, 2021. The fact that lower values were found at night challenges previous understanding based on Morin et al (2011). This is not discussed in the current work, but should be presented here or summarized based on what appears in Albertin et al, 2021 to better set up the expectations for diurnal variability in the urban sampling site.**

The introduction has been modified/reworded in the revised manuscript accordingly (see the revised introduction in our response above).

With regard to the of Morin et al. (2011) study, we would like to stress that our results do not "challenge" Morin's model. They are perfectly in line with the transfer model. Simply, Morin's model inputs did not include any fresh $NO_x$ resupply in the box model.

- **Line 109-110: This line does not include the fact that nitrate is also highly susceptible to large fractionations between the gas and particle phases. Why is that not included here and never discussion in the discussion? (see General comment and citations above)**

We do not discuss the N fractionation associated to the speciation of $NO_3^-$ between the gas and particle phase because our sampling method is design to collect both $HNO_3$ and $p$-$NO_3^-$ (see our response to your comment above) and it is the total nitrate that is the right metric to compare with $NO_2$. $HNO_3$ and $p$-$NO_3^-$ are in permanent exchange between these two phases, (e.g., $H_2SO_4 + p$-$NO_3^- \rightarrow p$-$HSO_4^- + HNO_3$ is a well-known acid displacement reaction) and it is even possible that $^{15}N$ exchange happens which will add even more complexity. We do not believe that speciation is an added value for our transfer study. The mechanistical and methodological complexity is not worth it. This is why we deliberately ignore this discussion in the paper, as it will only add unnecessary and useless complexity.

Please note that we acknowledge that the significant change in $\delta^{15}(NO_3^-)$ during SP 2 in the presence of Saharan dust could reflect N fractionations related to a change of $NO_3^-$ origin. This could be due to several reasons such as a change in $NO_x$ precursors and/or in the contribution of $NO_3^-$ formation pathways, especially if part of $NO_3^-$ has a non-local origin. Our limited dataset (only two days) precludes drawing general conclusions but, rather highlights the sensitivity of our approach in investigating the sources and formation pathways of $NO_3^-$ in complex environments.

- **The "during transport of $NO_3^-$ in the atmosphere" should be a #3 here as it represents different process/processes than the prior 2 examples.**

We kind of agree. The idea here was to emphasise that the $\delta^{15}N$ of $NO_3^-$ is determined by two main factors, namely the $\delta^{15}N$-fingerpring of the overall $NO_x$ emission sources and the N fractionation effects that can alter the original $\delta^{15}N(NO_x)$ value. However, for the sake of comprehensiveness, we prefer to maintain the mention of the N fractionation effect associated with $NO_3^-$ transport along with the other processes inducing N fractionation.

- **Line 113: remove or rephrase "which seemingly ignore the potential impact of post-emission fractionation" or find better references. The Hastings et al 2009 paper does not seem relevant here (it is based on ice core reconstruction – not at all comparable measurements to those taken here). Altieri et al. 2022 does not simply ignore this.**

We acknowledge that Hastings et al. (2009) is not directly relevant to the focus of our research, so it has been removed in the revised manuscript. Similarly, although we feel it is important to emphasise that N fractionation effects cannot be ignored in urban atmospheres, we agree to remove the citation of Altieri et al. (2022) in the revised manuscript. Nonetheless, Altieri et al. (2022) do ignore N fractionation effects (mentioned only once and the reason why they are not estimated is unclear/poorly explained) and the authors use their measured $\delta^{15}N$ of atmospheric nitrate directly to performed $NO_x$ emission source apportionment. This approach is fundamentally flawed in regions affected by polluted air masses, as it is well addressed by Bekker et al. (2023).

*Methods:*

- **Line 141: Given that the area was partly covered in snow, and Savarino's rich literature on the impact of photolysis on snow nitrate and the release of $NO_x$ from this source, with very depleted isotopic ratios. Why is this snow-sourced $NO_x$ not included in the discussion here?**

In winter in Chamonix, the snow-sourced $NO_x$ will likely be very small due to the short day length, the reduced irradiation time of the snow surface due to high altitude mountains surrounding the sampling site, and a large solar zenith angle. Additionally, it is very likely that the local flux of anthropogenic $NO_x$ emissions far exceeds the potential flux from the snowpack. For these reasons, snow-sourced $NO_x$ is not included in the discussion. Furthermore, as already mentioned above, data presented in Section 3.4.1 clearly show that N isotopic equilibrium effects between NO and $NO_2$ drive the main variability of $\delta^{15}N$ at our site and that there is one dominant source of $NO_x$ at our site from road traffic.

- **Lines 141-145: Evidence must be provided to support the assumption that total nitrate is collected ($HNO_3$ gas and particulate nitrate). The studies cited do not prove this. This is an assumption made, and under the conditions in those studies, the alkaline quality of the aerosols was suggested to scavenge the $HNO_3$ onto the filters. Is this true in the case study here?**

Please refer to our response to your general comment #2.

- **It should be made clear here also what is being collected – i.e. total suspended particles or PM$_{10}$ or ?**

"TSP" means "total suspended particles". The full term has been added in the revised manuscript for clarity as (lines 183-184):

"Atmospheric particles (aerosols) were collected using a high-volume sampler (Digitel®, DH77, total suspended particle inlet, 1 m$^3$ min$^{-1}$)"

- **For the preservation of samples prior to isotopic analysis, how is it ensured that there is not exchange between the nitrite analyte (for NO$_2$) and water? This has been shown to be extremely important and would yield incorrect results since the samples would no longer be representative of the atmosphere (see General comment and citations above)**

Please refer to our response to your general comment #2.

- **Line 155: This first sentence is difficult to understand. What does "corrected by the arithmetic mean" mean? Was it subtracted from all results? It is stated that is represents 8% ± 8% of what? The average total nitrate collected? Or is this the average of the average based on comparison of the blank concentration to each sample's concentration? It would be better to state the average concentration of the blanks here, perhaps compared against the overall average nitrate concentration, and more clearly state how this was "corrected" for to make this explicitly clear.**

We agree that the overall meaning of this sentence was not clear. In the revised manuscript, we have added information on filter blanks as (line 193):

"Two field blanks were performed to evaluate the initial content of trace elements and possible contamination during handling."

 and rephrased line 155 as (lines 211-214 in the revised manuscript):

"Reported to the total filter surface, the NO$_3^-$ contribution from blank filters represented on average $(8 \pm 9)$ % of sampled NO$_3^-$. Atmospheric mass concentrations (expressed in µg m$^{-3}$) were calculated as the ratio of the total ion filter loading (corrected for the blank contribution) to the total volume of air pumped through the filter at STP conditions."

- **How many field blanks were collected? How were they collected? The impact of a high blank on one collection period is discussed several times later in the manuscript so clarity is needed here.**

We suppose you refer here to denuder sampling. This info has been added to the revised manuscript as (lines 229-236):

"The isotopic composition of $NO_3^-$ samples was analysed in triplicate (the mean value of replicate measurements and the associated repeatability are reported in Table S2 in the Supplement). The limited amount of $NO_2$ samples did not allow for replicate measurements. From UV-vis analysis, all $NO_2$ samples presented a negligible blank (< 4 %; mean of 1.7 nmol ml$^{-1}$) except for the sample collected between 13:30 and 16:30 LT during SP 2 which shown a blank around (14.0 ± 1.4) %. Therefore, the measured $\Delta^{17}O$ of this sample was corrected for blank effect assuming that the contaminated $NO_2^-$ possessed a $\Delta^{17}O = 0$ ‰. No correction from this blank effect was applied on the $\delta^{15}N$ measurements of $NO_2$ because the $\delta^{15}N$ fingerprint of the contamination could not be characterised. This uncertainty is propagated in the calculations of section **Erreur ! Source du renvoi introuvable.** and considered in the discussions."

- **Line 177: Table S4 should be Table S3.**

Thanks for pointing out this error, this has been corrected in the revised manuscript.

- **What are the average measurement uncertainties based upon? Repeated measures of reference materials? And how many times were run?**

For each IRMS analysis set, the average measurement uncertainties is estimated based on the standard deviation of the residuals from the linear regression between the measured value of the reference materials and their expected values.

For the $NO_3^-$ samples, as they were analysed in triplicate, i.e. analysed by three independent IRMS runs, we present here the average analytical error calculated on these three IRMS runs. For each IRMS analysis set, $NO_3^-$ reference materials (n = 4) were prepared in different concentrations (n = 5), from 40 nmol to 140 nmol.

As the azide method is specific to $NO_2^-$ ions, possible $NO_3^-$ contamination of denuder extractions does not impact the isotopic analysis of collected atmospheric $NO_2$.

- **Additionally, it should be made clear here in the methods whether both nitrate and nitrite reference materials are necessary to be included in the separate runs for NO2 (as nitrite) and NO3- (as nitrate).**

This section has been for better clarity in the revised manuscript as (lines 236-243):

"Possible isotopic changes resulting from the conversion and analysis process of $NO_3^-$ and $NO_2$ samples were evaluated using international $NO_3^-$ and $NO_2^-$ isotopic reference materials, respectively (Table S3 in the Supplement). Accuracy of the analytical method was estimated as the standard deviation ($\sigma$) of the residuals between measurements of the reference materials and their expected values. In our study, average measurement uncertainties on $\delta^{15}N$, $\delta^{17}O$, $\delta^{18}O$, and $\Delta^{17}O$ were estimated to be $\pm 0.3$ ‰, $\pm 0.9$ ‰, $\pm 1.3$ ‰, and $\pm 0.4$ ‰, respectively, for $NO_3^-$ samples and $\pm 0.3$ ‰, $\pm 0.4$ ‰, $\pm 0.9$ ‰, and $\pm 0.3$ ‰, respectively, for $NO_2$ samples. Detailed information about the calibration procedure can be found in Morin et al. (2009) and in Albertin et al. (2021) for $NO_3^-$ and $NO_2$ samples, respectively."

- **Line 183-184: Please clarify the "blank" here. Is this the field blank? Above it is stated that the blanks represented 8+/-9% so how is the mean here now 4%? Or is the previous blank about nitrate instead of nitrite? Were nitrate blanks also tested and what were the results?**

The $(8 \pm 9)$ % field blank is relative to $NO_3^-$ samples, not $NO_2$. We have rearranged and reworded Section 2.2 in the revised manuscript for more clarity (see our response above).

- **Section 2.4.1: Please make it clear that $\Delta^{17}O$ is only conserved in processes that are fractionating…i.e., this would not be true if exchange were to occur.**

This has been specified in the revised manuscript as (lines 261-264):

"Because $NO_2$ and $NO_3^-$ loss processes do not fractionate in terms of the oxygen mass-independent anomaly and considering that each source reaction induces a transfer of $\Delta^{17}O$ to $NO_2$ and $NO_3^-$, one consider the mass conservation of $\Delta^{17}O$ during fractionation processes in the $N_r$ cycle. Hence, one can implement $\Delta^{17}O$ in the general mass balance equation of $NO_2$ and $NO_3^-$."

- **Line 224: What is the lifetime for $NO_2$ during the day against loss? Why is this not also calculated? The table states that the lifetime against photolysis is 5 min, what is the lifetime against conversion to nitrate? The case for daytime comparison with a three-hour difference (i.e. $NO_2$ now, compared to $NO_3^-$ that was collected three hours later) needs to**

**be better justified. If the NO$_2$ lifetime is only minutes (Table B1) than the time frames should be the same for comparing NO$_2$ and NO$_3^-$.**

**This is very important. The lifetimes calculated are based on chemistry alone and do not seem to include wind speed or direction. For nighttime, it is stated that NO$_2$ has a 10 hour lifetime against loss (via deposition or chemistry). Therefore, the authors compare a single average nighttime measurement of NO$_2$ to NO$_3^-$. In a closed system this would absolutely make sense. But in an open system, how can we ensure that the NO$_2$ collected during that time represents THE NO$_2$ that was converted to the sampled NO$_3^-$? The authors make the assumption that this is true, then prove it by saying they can calculate the NO$_3^-$ based on the measured $\Delta^{17}$O-NO$_2$ – this seems circular.**

If we consider an OH mixing ratio at ca. $2 \times 10^6$ molec cm$^{-3}$ (average upper limit observed in different polluted environments at mid latitudes in winter; e.g., Tan et al., 2018; Stone et al., 2012) and $k_{NO_2+OH} = 1.2 \times 10^{-11}$ cm$^3$ molec$^{-1}$ s$^{-1}$ (Atkinson et al., 2004), the lifetime for NO$_2$ during the day against chemical conversion into HNO$_3$ (i.e., OH oxidation) would be ca. 12 h. With regard to dry deposition, if we consider a NO$_2$ dry deposition velocity of 0.25 cm s$^{-1}$ (Holland et al., 1999) and a mean boundary layer height if 500 m, the lifetime for NO$_2$ during the day against dry deposition would be at an upper limit of ca. 56 h. Hence, NO$_2$ lifetimes against chemical loss and dry deposition are negligible in comparison with NO$_2$ photolysis which is about few minutes. However, what does really matter to compare $^{17}$O-excess is NO$_3^-$ residence time, and please refer to our response to your general comment #4 on this topic. Note that we have removed line 224 in the revised manuscript and reworded/added information as (lines 290-298):

"Assuming a mean NO$_3^-$ deposition velocity of 0.5 cm s$^{-1}$ (mean value of the dry deposition velocities of HNO$_3$ and p-NO$_3^-$; Zhang et al., 2009), and considering the maximum daytime and minimum nighttime boundary layer heights of 500 and 100 m above ground level, respectively (estimations based on measured vertical temperature profiles; Fig. S2 in the Supplement), the estimated residence time of NO$_3^-$ against dry deposition can reach up to 28 hours during the day, and 6 hours at night (Table B1). Therefore, on sub-daily time scales, the $^{17}$O-excess in NO$_3^-$ during the day is more likely to reflect a combination of daytime and nighttime production processes than during the night. Note that, our estimated residence times for NO$_3^-$ against dry deposition are upper limits as they represent the time required to reduce by a factor e the concentration of NO$_3^-$ present at the top of the boundary layer; NO$_3^-$ close to the surface would have a much shorter residence time."

As the city of Chamonix is at an altitude of 1035 m and is located in an Alpine valley of ca. 2 km wide on average, surrounded by high-elevation mountains (up to 4810 m), we can consider that at night the valley floor is isolated from upper layers due to a stable temperature inversion layer characterised in previous studies (e.g., Chazette et al., 2005). Hence, we can assume in a first approach that at night the surface layer behaves as a close system due to weak vertical mixing, until the inversion layer breaks

down in the following daytime hours. Secondly, our approach is not to assert that the $NO_2$ present at the surface is that which is converted to $NO_3^-$ (which we also sample), but rather to make this assumption and test whether isotopic mass balances support it. If isotopic mass balances do not match the observations, this means that the sampled $NO_3^-$ does not originate solely from local oxidation of surface $NO_2$, as it is the case during SP 2.

- **Section 2.4.2 – I suggest the authors consider moving up the discussion of the Freyer, 1971 study in discussing the framework for interpretation of the isotopes. This is the closest relevant study to the current work for nitrogen isotopes.**

  **Why is the fractionation of $HNO_3(g)$ versus $NO_3^-(p)$ ignored in this section? It should clearly be stated why this is not included. While the collection of total $NO_3^-$ could mask this fractionation, this would only be the case if all of the nitrate gas and particles are locally derived. In other words, if gaseous nitrate is taken up on particles (e.g., see lines 326-330) transported to the observation site and collected as a sample, it will reflect fractionation of $HNO_3(g)$ versus $NO_3^-$ (p) that may have not occurred locally. How is this accounted for? This seems like it could be particularly important in the SP 2 case.**

Following your comment, we have added discussion on this topic in Section 3.4.3 of the revised manuscript. However, we do not discuss gas-particle N fractionation effects in Section 2.4.2 as we consider that both $HNO_3(g)$ and $p$-$NO_3^-$ are collected (see our reply above). For clarity, we prefer to introduce gas-particle N fractionation effects only when we underline changes in the $^{15}N$ distribution between SP 1 and SP 2 in Section 3.4.3.

- **Line 268: what is the "daytime $NO_2$ chemistry lifetime" used here?**

As $NO_2$ lifetime regarding OH oxidation is negligible in comparison to $NO_2$ photolysis (see our response above), here the daytime $NO_2$ chemistry lifetime refers to $NO_2$ lifetime against photolysis, which is translated in the expression we give of $A_{day}^*$.

- **Section 3.1 – This section is largely discussion, not just results. It is indeed important to characterize the general atmospheric observations but the section includes a lot of "likely" descriptions of how to explain the atmospheric observations, not simply a presentation of the results. The section title should be modified to Results and Discussion or the discussion points should be moved to the appropriate section.**

As recommended, we renamed Section 3 "Results and Discussion" and revised accordingly the titles of the sub-sections. Here is the plan of the revised manuscript:

- **Lines 305-310: As mentioned above, it would be useful to discuss ozone titration in the introduction with a more focused introduction around urban day and night $NO_x$ chemistry. Most of the section 3.1 is really discussion of the results, not results. This should be amended. Is it a hypothesis that $O_3$ titration occurred at 16:00 LT or do you have evidence of this?**

As recommended we have modified/reworded the introduction, it is more specific to urban environments (see our response above) and have merged sections Results and Discussion in a single section (see the revised manuscript structure above).

$O_3$ titration at 16:00 LT is supported by atmospheric observations (Figure 1) showing that $O_3$ rapidly drops at that time following the rapid increase of $NO_x$ concentration.

- **Line 318: "It turns out that a Saharan dust episode began on February 23 (Fig S3 in the Supplement." This is simply stated as fact. What is the evidence for this? In the Fig S3 a more impressive dust event occurs on Feb 7[th] and looks like a short term "event". The time**

**noted on the 23$^{rd}$ does not have a clear start and end and looks more like a large background enhancement. No transport information is provided (e.g. back-trajectories or the like). So how is this event known to be of Saharan dust origin? It is stated that this should lead to increase in coarse materials and high concentrations of alumino-silicates and potassium and calcium -- were these measured on these samples and can be shown as verification?**

Thank you for highlighting this point which was also pointed out by Reviewer #1. We have now provided complementary information and observation in the revised Supplement that back up the dust event idea. We invite you to read our response to comment #2 of Reviewer #1 on this subject.

- **Line 329: "…the origin of NO$_3^-$ during SP 2 at our site remains unclear…" – why is this case when line 318 stated as fact that it was Saharan dust? All of this needs much clarification!**

The presence of Saharan dust during SP 2 does not necessarily mean that there were exogenic nitrate inputs at our sampling site (i.e., not formed in the Chamonix area). So yes, despite the fact that there is no doubt that we had Saharan dust deposits during SP 2, we cannot quantitatively identify the impact of the presence of this dust on the nature of the NO$_3^-$ collected during SP 2. Indeed, Saharan dust can promote both the formation of NO$_3^-$ by heterogeneous hydrolysis and the condensation of HNO$_3$ already present in the atmosphere. The only statement we can make is that the change of O and N isotopic distribution of NO$_3^-$ between SP 1 and SP 2 could be linked to the presence of dust but we cannot state it as a fact.

- **Lines 352-365: Here it is stated that the conditions for Chamonix and Beijing are significantly different. But comparisons are made anyway. So this comes across as convenient. For seasonal considerations, this study is limited to only 2 days of wintertime observations. Suggest you include comparison with Kim et al., ACP, 2023 to suggest patterns of expected seasonal behaviour for mid-latitude conditions.**

We agree that the conditions in Chamonix and Beijing are different. However, this is the only previous study that has carried out high temporal resolution (ca. 3 h) NO$_3^-$ sampling as we have done in Chamonix. It is the only point of comparison. We therefore believe that it is important to mention this study, although any comparison of NO$_3^-$ isotopic records between the two sites must be done with great care (which we have done). We have modified/reworded this section in the revised manuscript, adding an additional comparison with another study that have collected NO$_3^-$ on day/night basis as (lines 497-516):

"Over the two sampling periods, $\Delta^{17}O(NO_3^-)$ varies significantly (from 18.3 ‰ to 28.1 ‰), with a weighted mean of (22.5 ± 3.1) ‰. While $\Delta^{17}O(NO_2)$ values are relatively similar during the two sampling periods, $\Delta^{17}O(NO_3^-)$ values are systematically higher during SP 2 than during SP 1, except during the 7:30−10:00 LT interval. $\Delta^{17}O(NO_3^-)$ in Chamonix is in the same range of most previous observations in urban environments (9−44 ‰; e.g., Kim et al., 2023; Wang et al., 2023; Fan et al., 2023; Zhang et al., 2022b; Lim et al., 2022; Li et al., 2022b), but lower than most values measured during the cold season which are typically >25 ‰. Unlike $\Delta^{17}O(NO_2)$, daytime and nighttime $\Delta^{17}O(NO_3^-)$ values at our site are not significantly different ($p$-value > 0.05, n = 14). Similarly, from 12 h resolved sampling in winter Beijing, He et al. (2018) found no significant difference between daytime and nocturnal $\Delta^{17}O(NO_3^-)$ and suggest that each sample reflects $NO_3^-$ produced during both the day and night. From high-time-resolved (3 h) aerosol sampling in winter Beijing, Zhang et al. (2022b) reported $\Delta^{17}O(NO_3^-)$ values between 23.4 ‰ to 39.3 ‰, with higher values observed at night ((31.0 ± 2.6) ‰) than during the day ((29.3 ± 3.0) ‰). This diurnal behaviour of $\Delta^{17}O(NO_3^-)$ was attributed to the changes in the branching ratio of nocturnal and photochemical reactions on $NO_3^-$ formation. In Chamonix, the range of $\Delta^{17}O(NO_3^-)$ values are very different from Zhang et al. (2022) observations, with consistently lower values and a distinct diurnal tendency. However, in the cases of $\Delta^{17}O(NO_3^-)$ measurements at sub-daily temporal scale, the atmospheric lifetime of $NO_x$ and $NO_3^-$ is critical for comparing $\Delta^{17}O(NO_3^-)$ records from one site to another. Pollutant levels and atmospheric conditions between Chamonix and Beijing are very different, notably in winter when Asian urban areas can experience severe haze pollution episodes with $NO_3^-$ mass concentration exceeding 70 µg m$^{-3}$, which is over 10 times higher than in Chamonix (Lim et al., 2022; He et al., 2018; Zhang et al., 2022b). In such conditions, PM can reach several hundreds of µg m$^{-3}$ for several days, which can significantly impact atmospheric processes involved in the formation of secondary species. Aside from the intrusion of Saharan dust during SP 2, the pollutant level in Chamonix is indicative of a moderately polluted region, with significant diurnal variations"

It is difficult to compare our measurements with Kim et al. (2023). First, $\Delta^{17}O$ have been measured separately for $HNO_3$ and $p$-$NO_3^-$ in Kim et al. (2023), whereas we collected both $HNO_3$ and $p$-$NO_3^-$. Second, Kim et al. (2023) performed weekly samplings whereas we only have 2 days measurements at high temporal resolution (ca. 3 h).

- **Line 372: The expectation that the $\delta^{15}N$ would be similar with Walters et al (2018) does not seem justified. Those measurements were made in a very different setting (ie not a valley surrounded by mountains) and during the summer. Comparing with Freyer et al would make a lot more sense. Commenting on the range and variability from different studies also makes sense, but a direct comparison with Walters does not seem appropriate unless it is justified.**

We do not expect $\delta^{15}N$ of $NO_2$ to be similar to that measured by Walters et al. (2018), nor to that measured by Albertin et al. (2021). This is because, as you mentioned, the environmental conditions of these two studies (spring and summer in urban locations) differ significantly from our study site. The point of comparing our work with these two studies is to highlight the marked change in behaviour of

$\delta^{15}N(NO_2)$ as a function of environmental conditions. This section has been reworded in the revised manuscript to clarify the message we want to convey. In addition, we have added to the revised manuscript the study of Freyer et al. (1993) which showed this seasonal change of N fractionation effects as (lines 615–631):

"Over the two sampling periods, $\delta^{15}N(NO_2)$ shows substantial diurnal variability (from −10.0 ‰ to 19.7 ‰, n = 16) with a weighted mean of (4.0 ± 9.1) ‰. In contrast, Albertin et al. (2021) reported a weak diurnal fluctuation of $\delta^{15}N(NO_2)$ in spring in Grenoble, in a narrow range from about −12 ‰ to −10 ‰. In summer in an urban/suburban location, Walters et al. (2018) also observed a wide range of $\delta^{15}N(NO_2)$ values, however, unlike our study, these are almost consistently negative (from −31.4 ‰ to 0.4 ‰) with an overall mean at (−11.4 ± 6.9) ‰. As shown in Eq.(8), fluctuations in $\delta^{15}N(NO_2)$ reflect changes in $NO_x$ emission sources and/or N fractionation effects, these latter being weighted by $1 - f_{NO_2}$ i.e., the more $NO_x$ is under the form of NO, the greater the N fractionation effects (see Section 2.4). Hence, in the previous works of Albertin et al. (2021) and Walters et al. (2018), due to high $f_{NO_2}$ (> 0.7), isotope effects were small (<2.7 ‰) and $\delta^{15}N(NO_2)$ was mostly driven by changing contribution of $NO_x$ emission sources. At our site, $f_{NO_2}$ shows a wider range, from 0.3 to 1.0, suggesting significant N isotopic fractionation effects, with minimum and maximum contributions corresponding to the highest and lowest observed value of $\delta^{15}N(NO_2)$, respectively. This pronounced seasonal behaviour of N isotope fractionation effects within the $NO_x$ cycle has previously been outlined in the seminal study of Freyer et al. (1993). Overall, compared with summer, lower $f_{NO_2}$ during winter months due to lower $O_3$ concentrations and higher $NO_x$ emissions favour EIE between NO and $NO_2$, which also has a higher fractionation factor due to the lower temperatures (see Appendix B). Besides, this seasonal fluctuation of $f_{NO_2}$ can be expected to be observed on smaller time scales, typically on the diurnal scale in urban areas where NO is generally fully oxidised into $NO_2$ at night due to a lower $NO_x$ emission rate resulting in higher $f_{NO_2}$ at night than during the day, as observed at our sampling (Figure 2)."

- **Line 419: This is a bit hard to follow. Prior to Equation (4) $RO_2$ is defined as $RO_2 = HO_2 + CH_3O_2$. Equation 4 thus does not include reaction with $HO_2$ separately. But this line makes it appear as if $RO_2$ is being calculated from $HO_2$ alone. Please clarify.**

- **Line 421: The calculated $RO_2$ values are NOT consistent between cases A and B. This is even less true for SP 2 than SP 1. On average, and considering the variability one could argue they are not statistically significantly different, but at each time interval there are significant differences between case A and case B. This is noted in the following sentences, but this should be made clear from the start when you are discussing a mean versus a particular time period.**

- **Then again on line 438 it is stated the "closeness between $RO_2$ estimates using $\Delta^{17}O(NO_2)$ observations and those from empirical calculations…" – this is simply not true. The**

**concentrations calculated for the different cases are very different so deriving conclusions based on their sameness is inappropriate. Further, the sensitivity of $\Delta^{17}O(NO_2)$ to the chemical dynamics is not surprising and here it is what is being both hypothesized/tested and concluded, which is circular.**

With regard to your three comments above, concerns about this section were also raised by Reviewer #1, who suggested that the comparison with Case B should be removed because there is too much uncertainty about the calculated values and about the applicability of the empirical formula to our site. We agree with both comments and have removed the comparison with Case B in the revised manuscript which now only discusses $RO_2$ derived from isotopic measurements. Here the revised Section 3.2 treating this subject (lines 420–466):

[revised manuscript text omitted]

- **Line 471: the sentence beginning with "To note…" does not make sense. Please rephrase as I am unclear on the scientific meaning here. The contribution of blank versus sample should be better stated again. Not sure what "pondered by the mean" means.**

This section has been reworded/modified in the revised manuscript for more clarity as (lines 550-554):

"It is important to point out that, although the $NO_2$ sample collected on Feb 24 between 13:30 and 16:30 LT presents an important blank (ca. 14 %), ambient $NO_2$ is low during the sampling period (mean of $(4.5 \pm 1.8)$ nmol mol$^{-1}$). Therefore, as each $\Delta^{17}O$ value used to estimate $\Delta^{17}O_{calc}(NO_3^-)$ is weighted by the mean ambient $NO_2$ mixing ratio over the sampling period, the incertitude related to this blank has little influence on the daily average of $\Delta^{17}O_{calc}(NO_3^-)$."

- **Lines 476-478: I am interpreting here that the emphasis is on "processes" meaning the same expected chemistry for both nighttime sampling times. This seems overly simplified given that SP 2 was impacted by dust events and SP 1 was not. Is there no impact of dust on the nighttime chemistry? This should impact the reaction rates for $N_2O_5$. But also note that not all reactions are included in the case study here – for instance $NO_2$ hydrolysis, which could be impactful with heavier dust loading (see Alexander et al., 2020 for example). I think a bit more clarity is needed to separate the two case studies. If differences in conditions between the two sampling periods (i.e. gas and particle concentrations) are not being considered in the context of the chemistry then it challenges the conclusion that transported nitrate has to play a role in explaining the nighttime SP 2 data. (Note that I**

**think the additional nitrate from aloft is a good explanation for the higher than expected $\Delta^{17}O$, but this needs to be set up better).**

This section has been reworded/modified in the revised manuscript for more clarity as (lines 554-560):

"Given the low $\Delta^{17}(NO_3^-{}_{calc} - NO_3^-{}_{obs})$ during SP 1, observed $\Delta^{17}O(NO_3^-)$ can be explained by the local and rapid (< 12 h) oxidation of $NO_2$, dominated by the OH and $N_2O_5$ pathway during the day and night, respectively. However, in contrast to SP 1, the $^{17}O$-excess measured in $NO_3^-$ during the day of SP 2 cannot be fully constrained by the oxidation of surface $NO_2$ through the OH pathway, suggesting that the formation mechanisms of $NO_3^-$ are different between SP 1 and SP 2 and/or the presence of $NO_3^-$ not formed locally during SP 2. Below we examine the changes in the sub-daily dynamics of $\Delta^{17}O(NO_3^-)$ between SP 1 and SP 2 in light of atmospheric observations."

We had to analyse SP1 and SP2 with the same framework before finding a rather strong disagreement for SP2. We agree that it is very tempting to attribute the higher than expected $\Delta^{17}O$ for SP2 to additional nitrate originating from aloft but we cannot prove it. We also agree that Saharan dust can change/promote oxidation pathways to $NO_3^-$.

- **Line 515-516: Needs clarifying. The nitrate does not represent "deposition" so this phrasing needs to be changed here and through the rest of the manuscript. The samples represent aerosol (+gas) loaded onto a filter – so this does not represent deposition. Perhaps rephrase to "…during SP 2 may have increased the $NO_3^-$ loading aloft, in comparison to SP 1."**

Thank you for pointing this out, the sentence has been reworded accordingly in the revised manuscript.

*Section 4.3*

- **The interpretation of the oxygen isotopic composition discussed the likely potential for photolysis of HONO to add significantly to the local oxidant budget. This is not mentioned at all as part of the nitrogen isotopic composition discussion, but this process would also produce NO with a very different $\delta^{15}N$. Furthermore, vehicle emissions could also be adding HONO, given that they are concluded as the primary source of local $NO_x$ and $NO_3^-$. These pathways are neglected in the chemistry calculations and HONO impact on the $\delta^{15}N$ assumptions should be discussed or justified as to why be ignored.**

Please, see our response to your General comment #5. Besides, no measurement have been reported on the $\delta^{15}N$ of NO produced from HONO photolysis, therefore, we cannot state that it would be very different from dominant $NO_x$ sources at our site.

- **Line 554-555: This result only holds for GP 2, i.e. only EIE regime. This sentence should be more specific.**

The section has been reworded in the revised manuscript for more clarity as (lines 660-667):

"The close match between the calculated average $F_N$ of GP 2 and the observed $F_N$ (($43.6 \pm 3.3$) ‰; slope of the regression line in **Erreur ! Source du renvoi introuvable.**) provides strong evidence for the reliability of Eq. **Erreur ! Source du renvoi introuvable.**), as well as the expression of $\alpha_{EIE(NO_2/NO)}$ used therein, to accurately describe the N fractionation between $NO_x$ emissions and $NO_2$ at our site, and hence, to describe most of the variability of $\delta^{15}N(NO_2)$ measurements. This result holds significant importance in confirming the theoretical N isotopic fractionation framework used in prior research studies. It is also important to stress the influence of LCIE effects for GP 1, highlighting the high dependency of $\delta^{15}N(NO_2)$ to local environmental conditions. According to the $A^*$ factor, a greater influence of LCIE in mid-afternoon could have contributed to the outlying of the two samples collected between 13:30 and 16:30 LT (GP 1)."

- **Line 588-589: In fact, Miller et al., (2017) in an on-road study showed no dependence on most of these factors (except heavy emitters, e.g. presence of diesel trucks). Heaton, 1990 only measured vehicles with none of the modern catalytic convertor technology. Felix and Elliott (2014) measurements were taken in a tunnel and are not representative of on-road traffic. Walters et al. represents tailpipe measurements. Miller et al. and Zong et al. represent on-road and near-road sites. Please use more consistent studies and/or justify the use of averaged values across studies that are not similar and/or not representative of the current environment being studied.**

Thank you for pointing this out, it is a common issue in our research community. We acknowledge that the selected value may not be the most accurate to represent our sampling site. Nonetheless, it is difficult to determine the most appropriate value given the specific characteristics of our sampling site. Chamonix is located in a deep valley encompassed by high altitude mountains. As a result, emissions of $NO_x$ at the surface are confined, particularly in winter. Hence, it is likely that ambient $NO_x$ at our site in the early morning are affected by fresh emissions trapped in the inversion layer. These conditions differ from the Miller et al. (2017) study carried out in a non-montainous region during spring/summer and along densely used roads. Temperature conditions and vehicle uses are very different in Chamonix. Second, our sampling site is situated in a residential area, and thus, we anticipate the influence of fresh and cold $NO_x$ emissions in the morning when individuals leave their homes for work. Subsequently, during the day, we concur that $NO_x$ emissions from vehicles are likely to be more influenced by two closely situated and major roads, namely Départementale 1506 and Nationale 205 which connect to the Mont-Blanc tunnel. Therefore, we can expect that the $\delta^{15}N(NO_x)$ becomes close to that of Miller et al. (2017).

For these reasons, we think it is more pertinent to refer to the Zong et al. (2017) value and then interpret the variability around this value in light of the specific conditions experienced during the day. As stated in the manuscript, our average $\delta^{15}N(NO_x)$ derived from corrected $\delta^{15}N(NO_2)$ is similar to that of Zong et al. (2017). Besides, we observe a slight increase of $\delta^{15}N(NO_x)$ during the day, with the highest value around midday. This is consistent with the study of Miller et al. (2017) and is likely influenced by background $NO_x$ from traffic emissions of the two main roads situated a few hundred meters away. In contrast, the lower $\delta^{15}N(NO_x)$ observed in the early morning, when the inversion layer still exists, align with the study of Walters et al. (2015b) from tailpipe measurements.

In response to your concern regarding the studies selected by Zong et al. (2017) to derive their mean traffic $\delta^{15}N(NO_x)$ value, it is worth noting that a recent study by Song et al. (2022), adopted a similar method. However, Song et al. (2022) reviewed a greater number of studies, including Miller et al. (2017), in their calculations of the mean $\delta^{15}N(NO_x)$ for traffic. Song et al. (2022) obtained a very similar result to Zong et al. (2017) with a value of (-7.1 ± 4.1). Apologies for missing this study in the first version of the manuscript. Since Song et al. (2022) study takes into account more studies, we suggest to use their value as a reference for traffic $\delta^{15}N(NO_x)$. The revised manuscript includes this value as (lines 694-699):

"Regarding road traffic emissions, we have to stress that $\delta^{15}N$ values reported in the literature are rather variable mainly because N fractionations during the process of $NO_x$ production can vary depending on the type of fuel used, the type of vehicle, the presence of an emission control system, and the time of commuting (Ammann et al., 1999; Felix and Elliott, 2014; Heaton, 1990; Miller et al., 2017; Walters et al., 2015b; Zong et al., 2020, 2017). We use here the mean vehicle-emitted $\delta^{15}N(NO_x)$ value given by Song et al. (2022) at (–7.1 ± 4.1) ‰, calculated from 181 measurements reported in the literature."

and Figure 4 has been updated accordingly:

[Figure]

**Figure 4.** Time evolution of $\delta^{15}N(NO_x)$ (black dashed line) estimated from $\delta^{15}N(NO_2)$ observations in Chamonix after correction of N fractionation effects (length of horizontal line = sampling period, black shaded area = overall calculation error bar). Coloured shaded areas represent the standard deviation of the mean $\delta^{15}N$ value of individual $NO_x$ emission source (coal combustion in red, biomass burning in blue, vehicle exhaust in grey, fossil gas in orange, and soil emissions in green). Grey backdrop shaded areas represent the nighttime (sunset to sunrise).

Note that it does not change the conclusion regarding the dominant $NO_x$ source.

In your comment, you underline that cold starts are brief, yet they emit the most $NO_x$, which means that we ought to detect the highest levels of NO in the morning rather than in the late afternoon. This is confirmed by Figure 1 of the manuscript showing much higher NO mixing ratios in the morning. However, we also acknowledge that the large uncertainty on the heating oil $\delta^{15}N(NO_x)$ precludes a definite conclusion on our estimated $\delta^{15}N(NO_x)$ reflecting only traffic emissions. It is possible that heating oil emissions play a role. However, $\delta^{15}N(NO_x)$ of heating oil is not reported in the literature. The N content in distillate oil is negligible, similar to that of petrol, diesel, and fossil gas. This implies that the $NO_x$ emission during its combustion can be classified as "Thermal-$NO_x$" (in contrast with "Fuel-$NO_x$"; Miller and Bowman, 1989). Thermal $NO_x$ is generally depleted in $^{15}N$ owing to kinetic fractionation effects when the $N_2$ triple bond is broken, favouring the dissociation of the lightest isotopologue (Walters et al., 2015b). The mechanisms by which $NO_x$ is produced by thermal processes vary according to combustion conditions such as temperature, pressure, and oxygen availability. These conditions affect the N fractionation extent. It is therefore challenging to predict precisely the $\delta^{15}N$ footprint of $NO_x$ emitted from oil burner exhausts. It can be inferred that it would be near that of diesel vehicle exhausts and fossil gas burners i.e., approximatively somewhere between -17 ‰ and -13 ‰ (Walters et al., 2015a, b). Hence, the lower $\delta^{15}N(NO_x)$ values in the morning can also be due in part to heating oil emissions. These aspects are detailed in the manuscript.

- **Line 598-602: Some care needs to be taken here. It only takes about 30-50 seconds for an engine equipped with a catalytic convertor to be "warm" – in fact in the Walters' study Figure 2, the $NO_x$ emission plateaus after ~50 seconds. It is challenging to believe that this is having such a large impact on early morning versus later daytime values given the 3-hour time period over which the $\delta^{15}N$ values are collected. Perhaps the authors could do a back of the envelope type calculation to confirm whether this suggestion is valid. Cold engines will emit very high concentrations of NO for a very short period, so could potentially impact $NO_x$ loading – is there evidence of this in the gas phase measurements? i.e. given similar traffic conditions in the early morning and late in the day the concentrations should be much higher under "cold start" conditions.**

Please, see our response to your comment above.

- **Line 601: "removal" should be "replacement" (not sure how $NO_x$ removes $NO_x$ in this scenario?)**

Thank you for pointing this out, the word has been changed accordingly in the revised manuscript.

*Summary and implications*

- **Line 662: add "local" as in "…independently of the local $NO_2$ to $NO_3^-$ conversion processes."**

Thank you for pointing this out, the word has been added accordingly in the revised manuscript.

- **Line 663: In the framework here, dry deposition takes $NO_3^-$ out of the system so please rephrase. Do you mean that nitrate production is more likely or do you mean that the particle nitrate is heavy enough to sink and mix into the boundary layer?**

Thank you for pointing out this ambiguity, we actually want to refer to nitrate production through heterogeneous processes. The section has been modified/reworded in the revised manuscript as (lines 789-798):

"We use $\Delta^{17}O$ mass balance equations of $NO_3^-$ constrained by observed $\Delta^{17}O(NO_2)$ to assess whether $NO_3^-$ could originate locally from the oxidation of $NO_2$ at our site. During the first day of sampling, $\Delta^{17}O$ records of $NO_2$ and $NO_3^-$ support the local oxidation of $NO_2$ to $NO_3^-$ by OH radicals during the day, and via the heterogeneous hydrolysis of $N_2O_5$ during the night. The second day of sampling was affected by a Saharan dust event,

accompanied by notable changes in the isotopic composition of $NO_3^-$. We propose that the formation of a surface inversion layer at night could have influenced the vertical distribution of $\Delta^{17}O(NO_2)$ and resulted in a positive gradient of $\Delta^{17}O(NO_3^-)$ with altitude, independently of the local $NO_2$ to $NO_3^-$ conversion processes near the surface. In such scenario, the presence of Saharan dust could have promoted heterogeneous $NO_2$ oxidation leading to higher $\Delta^{17}O$ in $NO_3^-$ formed aloft. The latter would have then mixed with the $NO_3^-$ formed near the surface when the inversion breaks up during the day. Although still uncertain, the influence of the boundary layer dynamics on the distribution of $\Delta^{17}O$ in $NO_3^-$ should be investigated in the future, notably for urban areas in winter."

- **Line 682: perhaps add "…to test the hypotheses raised here." At the end of the line? It's not clear that there is evidence for a vertical gradient (yet) though the hypotheses raised in this work are certainly compelling.**

Thank you for this comment, the sentence has been reworded in the revised manuscript as (lines 818-820):

"In addition, the vertical distribution of $NO_2$ and $NO_3^-$ isotopic composition should be documented in order to explore the possible role of the boundary layer dynamics in the variability of $NO_2$ and $NO_3^-$ isotopic composition observed at the surface."

**Technical comments:**

- **Line 33: add () around NO and $NO_2$, i.e. (NO) … ($NO_2$) as they are being defined here**

Thank you for pointing this out, which has been corrected accordingly in the revised manuscript.

- **Line 49: the Leighton reference has a typo with the added "1961 citation"**

Thank you for pointing this out, which has been corrected in the revised manuscript.

- **Line 50: suggests adding in reaction #'s in text and () around $HO_2$ and $CH_3O_2$. For example, "This cycle can be disturbed by peroxy radicals (via R4; $RO_2$ = hydroperoxyl radical ($HO_2$) + methyl peroxy radical ($CH_3O_2$)) leading to formation of $O_3$ (via R2) (Crutzen, 1979)." Similarly R1-R3 should be called out in the text on lines 46-49.**

Thank you for pointing this out, which has been added accordingly in the revised manuscript.

- **Line 58: "Reaction R7" should be just "R7" (ie R implies the word "Reaction")**

We have followed the ACP notation which presents reaction numbers like this : Reaction (RX), Reactions (RX) to (RY), Reactions (RX)-(RY), and (Reaction RX). However, we have noticed a few typos in the manuscript which have been corrected according to this notation.

- **Line 101: "Authors" should be "The authors" or given the overlap in authors perhaps "We" ?**

Thank you for pointing out this error. Please note to that we have reworded and revised this paragraph in the revised manuscript (see our response above with the revised introduction).

- **Line 129: rephrase "The first study case" to This case study is the first to carry out concurrent multi-isotopic…."**

Thank you for pointing this out, please refer to our response above with the revised introduction.

- **Line 133: "conduction" should be "conducted"**

Thank you for pointing it out, which has been corrected in the revised manuscript.

- **Equation (6) and (7) should contain the subscripts day and night for NO₂ as well. i.e. make it clear that for the nighttime nitrate the NO₂ comes from Eq. 5.**

Thank you for pointing this out, which has been added accordingly in the revised manuscript.

**Can you also clarify why equation 5 uses $\Delta^{17}$O-NO₂ at night rather than $\Delta^{17}$O-O₃? The only production channel at night is assumed to be NO+O₃, with presumably 1 oxygen from ozone and 1 from NO. Is it just implicit that $\Delta^{17}$O-NO₂ formed at night = $\Delta^{17}$O-O₃? This seems weird to not make this explicitly clear.**

Equation 5 uses $\Delta^{17}O_{night}(NO_2)$ and not $\Delta^{17}O(O_3)$ because it includes the residues of $\Delta^{17}O_{day}(NO_2)$. $\Delta^{17}O_{night}(NO_2)$ would be equal to $\Delta^{17}O(O_3)$ only when $x$, the fraction of NO₂ formed during the day to the total NO₂ measured at night, would be equal to 0.

- **Line 246: "later" should be "latter"**

Thank you for pointing this out, which has been corrected in the revised manuscript.

- **Line 246-247: This should be referenced to Li et al (2020) here. (it is not original to this work here)**

We recognise that this is not original to our work, but neither it is original to that of Li et al. (2020), although we agree the latter was the first to investigate in details the three isotopic effects in the $NO_x$ cycle. KIE, EIE, an PHIFE result from the isotopic fractionation theory, therefore we have added as references in the revised manuscript the reviews that deal with these theoretical concepts, namely Miller and Yung (2000) and Young et al. (2002).

- **Line 250: "of the fraction of $NO_x$ in the form of $NO_2$" is awkward phrasing, please rephrase.**

Thank you for pointing this out, which has been reworded in the revised manuscript as (lines 311-313):

"A general expression for $\delta^{15}N(NO_2)$ can be derived as a function of a factor $F_N$ which represents the overall N isotopic fractionation effects between $NO_x$ emissions and $NO_2$ (expressed in ‰), the fraction of $NO_2$ with respect to $NO_x$ ($f_{NO_2}$ = [$NO_2$]/[$NO_x$]), and of $\delta^{15}N(NO_x)$ (Albertin et al., 2021; Li et al., 2020)"

- **Line 286: "schematises" is not correct – rephrase.**

Thank you for pointing this out, the sentence has been reworded in the revised manuscript as (lines 347-349):

"The dominant $NO_x$ to $NO_3^-$ conversion processes considered in this study, along with corresponding $\Delta^{17}O$ transfer factors and the known $^{15}N$ enrichment factors at 298 K (determined from both experimental and computational studies) are illustrated in Figure C1."

- **Line 275: [$NO_2$] in the denominator should be [NO], correct?**

[$NO_2$] being at the denominator is correct as ($1/k_{NO+NO_2}[NO_2]$) is the NO lifetime with respect to isotopic exchange with $NO_2$.

- **Line 309: suggest rephrasing to "…recovers and stay relatively low throughout the night…"**

Thank you for pointing this out, which has been corrected accordingly in the revised manuscript.

- **Line 344-345: rephrase here – $\Delta^{17}$O-NO$_2$ is ONLY similar for two of the time periods, not during the whole daytime. The observations at 13:30-16:30 at SP2 are questionable, and then the 16:30-18:00 samples are 3 per mil different. So overall, they are more different than similar for these time periods.**

This section has been modified/reworded in the revised manuscript as (lines 420-437):

"Over the course of SP 1 and SP 2, $\Delta^{17}$O(NO$_2$) shows a large diurnal variability (from 19.6 ‰ to 40.8 ‰) with a weighted mean ± one standard deviation of (25.2 ± 7.1) ‰. $\Delta^{17}$O(NO$_2$) values during the day (7:30−18:00 LT, (28.5 ± 7.3) ‰) are significantly higher ($p$-value = 0.002, n = 16) than during the night (18:00−7:30 LT, (20.8 ± 1.0) ‰). By day, $\Delta^{17}$O(NO$_2$) follows a similar increasing trend during SP 1 and SP 2, reaching a respective maximum of 40.8 ‰ between 13:30−16:30 LT and 35.0 ‰ between 10h30−13h30 LT. For both sampling periods, after sunset, $\Delta^{17}$O(NO$_2$) stabilises between 21:00 and 7:30 LT at ca. 20 ‰. Using the same sampling methodology in a mid-latitude urban area in spring, Albertin et al. (2021) reported very similar $\Delta^{17}$O(NO$_2$) values over the course of one day (20.5−39.2 ‰), following a comparable diurnal pattern. As presented above (Section 2.4.1), according to the ISS (isotopic steady state) framework, the variability of $\Delta^{17}$O(NO$_2$) reflects changes in the relative contributions of Reaction (R3) (NO + O$_3$) and Reaction (R4) (NO + RO$_2$) to the overall production of NO$_2$. At our site, $\Delta^{17}$O(NO$_2$) drops rapidly during the 16:30−18:00 LT interval to 23.3 ‰ and 20.9 ‰ during SP 1 and SP 2, respectively. Since, the isotope recycling rate in the NO$_x$-O$_3$ system is driven at first order by $J_{NO_2}$ (Michalski et al., 2014), due to low solar radiation between 16:30−18:00 LT at our site, such a rapid drop of $\Delta^{17}$O(NO$_2$) suggests that sampled NO$_2$ might not be at ISS anymore when the NO$_2$ photolysis is very slow, notably at the end of the day in winter. Therefore, we only consider that ISS holds between 7:30 and 16:30 LT in order to avoid the questionable end-of-the day measurements in our ISS-based analysis. At night (i.e., no ISS), the observed drop of $\Delta^{17}$O(NO$_2$) in the early evening reflects the rapid replacement of NO$_2$ formed during the day by NO$_2$ produced during the night via the conversion of freshly emitted NO, in line with Eq. (5). Then, high NO$_2$ throughout the night, along with relatively low O$_3$, supports observations of low $\Delta^{17}$O(NO$_2$) at night (i.e., ≈20 ‰)."

- **Line 358: Is this Zhang et al, 2022a or 2022b?**

Thank you for pointing this out, this is Zhang et al. (2022b), it has been corrected in the revised manuscript.

- **Line 368: add "from" after "Aside" – i.e ."Aside from the intrusion…"**

Thank you for pointing this out, which has been corrected accordingly in the revised manuscript.

- **Line 385: "targeted" should be "found"**

Thank you for pointing this out, which has been corrected accordingly in the revised manuscript.

- **Line 392: Add "observed" in the figure caption to be sure reader follows that the $\Delta^{17}O$ values here are observed not calculated.**

Thank you for pointing this out, which has been added accordingly in the revised manuscript.

- **Line 412: this could use some rephrasing – it sounds here as if ozone were high over the whole time period. Perhaps rephrase to make clear that ozone peaks at 16:30 LT but is not high through the whole time from 10:30 – 16:30 LT.**

Thank you for pointing this out. For more clarity, we have shortened the described period to 13:30–16:30 LT, which corresponds to the highest observed $O_3$ mixing ratios. See the reworded Section 3.2 of the revised manuscript in our response above.

- **Line 420: The equations number in the text do not match those in the Table 1 footnotes.**

Thank you for pointing this out, which has been corrected in the revised manuscript.

- **Line 433: rephrase to "could be a significant source of VOCs…."**

Thank you for pointing this out, which has been rephrased in the revised manuscript.

- **Line 423: "…could be due to the important blank associated with this sample." The real issue here is the % contribution of the blank versus sample so perhaps rephrase to make that clear; currently it is not clear what "important blank" means.**

Thank you for pointing this out. This sentence has been removed because we have suppressed study case B. See the modified/reworded Section 3.2 of the revised manuscript in our response above.

- **Line 427: "The highest contribution of $RO_2$…is correlated with the highest NO levels." For this to be stated it must be shown to be true. The highest NO levels occur around 10:30-11am at both SP1 and SP2. This is not the time of the highest $RO_2$ concentrations. This IS the time of the lowest $T_{NO+O_3}$ which means that the relative contribution of $RO_2$ is high, but this is not the time when NO levels are highest. Note too that nothing is shown in terms of the sensitivity of $\Delta^{17}O$ to $RO_2$ concentration – i.e. calculated $\Delta^{17}O$ values are not directly compared with the calculated $RO_2$ concentrations.**

NO levels increase from 6:30 to 10:30 during SP 1 and SP 2, peak at around 10: 30 and then decrease. We agree that the maximum is not reach exactly at 10:30 but somewhere between 10:00 and 11:00 because the given average NO mixing ratio represents the average of 1 min measurements between 10:00 and 11:00. Still, if the denuder sampling periods are used (7:30-10:30 and 10:30-13:30), the highest average NO is between 7:30-10:30 because NO mixing ratio drops sharply from 11:00 onwards. In addition, we explicitly state that this is the contribution of $RO_2$ and not its concentration. The section has been reworded for greater clarity in the revised manuscript (see the revised Section 3.2 in our response above).

- **Line 440: rephrase to "be used to improve understanding of the oxidation processes …"**

Thank you for pointing this out, which has been reworded accordingly in the revised manuscript.

- **Line 458: $\Delta^{17}O$-$NO_2$ should be $\Delta^{17}O$-$NO_3$ here (with the subscript $NO_2+O_3$)**

Thank you for pointing this mistake out, the sentence has been reworded in the revised manuscript as (line 464):

"We think that our method may be very valuable in deciphering oxidation processes of $N_r$ species, down to sub-daily temporal scales."

- **Line 471: the formatting for the $\Delta^{17}O$ changes here**

Indeed, this is because this $\Delta$ symbolises a difference between the two same quantities, $\Delta^{17}O(NO_3{}^-_{calc})$ and $\Delta^{17}O(NO_3{}^-_{obs})$, while $\Delta$ symbolises the deviation of $\delta^{17}O$ to a reference value ($0.52 \times \delta^{17}O$).

- **Line 497: "in" should be "on" average**

Thank you for pointing this mistake out, which has been corrected accordingly in the revised manuscript.

- **Line 498: add were as in "values were more homogenous"**

Thank you for pointing this mistake out, the sentence has been reworded in the revised manuscript as (line 581):

"In summer, the $\Delta^{17}O(NO_3^-)$ values at the three altitudes were very similar."

- **Line 499: "attitude" should be "altitude"**

Thank you for pointing this mistake out, which has been corrected in the revised manuscript.

- **Line 525: "informations" should be "information"**

Thank you for pointing this mistake out, which has been corrected accordingly in the revised manuscript.

- **Line 564: the overall mean value stated here does not match what is in the Table.**

This is because is it the mean over GP #1 and GP #2, this information has been added to the in the revised manuscript for more clarity.

- **Line 585: Different papers seems to use different values for biomass combustion. Looking back at Fibiger and Hastings, 2016 the results indicate that latitudinal difference in $\delta^{15}N$ biomass exist and thus so should latitudinal difference in $\delta^{15}N$-NO from this source. Is that how Martinelli et al., 1999 is being used here? Please explain the values used if they do not directly represent the citations.**

We thank you for pointing this out and apologise for any lack of clarity. Similar to the choice of $\delta^{15}N$ of traffic emission, choosing a $\delta^{15}N$ for biomass burning is not straightforward regarding the variability observed in different studies. This is primarily due to the type of wood burnt. Our approach, which we agree is sorely lacking in detail in the initial manuscript, was to use an empirical relationship reported by Fibiger and Hastings (2016) between $\delta^{15}N(NO_x)$ emitted by biomass combustion and the $\delta^{15}N$ of the burnt vegetation: $\delta^{15}N(NO_x) = 0.98 \times \delta^{15}N_{biomass} + 1.7$. Regarding the $\delta^{15}N_{biomass}$ value, we decided to

take the average value given by Martinelli et al. (1999), $(-2.8 \pm 2.0)$ ‰ which represents an average over a large number of $\delta^{15}N$ measured in temperate forests.

However, following your comment, we would like to mention a more recent study, Chai et al. (2019), which is very similar to that of Fibiger and Hastings (2016). They confirmed the close linear relationship between biomass combustion $\delta^{15}N(NO_x)$ and $\delta^{15}N_{biomass}$, and combined their work with that of Fibiger and Hastings (2016) to obtain a new linear relationship, namely: $\delta^{15}N(NO_x) = (0.42 \pm 0.17) \times \delta^{15}N_{biomass} + 1.3$. Therefore, in order to account for more recent studies, we propose to use this relationship to derive a new mean value of biomass combustion $\delta^{15}N(NO_x)$, which is now at $(0.1 \pm 1.3)$ ‰, which is anyway very close to our initial value. More details are given in the revised manuscript as (lines 689-693):

"$\delta^{15}N$ of $NO_x$ released during biomass combustion is primarily driven by the $\delta^{15}N$ of the biomass burnt (Fibiger and Hastings, 2016). We estimate an average $\delta^{15}N$ of biomass combustion $NO_x$ at $(-0.1 \pm 1.3)$ ‰, using the empirical relationship of Chai et al. (2019) (which was derived from combustions of several North American wood species) and an average $\delta^{15}N$ of biomass at $(-2.8 \pm 2.0)$ ‰ representative of temperate forests (Martinelli et al., 1999)."

- **Line 627: what is the meaning of "faction" here?**

Thank you for pointing this out, this is a typing mistake of the word "factor". This section has been reworded in the revised manuscript for more clarity as (lines 741-744):

"Neglecting KIE associated with the $N_2O_5$ pathway and using the expression of the EIE fractionation factor between $N_2O_5$ and $NO_2$ given by Walters and Michalski (2015) (Appendix D) constrained with the mean nighttime temperature at our site, the isotopic composition of $NO_3^-$ is expected be enriched in $^{15}N$ by about 29 ‰ compared to $NO_2$. This estimated $^{15}N$ enrichment is about three times higher than the observed $\Delta^{15}(NO_3^- - NO_2)$ at our site."

- **Line 635: rephrase to "associated with a KIE effect…"**

Thank you for pointing this mistake out, which has been corrected in the revised manuscript.

- **Line 637: rephrase to "with the OH pathway"**

Thank you for pointing this mistake out, which has been corrected in the revised manuscript.

---

## Author Response (AR2)

**Response to the referee #1 (egusphere-2023-744)**

Dear Lei Geng,

We thank you for your positive review of the revised version. We agree that a more accurate calibration of $\Delta^{17}O(NO_2)$ measurements is desirable. As you know, this work is currently underway and we have no further elements to report at the moment.

The values of $\Delta^{17}O(NO_2)$ observed in Chamonix are in good agreement with previous observations in Grenoble and in both studies, the $\Delta^{17}O(NO_2)$ maximum and minimum values are fully consistent with the $\Delta^{17}O$ theoretical framework and measured $\Delta^{17}O(O_3)$ values for the cases when NO conversion to $NO_2$ is expected to be overwhelmingly dominated by ozone. Therefore, we think that, though not very accurate, the $\Delta^{17}O(NO_2)$ calibration is not a significant issue here. Nonetheless, we cannot totally rule out the possibility that future measurements of nitrite $\Delta^{17}O$ references might differ from our actual calibration data. If this should happen, we will publish a correction to the paper with updated values and possible implications.

---

## Author Response (AR3)

**Response to the editor**

Dear Markus Ammann,

Thank you for accepting the paper for publication. As you recommended, we have added to the main text an awareness of the current calibration of $\Delta^{17}O(NO_2)$ measurements as (Section 3.2, line 497-507 in the revised manuscript):

"It is worth pointing out that a more accurate calibration of $\Delta^{17}O(NO_2)$ measurements is desirable. There is currently no internationally accepted nitrite salt standard with a positive $\Delta^{17}O$. Nevertheless, we are rather confident in our present calibration methodology. Indeed, the $\Delta^{17}O(NO_2)$ values measured in Chamonix closely align with previous observations in Grenoble. In both studies, the maximum (daytime) and minimum (nighttime) $\Delta^{17}O(NO_2)$ measurements conform to the expected values derived from the $\Delta^{17}O$ theoretical framework (based on well-established $NO_x$ chemistry and $\Delta^{17}O(O_3)$ measurements) when $O_3$ overwhelmingly dominates the NO to $NO_2$ conversion. Therefore, at this stage, we do not consider that $\Delta^{17}O(NO_2)$ calibration is an issue for this study. However, as we cannot completely rule out a small bias in our calibration, we have manufactured enriched nitrite salts and are presently working on refined new nitrite salt standards. Note that the rapid exchange of nitrite O isotopes in aqueous solution (Casciotti et al., 2007) is a challenge for inter-laboratory data comparisons. If new measurements of nitrite $\Delta^{17}O$ references are found to differ from our current calibration, we plan to publish a correction to the paper with updated values and any potential implications."

Best regards

Sarah Albertin